# nnActive: A Framework for Evaluation of Active Learning in 3D Biomedical Segmentation

**Carsten T. Lüth**[1,2,3*], **Jeremias Traub**[1,2,4*], **Kim-Celine Kahl**[1,2,3], **Till Bungert**[1,2,3],
**Lukas Klein**[1,2,5], **Lars Kraemer**[1,2,3], **Paul F. Jaeger**[2,6],
**Fabian Isensee**[1,2,3†], **Klaus Maier-Hein**[1,2,3,7,8†]

[1]*German Cancer Research Center (DKFZ) Heidelberg, Division of Medical Image Computing, Germany*
[2]*Helmholtz Imaging, German Cancer Research Center (DKFZ), Heidelberg, Germany*
[3]*Faculty of Mathematics and Computer Science, University of Heidelberg, Germany*
[4]*German Cancer Research Center (DKFZ) Heidelberg, Division of Intelligent Medical Systems, Germany*
[5]*Institute for Machine Learning, ETH Zürich, Switzerland*
[6]*German Cancer Research Center (DKFZ) Heidelberg, Interactive Machine Learning Group, Germany*
[7]*Pattern Analysis and Learning Group, Department of Radiation Oncology, Heidelberg University Hospital, Germany*
[8]*National Center for Tumor Diseases (NCT) Heidelberg, Germany*

*{carsten.lueth, jeremias.traub}@dkfz-heidelberg.de*
*\*/†: These authors contributed equally to this work.*

**Reviewed on OpenReview:** *https://openreview.net/forum?id=AJAnmRLJjJ*

## Abstract

Semantic segmentation is crucial for various biomedical applications, yet its reliance on large annotated datasets presents a significant bottleneck due to the high cost and specialized expertise required for manual labeling. Active Learning (AL) aims to mitigate this challenge by selectively querying the most informative samples, thereby reducing annotation effort. However, in the domain of 3D biomedical imaging, there remains no consensus on whether AL consistently outperforms Random sampling strategies. Current methodological assessment is hindered by the wide-spread occurrence of four pitfalls with respect to AL method evaluation. These are (1) restriction to too few datasets and annotation budgets, (2) training 2D models on 3D images and not incorporating partial annotations, (3) Random baseline not being adapted to the task, and (4) measuring annotation cost only in voxels. In this work, we introduce nnActive, an open-source AL framework that systematically overcomes the aforementioned pitfalls by (1) means of a large scale study evaluating 8 Query Methods on four biomedical imaging datasets and three label regimes, accompanied by four large-scale ablation studies, (2) extending the state-of-the-art 3D medical segmentation method nnU-Net by using partial annotations for training with 3D patch-based query selection, (3) proposing Foreground Aware Random sampling strategies tackling the foreground-background class imbalance commonly encountered in 3D medical images and (4) propose the foreground efficiency metric, which captures that the annotation cost for background-compared to foreground-regions is very low. We reveal the following key findings: (A) while all AL methods outperform standard Random sampling, none reliably surpasses an improved Foreground Aware Random sampling; (B) the benefits of AL depend on task specific parameters like number of classes and their locations; (C) Predictive Entropy is overall the best performing AL method, but likely requires the most annotation effort; (D) AL performance can be improved with more compute intensive design choices like longer training and smaller query sizes. As a holistic, open-source framework, nnActive has the potential to act as a catalyst for research and application of AL in 3D biomedical imaging. Code is at: https://github.com/MIC-DKFZ/nnActive

# 1 Introduction

Semantic segmentation is vital for numerous biomedical applications, including the delineation and detection of structures and pathologies in computed tomography scans (CT), magnetic resonance imaging (MRI), and microscopy images. With the advent of deep learning, training-based approaches that require large annotated datasets have become the de facto standard for semantic segmentation. This trend is particularly evident in 3D medical imaging, where U-Net-like architectures like nnU-Net (Isensee et al., 2021) have received significant emphasis.

While there is an increasing number of medical scans created each day and stored in large databases (Smith-Bindman et al., 2019), the cost of data annotation remains one of the main obstacles when solving specific 3D biomedical tasks, due to the high manual effort required to create segmentation masks. This is particularly drastic for biomedical images, as specialized personnel have to carry out the annotation.

The promise of Active Learning (AL) is to reduce this cost of annotation by only querying the most informative samples to be annotated for the task. This reduction in annotation cost needs to be weighed against an increase in computational and setup costs stemming from the use of AL. Therefore, to justify a general recommendation for an AL method, it needs to reliably bring performance benefits over computationally cheap annotation strategies like Random sampling in multiple 'realistic scenarios' that are substantial enough to ensure amortization of its cost increases during application (Lüth et al., 2023; Munjal et al., 2022; Mittal et al., 2023).

While there are various studies on AL for 2D image and video segmentation (Mittal et al., 2023; Mackowiak et al., 2018), many open questions remain about its effectiveness in the 3D biomedical domain, largely due to fundamental differences between the 2D and 3D domains. Most importantly, the high annotation cost and the high redundancy in 3D image data, e.g. the similarity of neighboring areas, necessitate more efficient annotation strategies such as partial annotations, in the form of slices or patches. In addition, 3D biomedical images commonly have a background class that occupies most of the image, standing in stark contrast to the dense multiclass masks frequent in 2D natural image semantic segmentation tasks (Cordts et al., 2016).

The AL community lacks a common benchmark for the 3D biomedical domain as it is highly scattered in terms of evaluation practices (see table 1), making it nearly impossible to directly compare results across papers. Most importantly, there is no consensus on whether employing AL methods leads to reliable performance improvements over Random sampling. Many studies indicate that AL methods do not always outperform Random sampling (Nath et al., 2021; Gaillochet et al., 2023a;b; Föllmer et al., 2024; Vepa et al., 2024; Burmeister et al., 2022) and also commonly emphasize that it remains a surprisingly strong baseline (Nath et al., 2021; Burmeister et al., 2022). Further, Burmeister et al. (2022) concluded that AL methods do not reliably outperform advanced Random sampling strategies where the sampling is adapted to the 3D structure of the data. Crucially, this is the only work investigating improved Random baselines. When taking further into account that most studies neither make use of standardized segmentation models, proven to bring state-of-the-art performance, nor 3D models making explicit use of partial annotations during training. With both of these points potentially substantially reducing overall model performance, it is apparent that the current evaluation protocol does not allow for making practically relevant and generalizing assessments based on which a practitioner can make an informed decision whether to employ AL or not.

In response, this work introduces a novel framework for performance evaluation of 3D biomedical AL for semantic segmentation. It systematically addresses shortcomings in prior work, formalized as four pitfalls, by employing best practices for general AL evaluation. These practices are extended to account for the specific properties of 3D biomedical segmentation, enabling potential practitioners and developers to better assess the expected performance improvements when utilizing AL in a close-to-production scenario. Concretely, our contributions are:

1. We provide nnActive, a highly configurable AL extension for nnU-Net using partial annotations in the form of 3D patches that ensure state-of-the-art segmentation performance and out-of-the-box adaptation to new segmentation tasks.
2. We introduce Foreground Aware Random sampling as a stronger, more realistic baseline, which tackles the class imbalance typically encountered in 3D images.

3. We perform the largest study to date of AL methods with a specific focus on uncertainty based Query Methods (QMs), encompassing over 7500 nnU-Net trainings on 12 dataset-settings from four different datasets with three respective Label Regimes for each dataset, alongside numerous ablations to allow a holistic view of AL performance benefits.
4. We propose Foreground Efficiency (FG-Eff), a novel metric measuring annotation efficiency which takes into account that annotating background has a negligible annotation effort compared to foreground, setting it apart from other metrics using voxels as a proxy for annotation effort.

## 2   Requirements of Active Learning Evaluation

In its very foundation, AL represents a wager where an *expected reduction in annotation cost* is weighed against the additional experimental setup and compute costs induced by it. The expected annotation cost reduction can be estimated from the annotation cost and the expected performance gains. Finding the best among multiple AL methods for a real-world use case entails spending the annotation budget multiple times, leading to a net increase in annotation effort. Therefore, benchmarking AL in a use-case scenario directly contradicts the purpose of using AL in the first place. This is also referred to as 'validation paradox', as detailed in Lüth et al. (2023).

Therefore, the evaluation of AL methods must ensure that the measured performance gains are generalizable and practically relevant (Munjal et al., 2022; Zhang et al., 2024; Mittal et al., 2019; Lüth et al., 2023; Mittal et al., 2023; Zhan et al., 2022). To that end, the following four requirements (R1-R4) need to be taken into account ensuring:

**R1** Generalization over datasets, annotation budgets, and query parameters.
**R2** Performance gains in combination with orthogonal approaches increasing annotation efficiency e.g. Self- or Semi-Supervised Learning.
**R3** Performance gains over computationally cheap methods, such as improved variants of Random sampling.
**R4** Reduction in annotation effort of AL methods is measured.

The implementation of these requirements depends on the tasks (e.g. semantic segmentation or object recognition) and domains in which AL is applied. Each requirement maps to one pitfall and its implications are detailed for 3D biomedical imaging in section 3.

Finally, to prevent overfitting on development datasets, the best-performing AL methods should also be evaluated on separate held-out test datasets that are independent of the development datasets as in a roll-out scenario, ensuring a generalization gap to previously unseen datasets before widespread adoption.

## 3   Pitfalls and Solutions for a Systematic Validation of Active Learning Methods in 3D Biomedical Semantic Segmentation

Based on the requirements (R1-R4) stated in section 2 necessary to build a generalizable AL method, we discovered four respective pitfalls (P1-P4) in the evaluation protocols of the related literature for AL in the 3D biomedical domain, which hinder generalizable and reliable performance assessments. We emphasize that our goal here is not to assign blame but to highlight the importance of evaluation, as inadequate assessment can obscure which methods are truly the most effective, especially for potential practitioners getting first contact with AL. In table 1 the prevalence of these pitfalls is shown in the related literature alongside important design parameters. We address these pitfalls by building the nnActive framework and performing a large scale empirical study following the design solutions we detail here.

**P1: Evaluation is restricted to too few settings.** Evaluating AL methods on a wide variety of datasets and multiple different annotation budgets is becoming common practice in AL for classification (Lüth et al., 2023; Mittal et al., 2023; 2019; Zhang et al., 2024) and also becomes incorporated into 2D Semantic Segmentation (Mittal et al., 2023). Only by doing so is it possible to obtain generalizing performance estimates, as the only benefit of an AL method lies in its ability to generalize to novel scenarios during application. For example, in practice, it may not be clear what *an adequate annotation budget to avoid*

Table 1: Overview of the related work in AL for 3D biomedical image segmentation with regard to the described Pitfalls P1-P4 and key parameters. Retraining indicates whether a model is trained for each AL loop from a standard initialization. ✔ indicates addressed, (✔) partially addressed, and ✗ indicates unaddressed pitfalls. N/A is given, as in the experimental setup, this Pitfall can not occur. N.S. indicates an unspecified value in the manuscript and code. A detailed description of our rating is given in appendix A.

| | Query Design | P1 | P2 | P3 | P4 | #Datasets | Model | Retraining | #Seeds |
|---|---|---|---|---|---|---|---|---|---|
| Nath et al. (2021) | 3D Image | ✗ | ✗ | N/A | N/A | 2 | 3D U-Net | yes | 5 |
| Burmeister et al. (2022) | 2D Slice | (✔) | ✗ | ✔ | ✗ | 3 | 2D U-Net | no | 3 |
| Gaillochet et al. (2023a) | 2D Slice | (✔) | ✗ | ✗ | ✗ | 2 | 2D U-Net | yes | 5 |
| Gaillochet et al. (2023b) | 2D Slice | ✗ | (✔) | ✗ | ✗ | 1 | 2D U-Net | yes | 5 |
| Ma et al. (2024) | 2D Slice | ✗ | ✗ | ✗ | ✗ | 2 | 2D U-Net | no | N.S. |
| Föllmer et al. (2024) | 2D Slice | (✔) | ✗ | ✗ | ✗ | 3 | 2D nnU-Net | no | 2 |
| Vepa et al. (2024) | 2D Slice | ✔ | (✔) | ✗ | ✗ | 3 | 2D U-Net | yes | 5 |
| Shi et al. (2024) | 2D Slice | (✔) | ✗ | ✗ | ✗ | 4 | 2D U-Net | no | 5 |
| Ours | 3D Patch | ✔ | ✔ | ✔ | ✔ | 4 | 3D nnU-Net | yes | 4 |

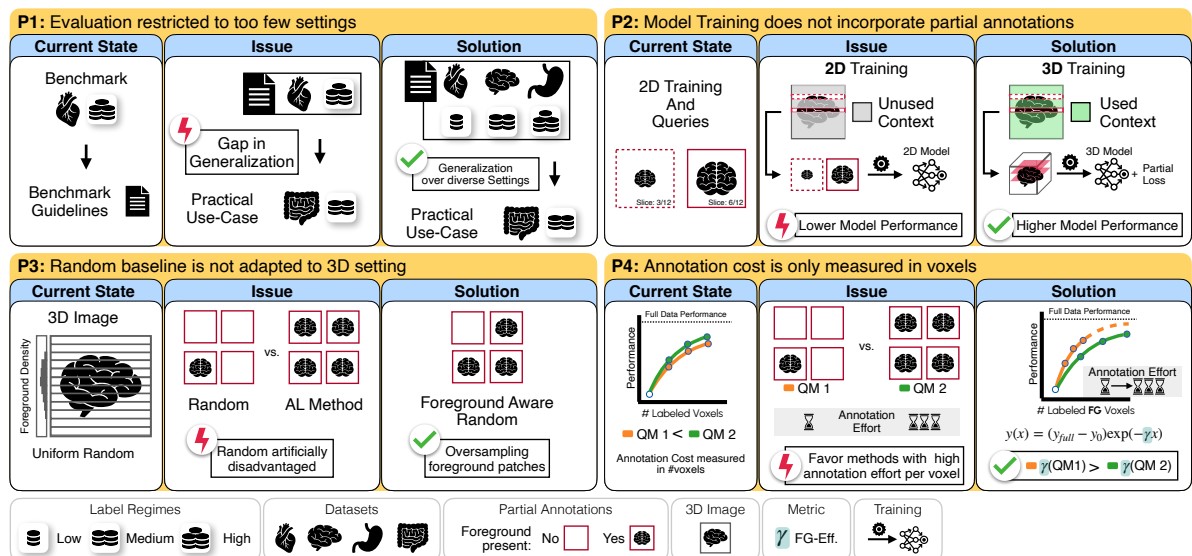

Figure 1: Visualization of the four Pitfalls (P1-P4) alongside our solutions and how their presence hinders reliable performance assessments of AL methods for 3D biomedical imaging. For visualization purposes, we use 2D slices as partial annotations.

*cold-start*, indicated by AL methods being outperformed by Random sampling due to insufficient model performance (Gao et al., 2020). Only by evaluating AL over multiple different annotation budgets can the cold-start problem be characterized. **State:** Currently, the amount of 3D biomedical datasets used is severely limited, with more than half of the related works using less than two datasets (Nath et al., 2021; Gaillochet et al., 2023b; Ma et al., 2024). Further, this evaluation is usually limited to one fixed annotation budget (Nath et al., 2021; Burmeister et al., 2022; Gaillochet et al., 2023a;b; Ma et al., 2024; Föllmer et al., 2024; Shimizu et al., 2024). Only Vepa et al. (2024) and Gaillochet et al. (2023a) evaluate multiple different annotation budgets for at least one dataset.

**Proposed solution:** We translate the best practices for AL evaluation proposed by Lüth et al. (2023) into a framework for method development and benchmarking, focusing on a wide variety of medical imaging tasks, including multi-organ, tumor, fine-grained, pathological, and non-pathological segmentation tasks. On each of these datasets, we perform experiments on three separate annotation budgets termed Low-, Medium- and High-Label Regime to ensure a holistic view of the performance of AL methods. Further, we perform multiple ablations with regard to the query size and query patch size.

**P2: Model Training does not incorporate partial annotations.** For 3D Images, image-based query selection is not suitable, given the immense costs for annotating a whole image. In addition, biomedical datasets usually consist of only a few individual images. However, partial annotations, such as 2D slices or 3D patches (see appendix B for a definition), are remarkably rich in information w.r.t. the labels of neighboring areas. This is largely due to the inherent homogeneity present in 3D biomedical images. Explicitly using partial annotations for training the models while making use of unlabeled contextual information significantly decreases the amount of annotated data required to achieve performance comparable to training on the entire dataset without necessitating pretrained models or often compute intensive semi-supervised training (Gotkowski et al., 2024). **State:** Most related works train 2D models on slice-based queries (Burmeister et al., 2022; Gaillochet et al., 2023a;b; Ma et al., 2024; Vepa et al., 2024; Shimizu et al., 2024). By training only on the 2D partial annotations, their surrounding context is discarded, which reduces the overall annotation efficiency. Two approaches to reducing annotation effort include Vepa et al. (2024), who train on 2D scribble annotations and use pretrained models, and Gaillochet et al. (2023b) who use Semi-Supervised pretraining. In addition, several other related works point out that using well-configured models is another simple way to reduce annotation effort (Lüth et al., 2023; Munjal et al., 2022; Mittal et al., 2023; 2019). Given the dominance of nnU-Net in 3D biomedical imaging on both benchmarks and challenges (Isensee et al., 2021; 2024) we would like to emphasize the work by Föllmer et al. (2024), who proposed an AL integration for nnU-Net that, however, only supports 2D training data and queries.
**Proposed solution:** We employ 3D nnU-Net models and train with the partial loss Gotkowski et al. (2024) on our queried 3D partial annotations alongside a specific sampling method ensuring that the partial annotations are used during training with sufficient surrounding area as context. By utilizing the automatic configuration of nnU-Net, we further ensure that our models are well configured for each dataset.

**P3: Random Baseline is not adapted to 3D setting.** In 3D biomedical image segmentation, many datasets have a background class and structure of interest (e.g. organ, tumor) that often occupies a small portion of the images and is located in a specific area. Based on this, the standard Random baseline is artificially disadvantaged when combined with partial annotations. It often queries image regions that are purely background, which require minimal annotation effort. This issue is already known for 2D slices (Ma et al., 2024). Further, specific classes occupy smaller regions in the image than others, leading to a selection bias favoring large classes, which leads to a strong selection bias towards larger classes. Given that the primary challenge in manual voxel-wise annotation lies in delineating the structures rather than identifying their rough location, random image selection with oversampling of foreground areas by drawing random classes is a feasible approach in practice. Alternatively, using information about the inherent structure of 3D biomedical tasks allows for multiple improved Random strategies. **State:** Burmeister et al. (2022) evaluate advanced Random strategies using stratified sampling (e.g. Strided) based on the 3D structure of the data and conclude based on its performance that Random and Strided sampling may be sufficient for many use cases. This is especially concerning given that all works acknowledge the surprising toughness of beating Random baseline (Nath et al., 2021) or many benchmarked AL methods underperforming and or being solely on par with Random (Gaillochet et al., 2023a; Ma et al., 2024; Föllmer et al., 2024). This leads to the question of how improved Random baselines would have changed the verdict of other works from 'AL is beneficial' into another direction.
**Proposed solution:** We employ additional Foreground Aware Random strategies, which simulate screening an image for a a random foreground class and ensure that foreground is present in a specific percentage of all queries. This also ensures that the class distribution of queries is diversified across classes. For more information, we refer to section 4.

**P4: Annotation Cost is only Measured in Voxels.** Measuring annotation effort of two competing QMs purely based on voxel based metrics does not capture that substantial differences can arise based on the structures with the queries. For example, a query completely consisting of background with minimal structure, requiring minimal annotation effort, has the same number of voxels as a query containing multiple structures of interest that need to be delineated with a large amount of effort. Therefore evaluation methods purely using voxel based metrics measuring annotation effort can lead to a systematic bias favoring QMs which require a large annotation effort per queried voxel. **State:** To our knowledge, none of the related work takes this factor into account with any explicit measurement or discusses this behavior as problematic.

**Proposed solution:** We measure the annotation efficiency by proxy of the amount of foreground annotation using the decay parameter $\gamma$, we term Foreground Efficiency (FG-Eff). It stems from an exponential decay fitted to the performance gap to a model trained on the entire dataset against the number of foreground voxels. Higher values indicate that a QM is more annotation efficient as it converges faster to the performance obtained when training on the entire dataset (example given in fig. 1). Due to its nature, the FG-Eff only allows meaningful comparisons of QMS across a single Label Regime with identical training setups. As the number of foreground voxels represents a proxy for annotation effort, the FG-Eff does not replace other performance metrics but should be seen as an extension of them. For more details on the FG-Eff with a mathematical definition and its interpretation, we refer to appendix D.

## 4    nnActive Framework & Study Setup

The entire study is based on our proposed nnActive framework, an extension of nnU-Net for AL with 2D and 3D biomedical semantic segmentation enabling querying 3D patches with AL methods. We focus on 3D patch-based AL to keep the general framework versatile and because 3D patches can be annotated with multiple different strategies, e.g. dense or sparse with slice annotation. The design of nnActive allows it to be used directly in combination with the standard nnU-Net framework for both benchmarking[1] and application. Hence, it allows easy implementation of future methodological developments in the benchmarking and application of AL since nnU-Net (Isensee et al., 2021; 2024) is often extended with an ecosystem of projects built directly on top (Gotkowski et al., 2024; Roy et al., 2023).

We now present the overall design of the nnActive framework, along with study-specific design choices made for our benchmark evaluation.

**Model Architecture and Training Strategy**   We use nnU-Net (Isensee et al., 2021), a self-configuring deep learning framework, as our segmentation model. However, we enhanced the standard patch-based model trainer through region sampling, enriching the region observed by the model with additional unlabeled context from the rest of the image. **Study Specific:** We used the `3D full resolution` configuration of nnU-Net and trained for 200 epochs. To increase model robustness, we used an ensemble of five models trained via 5-fold cross-validation, as previous research has demonstrated that ensembles improve AL performance by providing more reliable uncertainty estimates (Beluch et al., 2018; Kahl et al., 2024). Further, we perform complete retraining of the models for each AL loop as finetuning leads to reduced model performance (Beck et al., 2021; Ash & Adams, 2020) presumably due to the model getting stuck in a local optimum. The training of the models themselves is not seeded, but all dataset-related parameters are. All experiments were averaged over four seeds.

**3D Query Methods**   We implemented the QMs for the 3D volumetric data in two steps. First, we draw a set of *best patches* with a maximum allowed overlap ($o$) with respect to previously drawn patches for each image using an uncertainty function, followed by an aggregation method. In the second step, the final query is drawn from the patches of the entire training & pool dataset. An example of an uncertainty-based QM is given in algorithm 1. **Study Specific:** We evaluate the following 8 QMs in our study, described in the following two paragraphs, with no allowed overlap ($o = 0$) between patches.

**AL Query Methods** We implemented the following five uncertainty-based AL QMs 1) Predictive Entropy (Settles, 2009), 2) Bayesian Active Learning by Disagreement (BALD) (Houlsby et al., 2011; Gal et al., 2017), 3) PowerBALD (Kirsch et al., 2023), 4) SoftrankBALD (Kirsch et al., 2023), and 5) PowerPE (Kirsch et al., 2023). Both Predictive Entropy and BALD greedily select the top-k uncertainty scores and are therefore referenced as 'Greedy'. PowerBALD, PowerPE, and SoftrankBALD use a selection mechanism with additional noise perturbations, which promotes the diversity of the samples and are therefore referenced as 'Noisy'. **Study Specific:** We use for all QMs a mean aggregation where the aggregation size is equal to the query patch size and set the $\beta$-parameter for PowerBALD, SoftrankBALD and PowerPE to 1 as proposed by Kirsch et al. (2023).

---

[1]When extending our results we highlight the importance of using our exact nnU-Net version to ensure compatibility

**Random Strategies** We use three random strategies as baselines: 1) the standard Random sampling baseline and two more advanced **Foreground Aware Random strategies**, 2) Random 33% FG, and 3) Random 66% FG. The Random 66% FG baseline selects a completely random patch with a probability of 33%, while the remaining 66% of patches prioritize regions containing anatomical structures with foreground oversampling, i.e. where half of the patches are centered on a randomly chosen foreground class and the other half are centered on the border of a foreground class. Similarly, the Random 33% FG baseline increases the proportion of fully random selections to 66%, while 33% of patches are drawn with the aforementioned foreground oversampling. These modifications ensure that Random baselines remain a fair point of comparison by accounting for the natural biases present in medical imaging data.

**Evaluation Metrics** The general model performance is evaluated using the Mean Dice Score (per 3D image) (Dice, 1945). We use four different metrics, whereby the first three metrics allow relative comparisons of QMs within individual Label Regimes: 1) the Mean Dice score of the final AL loop (Final Dice), 2) the Area Under Budget Curve (AUBC) (Zhan et al., 2021; 2022) aggregating the Mean Dice scores over all AL loops, 3) our proposed FG-Eff measure which is a proxy for the annotation efficiency and 4) the Pairwise Penalty Matrix (PPM) (Ash et al., 2020), which assesses pairwise performance differences between QMs across multiple annotation budgets, based on a t-test with a significance level of $\alpha = 0.05$. We argue that only a combination of these metrics provides a holistic assessment of AL by considering absolute performance (Final Dice, AUBC), relative performance (PPM), and annotation efficiency (FG-Eff). More details on our metrics and how we use them for our analysis are given in appendix D.

## 5 Empirical Study

### 5.1 Experimental Setup

**Datasets and Preprocessing** Our study spans four prominent medical imaging datasets: AMOS2022 (challenge task 2) (Ji et al., 2022), Medical Segmentation Decathlon – Hippocampus (Antonelli et al., 2022), KiTS2021 (Heller et al., 2023), and ACDC (Bernard et al., 2018). Each image is resampled to the median dataset spacing with a training & pool split (75%) and test split (25%) which is identical across all seeds and experiments. Then the nnU-Net "fingerprints" are created using the training & pool split, ensuring consistent input distributions across experiments. All following preprocessing steps were performed within the nnU-Net pipeline to maintain methodological consistency. More details are given in appendix E.

**Query Design and Annotation Budget** The selected query patch sizes for the datasets were selected taking into account the median image size and the size of the structures of interest for the corresponding datasets, leading to the following values: AMOS ($32 \times 74 \times 74$), KiTS ($64 \times 64 \times 64$), ACDC ($4 \times 40 \times 40$), and Hippocampus ($20 \times 20 \times 20$). We assess the performance of QMs under three Label Regimes (Low-, Medium-, and High-Label) each corresponding to 5 AL loops that simulate real-world annotation constraints on all datasets. The entire annotation budget for the Low-, Medium- and High-Label Regimes correspond to: 150, 300 and 450 patches for ACDC; 200, 1000, 2500 patches for AMOS; 200, 1000, 2500 patches for KiTS; 100, 200, 300 patches for Hippocampus. We use a starting budget and query size equal to 20% of the full annotation budget of each Label Regime. To ensure a representative starting budget, it is allocated to sample random foreground regions of each class, so that all classes are present in at least two patches. The rest of the starting budget is selected using the Random 33% FG strategy. More details on the dataset are given in appendix E.

### 5.2 Main Study

The results of the main study are visualized in a PPM in fig. 2 and two Win-/Lose-Barplots in fig. 3, all of which are aggregated over all experiments. Further, the ranking of all QMs with regard to AUBC, Final Dice, and FG-Eff for all Label Regimes of each dataset is shown in fig. 4 alongside a mean ranking. Detailed results are shown in appendix F. Using the aggregated results, we discuss the following five questions (Q1-Q5) with regard to the general performance of AL methods:

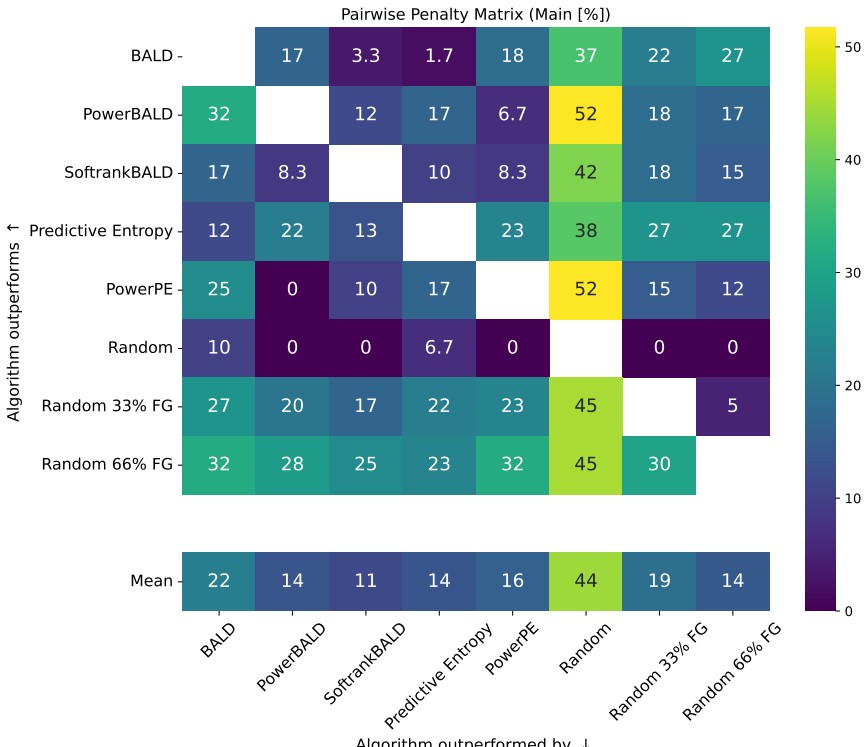

Figure 2: PPM aggregated over all experiments of the main study. At each position $(i, j)$ the values indicate the fraction of pairwise comparisons in % where method $i$ significantly outperformed method $j$.

**Q1: How do AL methods compare against Random?**  We observe that all AL methods consistently outperform Random with regard to performance metrics comparing patch budgets. This is indicated by first both the rankings of the AUBC and Final Dice in fig. 4 where it is consistently among the worst performing methods, especially for the Final Dice, and second all AL Strategies outperform it in over 37% of all evaluated budgets (fig. 3a).

However, we also observe that Random consistently draws the least amount of foreground voxels, indicated by its overall good ranking based on the FG-Eff metric, despite its bad ranking for Final Dice and AUBC (fig. 4). This good FG-Eff performance makes it unclear how much the annotation effort is actually reduced when employing AL methods over Random.

**Q2: How does AL compare against Foreground Aware Random?**  We observe that Foreground Aware Random strategies often outperform AL methods. Further, they outperform Random across all measured metrics with the exception of FG-Eff. Random 33% FG generally performs slightly worse than most AL methods in terms of both AUBC and Dice (fig. 4) as well as in the mean PPM (fig. 2). Meanwhile, Random 66% FG seems to be the best overall method based on the AUBC mean rank (fig. 4) and the positive Win-/Lose-Ratio against all AL methods except for Predictive Entropy (fig. 3b). Measured by the mean PPM Random 66% FG is tied with PowerBALD and Predictive Entropy in the second place (fig. 2). With regard to the Final Dice, it is, however, apart from Random, the worst performing method as AL methods become better for later annotation budgets (fig. 4).

In conclusion, Foreground Aware Random methods are a much harder baseline than purely Random, and most AL methods have issues outperforming them reliably. This behavior demonstrates that the amount of foreground selected is an important factor for the performance of a QM.

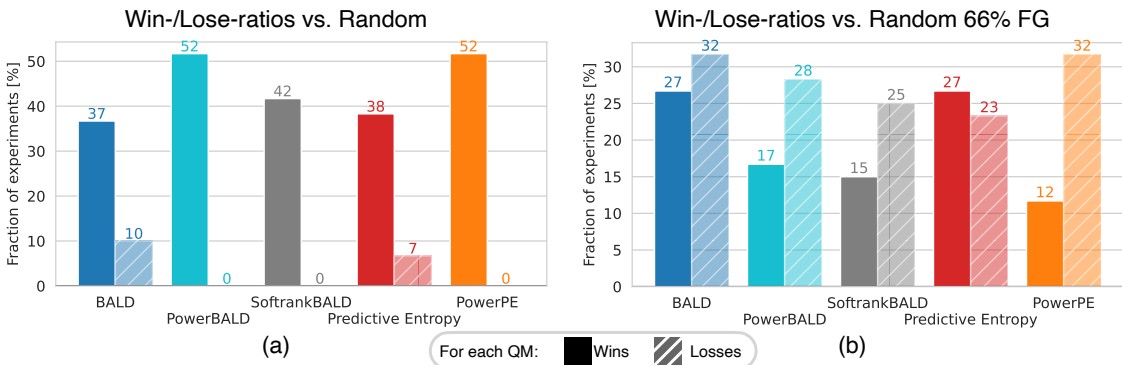

Figure 3: A detailed view into the Win-/Lose-ratios of AL methods in the PPM (fig. 2) for the main study against Random (a) and Random 66%FG (b). All AL methods outperform Random substantially more often than being outperformed with Noisy QMs, showcasing no Lose-scenarios (a). However, only Predictive Entropy outperforms Random 66% FG slightly more often than it is outperformed (b).

**Q3: Which AL method shows the best performance?** Predictive Entropy demonstrates strong overall performance across multiple evaluation metrics. It achieves the best mean rank in both the AUBC and Final Dice score of all AL methods (fig. 4) and is the only AL method with a positive win-loss ratio against Random 66% FG (fig. 3b). With regard to mean PPM (fig. 2) performance, it is tied with PowerBALD and Random 66% FG in the second place. Generally, performance gains of Predictive Entropy are observed especially in the later stages of AL experiments, which is showcased by its rank w.r.t. Final Dice generally being better than its rank w.r.t. AUBC (fig. 4). This behavior also leads to high variability in performance, particularly in low-label scenarios where selected queries are highly similar, and it negatively impacts its effectiveness, leading to it being outperformed by Random in some scenarios (fig. 3b). The queries of Predictive Entropy also commonly focus on foreground classes and thus query a lot of foreground, resulting in a relatively low FG-Eff compared to all other methods (fig. 4).

**Q4: How does the dataset influence AL performance gains?** We observe strong differences across our datasets with regard to AL performance gains. When comparing against Random 66% FG on Hippocampus and KiTS, AL is beneficial, whereas the trend is more neutral for ACDC. Contrastingly on AMOS all AL methods are generally outperformed by Random strategies, as can be seen for the AUBC and Final Dice ranks in fig. 4. We qualitatively discuss now these trends with the dataset-specific properties.

The primary challenge of **ACDC** lies in the anisotropic spacing and exact delineation of three spatially close cardiac structures. These structures are present in healthy and pathological conditions and most images are cropped to the chest area. The most challenging part is the exact delineation of structures, leading to Greedy QMs or Random 66% FG, which query large amounts of foreground, having good overall performance in AUBC and Final Dice.

For **AMOS**, the main challenge lies in correctly annotating 15 organs of varying sizes located in a large area. Here we observe that the models trained with queries from Random and all AL methods have issues with reliably capturing specific small organs, such as adrenal glands, sometimes leading to a Final Dice of 0. For Random this is due to the small probability of drawing patches that contain these organs. For the AL methods, this is likely due to the redundancy of queries focusing on specific classes. Consequently, this issue is less severe for Noisy QMs than for Greedy QMs. Random 66% and 33% FG do not exhibit this behavior (appendix F.1).

For the **Hippocampus** dataset, the main challenge lies in delineating the anterior from the posterior hippocampus, which is why query methods focusing on borders and uncertain regions greedily, such as BALD and Predictive Entropy, perform well. As the dataset is cropped to the brain region, the ratio of foreground to background is relatively high, leading to Random being more competitive than on the other datasets and

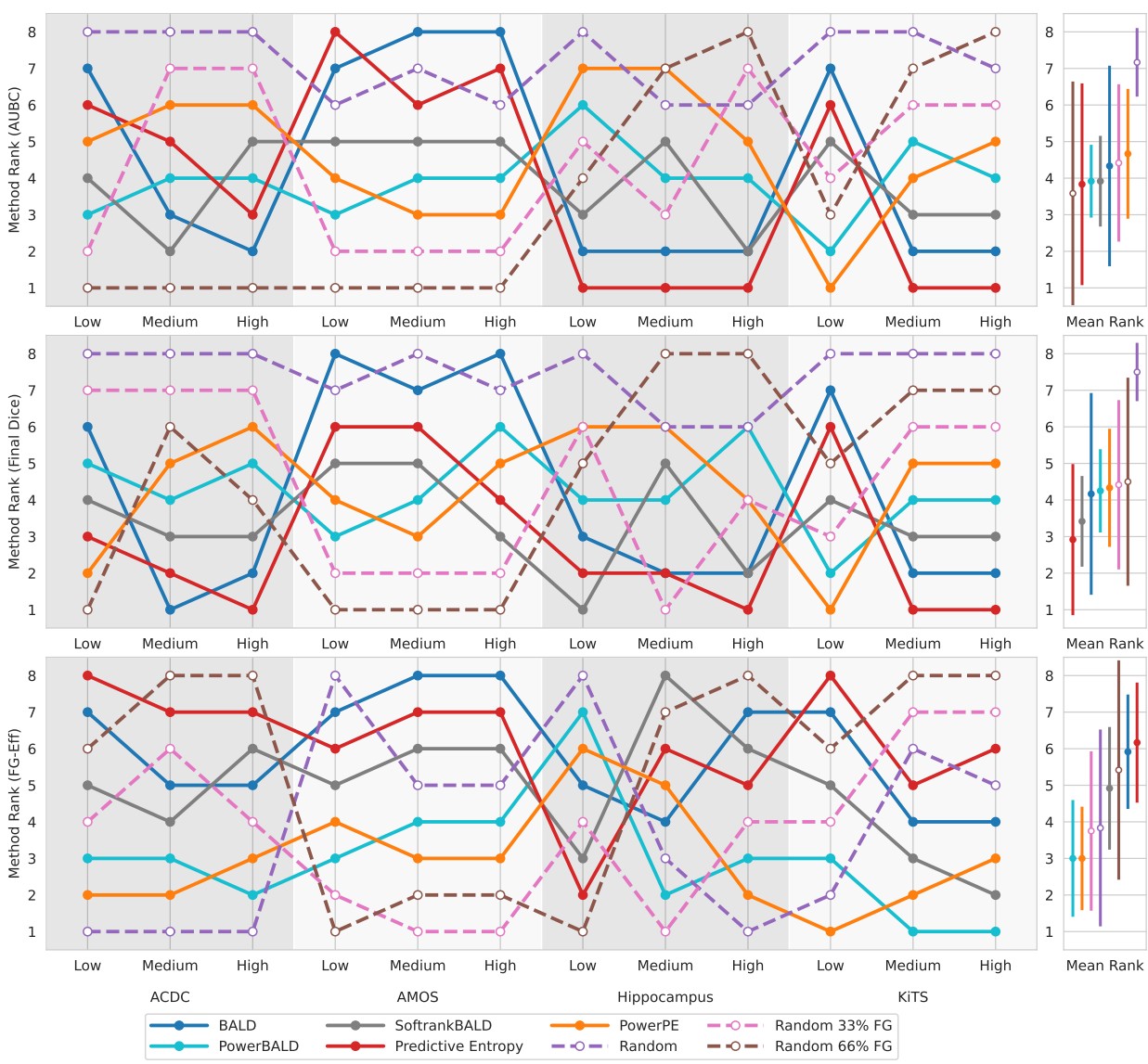

Figure 4: Ranking of methods according to AUBC, Final Dice and FG-Eff for each dataset and its Label Regimes (Low, Medium & High) alongside mean with standard deviations (bar).
The trend across datasets with regard to the benefit of AL differs over Foreground Aware Random strategies. On AMOS we observe no benefits when using AL across all Label Regimes whereas on KiTS and Hippocampus AL methods lead to performance improvements and a more neutral result for ACDC. Further, we observe a trend with regard to different Label Regimes where Noisy QMs outperform their Greedy counterparts (e.g. PowerBALD and BALD) on the Low-Label Regime.

Table 2: **Do smaller query sizes improve AL performance?** Kendall's $\tau$ measuring correlation between smaller query size and Final Dice. Large values indicate that smaller query sizes lead to performance improvements over larger query sizes. Dark colors indicate the significance of a two-sided test ($\alpha = 0.1$).

| Dataset | ACDC | | AMOS | | KiTS | |
| Label Regime | Low | High | Low | High | Low | High |
| Query Method | | | | | | |
| BALD | 0.640 | 0.284 | 0.107 | 0.711 | 0.426 | 0.178 |
| PowerBALD | 0.142 | 0.213 | 0.320 | -0.107 | -0.071 | 0.604 |
| SoftrankBALD | 0.533 | 0.426 | 0.391 | -0.036 | 0.142 | 0.249 |
| Predictive Entropy | 0.355 | 0.569 | 0.569 | 0.853 | 0.391 | 0.462 |
| PowerPE | 0.213 | 0.426 | 0.497 | 0.036 | 0.320 | -0.391 |

even outperforming Random FG 66% on the Medium- and High-Label Regime for AUBC and Final Dice (fig. 4). Overall, the performance of models is close to the performance on the entire dataset (table 6).

For **KiTS**, the kidney structure and tumors are clustered together with the scan covering large surrounding areas and also large areas containing only air. Generally, foreground-aware strategies have many false positives in their scans due to the queries covering mostly foreground but not all derivations of background (appendix F.2). This behavior is not observed for the AL methods due to them querying exactly these background areas, leading to the rankings generally favoring AL methods for AUBC, Final Dice, and also FG-Eff (fig. 4). Due to the overall diversity of different structures, a tendency for redundant queries by Greedy QMs can be observed, leading to Predictive Entropy and BALD being outperformed on the Low-Label Regime.

**Q5: What is the influence of the annotation budget on AL Performance?** Generally, we observe that low annotation budgets are the most challenging setting for AL methods due to the potential redundancy of queries, which is especially strong for the Greedy Query Methods BALD and Predictive Entropy. Consequently, these are among the worst-performing methods on the Low-Label Regime on ACDC, AMOS, and KiTS, especially when considering AUBC (fig. 4). As Noisy QMs query more diversified, they are more robust, leading to them never being outperformed by Random (fig. 3a). However, on larger annotation budgets and later stages, the Noisy QMs do not perform as well as their Greedy counterparts. For example, on ACDC, the difference in AUBC rankings between the Low- and High-Label Regime indicates that later stages are not as affected by the redundancy of queries (fig. 4).

Our findings suggest that in early loops of AL and especially for low annotation budgets Noisy QMs are more reliable than Greedy QMs.

### 5.3 Ablations

We give a short description of the experiment setup and a summary of our main findings and their analysis for each of our four ablation studies. Detailed information and analysis for each of the four ablations are given in appendix G.

**Query Size** To evaluate the influence of the query size we conduct experiments on the ACDC, AMOS and KiTS datasets on the Low- and High-Label Regimes using three different settings of the query size based on the main study which is halved ($\text{QS} \times \frac{1}{2}$), identical ($\text{QS} \times 1$) and doubled ($\text{QS} \times 2$).

We observe the trend that smaller query sizes with more AL loops lead to performance improvements over larger query sizes with fewer AL loops when measured with Kendall's $\tau$ (Kendall, 1948) (example in table 2), which is more pronounced for Greedy QMs than for noisy QMs. There is no setting in which a smaller query size leads to a significant performance decrease. Notably, these performance improvements can alter the method ranking substantially toward favoring AL QMs over Random strategies, especially from the ranking of $\text{QS} \times 2$ to $\text{QS} \times \frac{1}{2}$ and $\text{QS} \times 1$ when measured with Kendall's $\tau$. While this indicates that smaller query sizes

Table 3: **Does longer training improve AL performance?** $\Delta$Final Dice = (Final Dice(500 Epochs) − Final Dice(Precomputed)) × 100. Positive values indicate that longer training leads to better queries even when accounting for performance differences stemming from longer training. Dark colors indicate the significance of a two-sided t-test ($\alpha = 0.1$).

| Dataset | AMOS | | KiTS | |
|---|---|---|---|---|
| Label Regime | Medium | High | Medium | High |
| Query Method | | | | |
| BALD | 0.98 | 0.87 | 1.73 | 1.64 |
| PowerBALD | 0.94 | 0.52 | 2.30 | 1.38 |
| SoftrankBALD | 1.00 | 0.32 | 1.41 | 1.04 |
| Predictive Entropy | 0.15 | 0.70 | 0.58 | 0.88 |
| PowerPE | 0.16 | 0.46 | 1.89 | 1.71 |

are preferable in practice, we want to highlight that this comes at the cost of increased computational cost, which scales inversely proportional to the query size. For the detailed analysis, we refer to appendix G.1.

**Training Length** We evaluate the influence of the training length with three different settings: training the model for 500 epochs (500 Epochs), training the model for 200 epochs as in our main study (200 Epochs), and training the models for 500 epochs on the entire query trajectories from the models trained with 200 epochs (Precomputed). The experiments are performed on AMOS and KiTS Medium- and High-Label Regimes, as on these datasets, longer training leads to substantial performance differences when trained on the entire dataset.

We find that longer training leads to significantly better queries resulting in performance gains for AL methods, even when taking into account that longer trained models generally yield higher Dice scores (i.e., comparing the Precomputed and 500 Epochs settings), as shown in table 3. Measuring the robustness of rankings with Kendall's $\tau$, we find the ranking of methods stays similar from shorter to longer-trained models when AL methods already perform better than Random strategies (KiTS), whereas in settings where AL methods do not outperform Random strategies (AMOS), there is a general shift in ranking towards favoring AL methods over Random strategies. This shift is stronger for the 500 Epoch setting than for the Precomputed setting, underlining the improved quality of queries for longer training. For the detailed analysis, we refer to appendix G.2.

**Noise strength in Noisy QMs** Through an exemplary but systematic ablation of PowerBALD, we aim to understand the influence of the noise strength for the Noisy QMs. We assess the influence of the noise by performing experiments on the ACDC, AMOS, and KiTS datasets for the Low-, Medium- and High-Label Regime. In these experiments, we reduce the noise over 6 steps (controlled with the parameter $\beta$) from the noise level used in our main study from PowerBALD level ($\beta = 1$) to BALD level ($\beta = \infty$) without noise.

We observe as a general trend that for the smaller annotation budgets, the best performance (in terms of AUBC and Final Dice) is obtained through stronger noise levels ($\beta = 1$), while for larger annotation budgets, less noise is beneficial. For FG-Eff, we observe a decreasing trend as we decrease the noise strength (fig. 5 for exemplary results on KiTS). Both observations show that the hyperparameter of QMs can have a substantial impact on the performance. However, the optimal noise values vary greatly across datasets and are dependent upon a variety of different factors, s.a. query size, training length, data redundancy, query patch size and annotation budget. We believe more research is necessary to optimize it on yet unseen datasets. For the detailed analysis, we refer to appendix G.3.

**Query Patch Size** Unlike previous work that restricts queries to entire 3D volumes or 2D slices, our setup allows free 3D patch selection, introducing an additional hyperparameter, the query patch size. To systematically assess its effect, we repeat our entire primary benchmark across four datasets, halving the query patch size along each axis while maintaining the same number of queried patches per label regime. This setup enables a fine-grained selection of annotation regions.

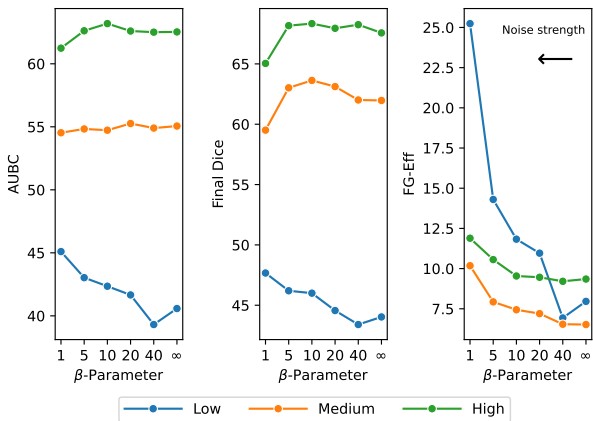

Figure 5: **How does the noise strength influence PowerBALD?** AUBC, Final Dice and FG-Eff for different $\beta$ parameters of PowerBALD on the KiTS dataset across Low-, Medium- and High-Label Regime. Higher $\beta$ leads to a reduced perturbation of the rankings.

Contrary to our expectations, we observe no significant drop in general AL method performance compared to Random strategies from Patch$\times1$ to Patch$\times\frac{1}{2}$, with the relative ranking according to AUBC even improving, despite an 8-fold reduction in absolute annotated voxels. Similarly, comparing the PPMs of Patch$\times1$ (fig. 2) and Patch$\times\frac{1}{2}$ (fig. 6), more AL methods have a better mean rank than the Random strategies for Patch$\times\frac{1}{2}$. Based on our examination of the ranking stability of the AUBC and Final Dice with Kendall's $\tau$, we find that depending on the dataset, rankings remain stable on KiTS and AMOS, whereas on Hippocampus and ACDC they are unstable. We observe a general trend that Noisy QMs perform better than Greedy QMs for the smaller patch size, indicated by PowerPE being the best method for the smaller patch size compared to Predictive Entropy for the larger patch size. In conclusion, while the relative performance of AL methods compared to Random strategies remains remarkably resilient w.r.t. changes in query size, their relative ranking is susceptible to change. To ensure optimal method selection, a systematic evaluation of AL strategies under varying patch sizes is necessary. For the detailed analysis, we refer to appendix G.4.

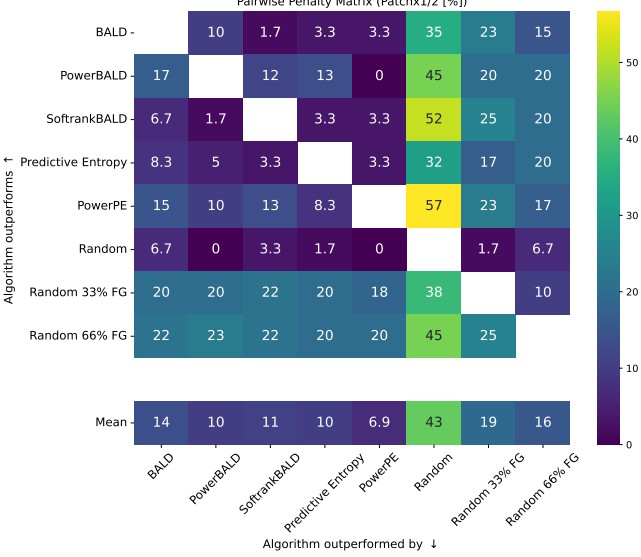

Figure 6: PPM for the Patch$\times\frac{1}{2}$ configuration aggregated over all settings. Mean row results change compared to the Patch$\times1$ (fig. 2).

# 6 Discussion & Conclusion

We propose the nnActive framework for semantic segmentation in 3D biomedical imaging, an AL extension of nnU-Net, which allows for measuring performance estimates of AL methods that are generalizing and practically relevant, which is crucial for real-world application. In addition, we conduct the largest to date empirical AL study in the 3D biomedical imaging domain, from which we obtain the following findings with regard to uncertainty-based AL methods:

- **AL vs. Random:** All evaluated AL methods lead to substantial performance improvements over pure Random sampling, but select substantially more foreground, which likely leads to a higher annotation effort per query [section 5.2 Q1].

- **A new Baseline:** Foreground Aware Random sampling is a trivial yet hard to beat baseline. No AL method appears to outperform it reliably [section 5.2 Q2].

- **Best AL Method:** Predictive Entropy is overall the best-performing AL method measured by AUBC, Final Dice and PPM, but its performance is highly variable, e.g., for small annotation budgets, and it has the worst overall FG-Eff, which indicates a high annotation effort per query [section 5.2 Q3].

- **AL generalization:** AL performance gains strongly depend on dataset and task properties like the ratio of foreground to background and the number of structures to segment [section 5.2 Q4].

- **Noisy QMs:** Noisy Query Methods like PowerPE are more reliable in earlier stages of AL and lead to better FG-Eff than Greedy Methods like Predictive Entropy [section 5.2 Q5].

- **Improving AL performance:** AL method performance can be substantially increased with more compute-intensive settings like longer training and smaller query sizes [section 5.3, Query Size & Training Length].

- **AL hyperparameters:** AL method hyperparameters, such as the noise strength, lead to substantial performance differences, but optimal hyperparameters differ between datasets and annotation budgets [section 5.3, Noise strength in Noisy QMs].

**Guidelines** Based on the findings listed above we provide the following guidelines for AL on 3D biomedical images: **For Practitioners:** 1) Based on the strong performance of Foreground Aware Random strategies, we agree with Burmeister et al. (2022) that in many practical scenarios, improved Random strategies, that do not require iterative re-training, may be sufficient. 2) When employing AL, longer training and smaller query sizes represent ways to substantially reduce annotation effort at the cost of more compute. **For developers:** 1) Improvements over the naive Random baselines are not sufficient to give a recommendation for widespread use of AL. 2) Method evaluation can be performed using shorter trainings, as performance improvements through longer trainings are consistent across AL methods.

**Relevance of our Framework & Benchmark** We believe that the nnActive framework, in combination with our study, will serve as a catalyst for future method development by providing a reliable and unifying benchmark. This will lead to wide-spread adoption to the best practices laid out in sections 2 to 3 by overcoming key barriers w.r.t. their adoption which are the high implementation and computational costs required for integrating AL methods into state-of-the-art frameworks due to their complexity and evaluation of multiple AL methods and baselines.

**Limitations** Due to the depth and rigor of our evaluation, combined with several orthogonal improvements to the AL experiment design s.a. partial loss and queries in form of freely adaptable 3D patches, we focus our evaluation on uncertainty-based AL methods, which are widely used and generally among the best-performing AL methods for 3D biomedical segmentation (Föllmer et al., 2024) whilst not requiring changes in model architecture and training. We therefore did not evaluate methods like Learning Loss Active Learning (Yoo & Kweon, 2019), changing the training and diversity-based methods like Core-Set (Sener & Savarese, 2018). However, our selection of AL methods is still a comprehensive set that we believe to be representative of the current state-of-the-art for 3D biomedical AL.

**Future directions**   Directly building on top of our nnActive framework and study, the following directions are promising: 1) Scaling of diversity-based AL methods like Vepa et al. (2024) and Föllmer et al. (2024) to our performance optimized setting with 3D models and ensembles, as they are, as of now, not represented in our benchmark. 2) Incorporation of Foundation Models for 3D biomedical imaging into our benchmark using nnU-Net due to the decreased time necessary for finetuning and better performance on low annotation budgets. 3) Extension of our proposed FG-Eff metric to a measure which *more accurately* measures annotation effort than number of foreground voxels, e.g. number of clicks for regions (Mackowiak et al., 2018). 4) Incorporation and benchmarking of methods for starting budget selection, as a well-selected starting budget can increase AL performance (Gupte et al., 2024).

## Acknowledgements

This work was funded by Helmholtz Imaging (HI), a platform of the Helmholtz Incubator on Information and Data Science. This work is supported by the Helmholtz Association Initiative and Networking Fund under the Helmholtz AI platform grant (ALEGRA (ZT-I-PF-5-121)).

The authors gratefully acknowledge the computing time provided on the high-performance computer HoreKa by the National High-Performance Computing Center at KIT (NHR@KIT). This center is jointly supported by the Federal Ministry of Education and Research and the Ministry of Science, Research and the Arts of Baden-Württemberg, as part of the National High-Performance Computing (NHR) joint funding program (https://www.nhr-verein.de/en/our-partners). HoreKa is partly funded by the German Research Foundation (DFG).

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

## Author Contributions

### Exact Details of Contributions

**Core Contributors:** Carsten T. Lüth, Jeremias Traub

**Writing**
First Draft: Carsten T. Lüth
Revising Text: Carsten T. Lüth, Jeremias Traub
Reviewing Text: Carsten T. Lüth, Jeremias Traub, Kim-Celine Kahl, Lars Krämer, Lukas Klein, Paul F. Jaeger, Fabian Isensee
Figure Creation: Lukas Klein & Carsten T. Lüth
Result Visualization: Carsten T. Lüth & Jeremias Traub

**Experiments & Framework**:
Code Contributors: Carsten T. Lüth, Kim-Celine Kahl, Till Bungert, Fabian Isensee, Jeremias Traub
Experiment Design and Concepts: Carsten T. Lüth, Fabian Isensee, Paul F. Jaeger, Jeremias Traub
Analysis Framework: Carsten T. Lüth
Running Experiments: Jeremias Traub, Carsten T. Lüth
Experiment Handling: Jeremias Traub, Carsten T. Lüth
Releasing of the Framework: Jeremias Traub

**Supervision**: Klaus Maier-Hein, Fabian Isensee, Paul F. Jaeger

**Leads**: Carsten T. Lüth

### Historical Description

This work was performed over three years with multiple people contributing a lot to the project, making the declaration of exact author contributions challenging.

In general, over the entire span of the time, Carsten T. Lüth acted as lead for the project with numerous and large contributions, especially from Kim-Celine Kahl, Till Bungert, Paul F. Jaeger, and Fabian Isensee. In the last year of the project, it became apparent that the workload simply was too large for one person to handle; therefore, Jeremias Traub joined the project full-time, at first just to support Carsten T. Lüth. His dedicated work, however, was much more than just pure support and his analytical thinking and flow of ideas led to many overall improvements of the work, increasing it greatly in quality. Therefore, Carsten and Jeremias agreed to share the first authorship.

As Paul F. Jaeger left the project half a year before finalization and was therefore not present during the writing phase, he offered to concede his last author position.

# Appendix

## Table of Contents

# A    Related Works

**Pitfalls**

We present a detailed comparison of related works and their evaluation protocols in table 4.

The scoring rules for our pitfalls criteria in table 1 used for ✔, (✔) and a ✗ if not addressed:

**P1** A. Evaluate performance on at least 3 datasets (only counting 3D biomedical). (✔)
B. Evaluate at least two different starting budgets and query sizes. (✔)
If A. & B. ✔
**P2** Use 3D models with training optimized for partial annotations. ✔
2D models: pretrained, Semi-Supervised Training or partial annotations. (✔)
**P3** Evaluate Random Baselines that take into account that for 3D Biomedical image datasets large areas of the images are pure background and/or make use of the 3D structure of the data. ✔
**P4** Use metrics that take into account that the effort to annotate background is very low compared to foreground. ✔

**Compared Literature**

**Nath et al. (2021)**   Contribution: Propose to query samples from dataset pool without removing them and enforce diversity with Mutual Information over histogramms.
Query Methods: BALD, BALD with Mutual Information on Histogramms, Random.
Datasets: MSD Hippocampus and Pancreas
Evaluation Metric: Best Mean Dice (3D) over Experiment
Evaluated Query Sizes per dataset (max): 1
Evaluated Starting Budgets per dataset(max): 1

**P1** 3 biomedical datasets, no ablations for multiple annotation budgets.
**P2** Do use 3D models but no partial annotations, also no pretrained models or 3D models in combination with partial annotations and also no Data Augmentations.
**P3** No improved random baseline.
**P4** No Measurement taking into account the annotation effort.

**Burmeister et al. (2022)**   Contribution: Evaluate Strided and Stratified Sampling Strategies.
Query Methods: Least Confidence, Entropy, Distance-based representativeness sampling, Cluster-based representativeness sampling, Strided Random Sampling and Stratified Random Sampling. Additional experiments with label interpolation.
Datasets: MSD Hippocampus, Prostate and Heart
Evaluation Metric: Mean Dice (3D) Plots.
Evaluated Query Sizes per dataset (max): 1
Evaluated Starting Budgets per dataset(max): 1

**P1** 3 biomedical datasets, not multiple annotation budgets per dataset.
**P2** Does neither use pretrained models or 3D models in combination with partial annotations.
**P3** Do use Strided and Stratified Random Sampling.
**P4** No Measurement taking into account the annotation effort.

**Gaillochet et al. (2023a)**   Contribution: Propose Stochastic Batches as Query Methods.
Query Methods: Stochastic Batches, Entropy, BALD, Test-Time Augmentations, Learning Loss, Core-Set, Random.
Datasets: Prostate MR Image Segmentation (PROMISE) challenge 2012, MSD Hippocampus
Evaluation Metric: Mean Dice (3D) and Hausdorff Distance.
Evaluated Query Sizes per dataset (max): 3 (ablation one dataset)
Evaluated Starting Budgets per dataset(max): 3 (ablation one dataset)

**P1** 2 biomedical datasets, not multiple annotation budgets per dataset; however, multiple annotation budget ablations for one dataset.

**P2** Does neither use pretrained models or 3D models in combination with partial annotations.

**P3** No Improved Random Baselines.

**P4** No Measurement taking into account the annotation effort.

**Gaillochet et al. (2023b)**   Contribution: Propose Test-Time Augmentations as Query Method.
Query Methods: Entropy, BALD, Test-Time Augmentations, Core-Set, Random.
Datasets: ACDC
Evaluation Metric: Mean Dice (2D and 3D).
Evaluated Query Sizes per dataset (max): 1
Evaluated Starting Budgets per dataset(max): 1

**P1** 1 biomedical datasets, not multiple annotation budgets per dataset, however, multiple annotation budget ablations for one dataset.

**P2** Use 2D Semi-Supervised models.

**P3** No Improved Random Baselines.

**P4** No Measurement taking into account the annotation effort.

**Ma et al. (2024)**   Contribution: Add target & boundary awareness to existing Query Methods.
Query Methods: Entropy (with and without Dropout), BALD, Margin Sampling, Least Confidence.
Datasets: MSD Spleen, BraTS
Evaluation Metric: Mean Dice (2D and 3D) – % required data to achieve fully annotated performance and peak performance.
Evaluated Query Sizes per dataset (max): 1
Evaluated Starting Budgets per dataset(max): 1

**P1** 1 biomedical datasets, not multiple annotation budgets per dataset, however, multiple annotation budget ablations for one dataset.

**P2** Does neither use pretrained models or 3D models in combination with partial annotations.

**P3** No Improved Random Baselines.

**P4** No Measurement taking into account the annotation effort.

**Föllmer et al. (2024)**   Contribution: Propose Uncertainty-Aware Subomdular Information Measure (USIM) as Query Method.
Query Methods: USIMF, USIMC, Mean STD, Core-Set, BADGE (LL), Stochastic Batches, Entropy, BALD, Random.
Datasets: MSD Spleen, Liver and Hippocampus
Evaluation Metric: Mean Dice (3D) – Pairwise Penalty Matrix.
Evaluated Query Sizes per dataset (max): 1
Evaluated Starting Budgets per dataset(max): 1

**P1** 3 biomedical datasets, not multiple annotation budgets per dataset, however, multiple annotation budget ablations for one dataset.

**P2** Use 2D Semi-Supervised models.

**P3** No Improved Random Baselines.

**P4** No Measurement taking into account the annotation effort.

**Vepa et al. (2024)**   Contribution: Propose Metric Learning Based Query Method building upon Core-Set (Core-Metric).
Query Methods: Core-Metric, Core-Set, Random, CoreGCN, TypiClust, Stochastic Batches, VAAL, Variance Ratio, BALD.
Datasets: ACDC, CHAOS (Combined Healthy Abdominal Organ Segmentation), MS-CMR (Multi-sequence Cardiac MR Segmentation Challenge) and DAVIS (Densely Annotated Video Segmentation)[2]

---

[2]Not included in dataset count as it is a non-medical non-3D dataset

Evaluation Metric: Mean Dice (3D) – Pairwise Penalty Matrix.
Evaluated Query Sizes per dataset (max): 2 (1 Pretrained and 1 Trained from Scratch)
Evaluated Starting Budgets per dataset(max): 2 (1 Pretrained and 1 Trained from Scratch)

**P1** 3 biomedical datasets and multiple annotation budget ablations for one dataset.
**P2** Use 2D models, both pretrained and trained from random initialization.
**P3** No Improved Random Baselines.
**P4** No Measurement taking into account the annotation effort.

**Shi et al. (2024)**   Contribution: Propose Predictive Accuracy-based Active Learning (PAAL).
Query Methods: Random, Entropy, Variation Ratio, Margin, KMeans, CoreSet, Entropy+KMeans, AB-UNet, CEAL, LPL, PAAL
Datasets: ACDC, SegThor, MSD Brain, Liver OAR (in-house dataset)
Peculiarity: Use 1/5th of the data as a validation set used during training to determine whether the query step will be performed.
Evaluation Metric: Mean Dice (not specified whether 2D or 3D in paper and not clearly described in code)
Evaluated Query Sizes per dataset (max): 3
Evaluated Starting Budgets per dataset (max): 3

**P1** 4 biomedical datasets, no multiple annotation budgets per dataset.
**P2** Does neither use pretrained models or 3D models in combination with partial annotations.
**P3** No Improved Random Baselines.
**P4** Show the number of slices for all classes for different QMs.

**Ours**   Query Methods: BALD, Entropy, PowerBALD, SoftrankBALD, PowerPE, Random, Random 66%FG, Random 33%FG. Datasets: ACDC, AMOS, Hippocampus, KiTS Evaluation Metrics: Mean Dice (3D) – Pairwise Penalty Matrix, Area Under Budget Curve, Final Mean Dice, Foreground Efficiency.
Evaluated Query Sizes per dataset (max): 3
Evaluated Starting Budgets per dataset (max): 3

**P1** 4 biomedical datasets with experiments on three different label regimes each with one query size and starting budget.
**P2** Using 3D models that are trained with a partial loss on the annotated regions.
**P3** Random 33%FG and Random 66% FG alleviate background selection issue of Random.
**P4** We propose the dedicated measure named *Foreground Efficiency* (FG-Eff) (see appendix D.4 for details).

Table 4: Comparison of works in the field of Active Learning for 3D biomedical imaging.
**Notation**: #Datasets[‡]: Only counting 3D biomedical datasets; no[†]: not specified in paper and not found in code; **N.S.**: not specified in paper and code.

| Name | Seg. Model | Full Retraining | Query Design | #Datasets[‡] | Novel QM | Ensemble | Seeds | Advanced Random | Training Strategies | Data Augmentations |
|---|---|---|---|---|---|---|---|---|---|---|
| Föllmer et al. (2024) | 2D nnU-Net (v.1) | no | 2D Slice | 3 | yes | no | 2 | no | Standard Training | yes |
| Ma et al. (2024) | 2D U-Net | no | 2D Slice | 2 | yes | no | N.S. | no | Standard Training | yes |
| Burmeister et al. (2022) | 2D U-Net | no | 2D Slice | 3 | no | no | 3 | yes | Standard Training | no[†] |
| Gaillochet et al. (2023b) | 2D U-Net | yes | 2D Slice | 1 | yes | no | 5 | no | Semi-SL | yes |
| Gaillochet et al. (2023a) | 2D U-Net | yes | 2D Slice | 2 | yes | no | 5 | no | Standard Training | yes |
| Nath et al. (2021) | 3D U-Net | yes | 3D Image | 2 | yes | yes | 5 | no | Standard Training | no |
| Vepa et al. (2024) | 2D U-Net | yes | 2D Slice | 3 | yes | no | 5 | no | Pre-Trained& 2D Partial Loss | yes |
| Shi et al. (2024) | 2D U-Net | no | 2D Slice | 4 | yes | no | 5 | no | Standard Training | yes |
| Ours | 3D nnU-Net (v.2) | yes | 3D Patch | 4 | no | yes | 4 | yes | Partial Loss | yes |

# B   Task Description

In Active Learning (AL) for 3D biomedical image segmentation, acquiring full annotations for an entire volumetric scan is often infeasible due to the extensive time required. Instead, partial annotations allow for selective labeling of subregions within a 3D image, reducing annotation effort while still guiding model learning effectively. This section formalizes the task of querying and incorporating partial annotations in a 3D AL framework.

**Mathematical Formulation**   Let $\mathcal{X}$ denote the space of 3D volumetric images, where each sample is a 3D image $X \in \mathbb{R}^{M \times H \times W \times D}$, with number of modalities $M$, height $H$, width $W$, and depth $D$. The corresponding dense ground-truth segmentation is given by $Y \in \{0, 1, \ldots, C\}^{H \times W \times D}$, where $C$ is the number of classes.

In a standard supervised learning setting, a model $f_\theta$ is trained using full annotations $(X, Y)$ from a dataset $\mathcal{D} = \{(X^{(i)}, Y^{(i)})\}_{i=1}^{N}$. However, in AL with partial annotations, we define a Query Method that can select multiple subsets of the volume of a single image $Q(X)$ spread over the entire dataset. For a single image, the annotated subset is denoted as:

$$\tilde{Y} = Q(X), \quad \tilde{Y} \subseteq Y$$

where $\tilde{Y}$ represents the annotated queries where only a fraction of the full annotation is provided.

The unobserved regions remain unannotated and are ignored or used for weakly supervised training.

In this work, we focus on 3D patches for partial annotation. Thus, a partial annotation for one image is defined as $\tilde{Y} = \{Y_{h:h_p, w:w_p, d:d_p} \mid (h, w, d) \in \mathcal{S}_P\}$, with $(h_p, w_p, d_p)$ denoting the size of the 3D patch and $\mathcal{S}_P$ the set of patch locations. [3]

Given a dataset $\mathcal{D} = \{(X^{(i)}, \tilde{Y}^{(i)})\}_{i=1}^{N}$, where only $\tilde{Y}^{(i)}$ is available for training, the loss function is adapted to account for missing labels:

$$\mathcal{L}(\theta) = \sum_{i=1}^{N} \sum_{j \in \mathcal{S}^{(i)}} \ell(f_\theta(X_j^{(i)}), \tilde{Y}_j^{(i)})$$

where $\mathcal{S}^{(i)}$ denotes the queried (labeled) locations in image $i$.

---

[3]2D Slices represent a subset of 3D patches, defined by e.g. $h_p = H, w_p = W, d_p = 1$.

## C  Active Learning Framework

---
**Algorithm 1** Active Learning Patch Selection

---
**Input:**
Set of images $\{X^{(i)}\}_{i=1}^N$, query size $n$, labeled set $\mathcal{L}$, Uncertainty function $U$, Aggregation function $A$, $o$ allowed overlap **Output:** Final query set $\mathcal{Q}$

  1: Initialize final query set $\mathcal{Q} \leftarrow \emptyset$
  2: **for** each image $X^{(i)} \in \{X^{(i)}\}_{i=1}^N$ **do**
  3:      $\mathcal{U} \leftarrow U(X^{(i)}, \mathcal{M})$ # compute uncertainty for image
  4:      $\mathcal{U}_{\mathrm{Agg}} \leftarrow A(\mathcal{U})$ # aggregate uncertainties to patch-level
  5:      $\mathcal{Q}_{\mathrm{Image}} \leftarrow \emptyset$ # initialize best patches for current image
  6:      **for** $q$ in sort($\mathcal{U}_{\mathrm{Agg}}$)[::-1] **do** # sort in descending order according to uncertainty
  7:          **if** overlap($q, \mathcal{Q}_{\mathrm{Image}} \cup \mathcal{L}$) $\leq o$ **then** # ensure that
  8:              $\mathcal{Q}_{\mathrm{Image}} \leftarrow \mathcal{Q}_{\mathrm{Image}} \cup \{q\}$
  9:          **end if**
10:      **end for**
11:      $\mathcal{Q} \leftarrow \mathcal{Q} \cup \mathcal{Q}_{\mathrm{Image}}$
12: **end for**
13: $\mathcal{Q} \leftarrow$ sort($\mathcal{Q}$)[::-1] # sort in descending according to uncertainty
14: **Return** $\mathcal{Q}$

---

To ensure that nnActive can be used both for benchmarking and in production, we perform all perturbations of the images inside of the nnU-Net dataset structure. More specifically, inside the *nnUNet_raw* folder where we also store *loop_XXX.json* files, which store all relevant information of the queried patches. This allows to change the labels of all images directly in-place. Changes in the *nnUnet_raw* folder are transferred to the preprocessed dataset used for training using the standard *nnUNet_preprocessing* step.

For the query stage we build it on the patchwise inference of nnU-Net in a final stage after each image is predicted for all ensemble members. The algorithm used in our framework for a top-k uncertainty method (e.g., BALD or Predictive Entropy) is outlined in algorithm 1.

To enrich the spatial context available to the model, we enhanced the standard patch-based nnU-Net trainer through region sampling. Specifically, the final patch used for a forward pass still contains at least one labeled voxel (based on random or class-specific sampling), but the patch is not centered on the annotated voxel (as for the standard nnU-Net trainer). Instead, the annotated voxel is randomly located within the final patch, following a uniform distribution over the valid patch region. Since not all voxels in the input patch are necessarily annotated, nnActive supports training with partial losses, applying the loss only where labels are available. Importantly, the patch size used during the model's forward pass is always determined by the nnU-Net plans and configurations, which is fixed for each dataset. The query patch size used in the nnActive experiment configuration is not necessarily identical to the nnU-Net patch size.

# D   Evaluation Metrics

In our evaluation, we performed an analysis based on all of the metrics described in this section.

In the analysis of our main study, we focused on all metrics, whereas in our ablations, we put special emphasis on the AUBC and Final Dice as they allow easier direct comparisons of values. This is also visualized in the overview figure fig. 8.

Our newly proposed metric, FG-Eff, measuring the annotation efficiency by proxy of foreground voxels, is described in appendix D.4.

## D.1   Final Dice

We use the Final Dice value after the annotation budget is exhausted for evaluation, as it allows for easy interpretation and puts a special emphasis on later stages of AL experiments.

## D.2   AUBC

We compute the Area Under the Budget Curve (AUBC) for each dataset and Label Regime based on the Mean Dice to allow assessing the absolute performance each QM brings (see (Zhan et al., 2021; 2022) for more details). It aggregates the results of one Label Regime using the trapezoid method, and higher values indicate better performance under all budgets of the label regime.

Our normalization of the AUBC is set so that if all values on one Label Regime are equal to 0.8, the AUBC will return 0.8.

## D.3   Pairwise Penalty Matrix

We employ the Pairwise Penalty Matrix (PPM) to assess whether one QM significantly outperforms others in terms of Mean Dice. This metric reflects how frequently a method yields statistically superior performance compared to another, based on a two-sided t-test with a significance level of $\alpha = 0.05$ (see (Ash et al., 2020) for further details) and whether the mean performance of method i is larger than that of method j and vice-versa. The PPM enables aggregation across multiple datasets and label regimes, though it does not account for absolute performance differences.

In the final matrix, we show values in % where each row i represents the fraction of settings where method i significantly outperforms other methods, whereas each column j shows the fraction of settings where another significantly outperforms method j.

## D.4   Foreground Efficiency

**Overview**   We measure the annotation efficiency by proxy of the amount of foreground annotation using the decay parameter $\gamma$, we term Foreground Efficiency (FG-Eff) for an exponential decay fitted to the performance gap to a model trained on the entire dataset and the number of foreground voxels. It allows for a simpler interpretation of plots like the following: As the number of foreground voxels represents a proxy for annotation effort, the FG-Eff does not replace other performance metrics s.a. AUBC, Pairwise Pen but should be seen as an extension of them.

**Mathematical Definition**   The formula for the fitted exponential decay is given in eq. (1), where values with a ^ are estimated empirically based on the data prior to the fit of $\gamma$ and $t$ is the mean % of annotated foreground voxels (therefore $t \in [0, 1]$) and $\hat{t}_0$ is it on the starting budget while $y$ is the performance (Mean Dice). $y_{\text{full}}$ is the performance on the entire dataset using a trainer with identical length trained on the entire dataset and $\hat{y}(\hat{t}_0)$ is the mean performance on the starting budgets.

$$y(t) = (\hat{y}(\hat{t}_0) - \hat{y}_{\text{full}}) \exp(-\gamma(t - \hat{t}_0)) + \hat{y}_{\text{full}} \tag{1}$$

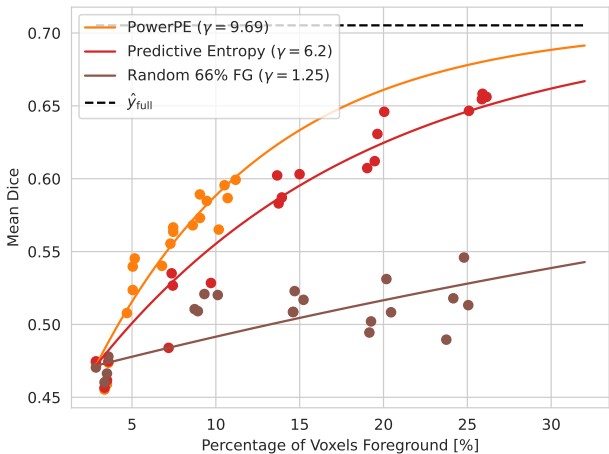

Figure 7: Visualization of a fit for the FG-Eff on the KiTS Medium-Label Regime showing the QMs: Predictive Entropy, PowerPE and Random 66% FG. The points show the actual performance of all 4 seeds. The $\gamma$ (FG-Eff) values allow to capture that PowerPE requires much less foreground to achieve a similar performance than Predictive Entropy and also merges the information that even though Random 66% FG and that while Predictive Entropy queries a similar amount of foreground as Random 66% FG, the latter is much less performant.

Fit values: $\hat{t}_0 = 0.028$, $\hat{y}_{\text{full}} = 0.705$, $\hat{y}(\hat{t}_0) = 0.472$

**Mathematical Assumptions**

- The behavior can be modelled with an exponential decay.

- $y(t) < \hat{y}_{\text{full}} \forall t \in [t_0, t_{\max}]$. Caveat $y(1) = \hat{y}_{\text{full}}$

**Interpretation**   Higher values indicate that a QM is more annotation efficient as it converges faster to the performance obtained when training on the entire dataset. As the number of foreground voxels is a proxy for annotation effort, we also emphasize the importance of evaluating the performance based on the AUBC, Final Dice, and PPM. In a best-case scenario, a QM has a high FG-Eff and excels in the other metrics or is among the better-performing methods.

Generally speaking, a QM which has a high FG-Eff but a very low performance based on all other metrics is not recommended as a good method, as the metric potentially can also be *hacked* by simply querying a very small amount of foreground and a very steep increase in performance relative to the amount of queried foreground.

**Limitation**   The annotation efficiency as a metric is only meaningful when compared on precisely the same model and training with the same starting budget and annotation budget because the estimated values $\hat{y}(\hat{t}_0)$ and $\hat{y}_{\text{full}}$ change resulting $\gamma$ values substantially. As the number of foreground voxels represents a proxy for annotation effort, the FG-Eff does not replace other performance metrics but should be seen as an extension of them.

Table 5: Dataset descriptions and configurations for the main study.

| Dataset | ACDC | AMOS | KiTS | Hippocampus |
|---|---|---|---|---|
| # Classes w.o. Background | 3 | 15 | 3 | 2 |
| Median Shape | 16.5x237x206 | 237.5x582x582 | 526x512x512 | 36x50x35 |
| Used Spacing | 2x0.6875x0.6875 | 5x1.5625x1.5625 | 0.78125x0.78125x0.78125 | 1x1x1 |
| # Pool & Training | 150 | 150 | 225 | 195 |
| # Validation | 50 | 50 | 75 | 65 |
| Budget: Low [# Patches](% Voxels) | 150 (0.75%) | 200 (0.26%) | 200 (0.16%) | 100 (6,51%) |
| Budget: Medium [# Patches](% Voxels) | 300(1.50%) | 1000 (1.30%) | 1000 (0.80%) | 200 (13,02%) |
| Budget: High [# Patches](% Voxels) | 450(2.25%) | 2500 (3.25%) | 2500 (2.00%) | 300 (19,54%) |
| Query Patch Size | 4x40x40 | 32x74x74 | 60x64x64 | 20x20x20 |
| $\frac{\text{Query Patch Size}}{\text{\# Voxels Dataset}}$ | 0.0050% | 0.0013% | 0.0008% | 0.06513% |
| Test set Mean Dice (1000 Epochs) | 0.912 | 0.893 | 0.773 | 0.895 |
| Test set Mean Dice (500 Epochs) | 0.912 | 0.883 | 0.751 | 0.895 |
| Test set Mean Dice (200 Epochs) | 0.910 | 0.860 | 0.705 | 0.895 |

# E    Dataset Details

Key dataset characteristics are shown in table 5.

**ACDC**    Class names in order of labels (ascending): right ventricle, myocardium, left ventricular cavity

**AMOS**    Class names in order of labels (ascending): spleen, right kidney, left kidney, gall bladder, esophagus, liver, stomach, aorta, postcava, pancreas, right adrenal gland, left adrenal gland, duodenum, bladder, prostate/uterus

**Hippocampus**    Class names in order of labels (ascending): anterior hippocampus, posterior hippocampus

**KiTS**    Class names in order of labels (ascending): kidney, kidney-tumor, kidney-cyst

# F   Main Study Results

> **TLDR:**
> - The Random baseline is clearly outperformed by all AL methods
> - No AL method substantially outperforms the Foreground Aware Random baseline (for 66% Foreground rate).
> - Predictive Entropy is the overall best performing AL method, but performance is highly variable, and it has the lowest FG-Eff.
> - AL performance is strongly related to dataset properties and the task.
> - Noisy QMs are more reliable than Greedy counterparts, however their performance decreases for larger annotation budgets.

The overall design of the main study, alongside details of the ablation studies, is shown in fig. 8. The detailed results with regard to AUBC, Final Dice and FG-Eff for each dataset and Label Regime are shown in table 6.

Further, we show the aggregated PPMs for each dataset separately in fig. 9.

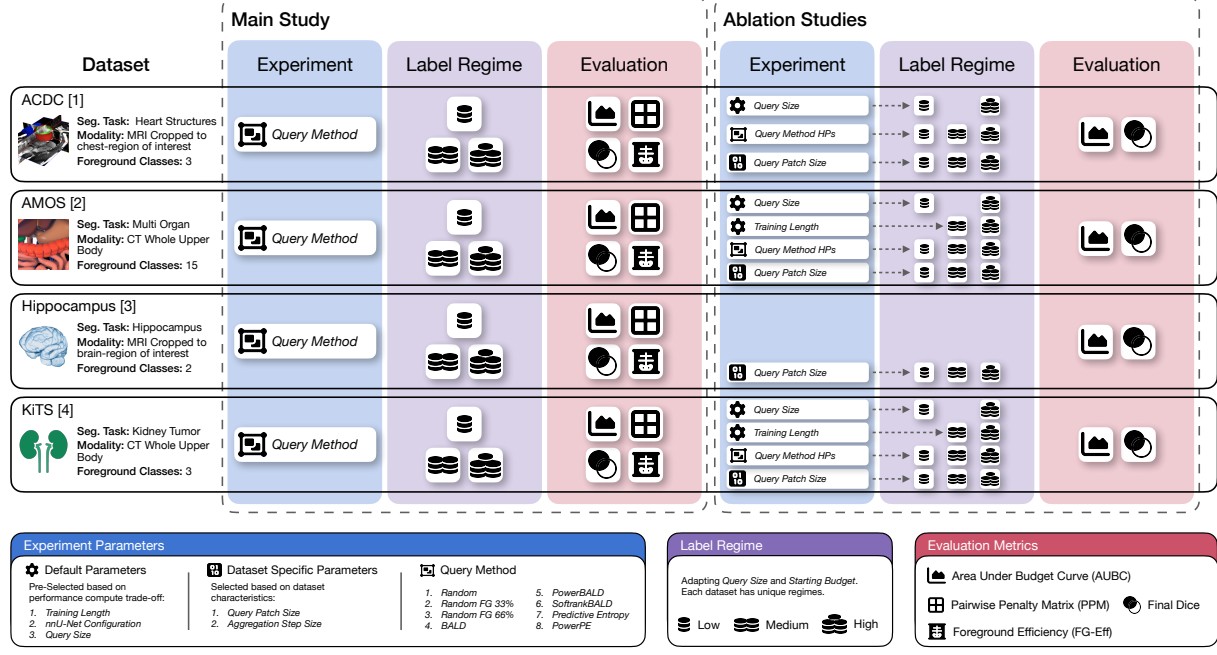

Figure 8: Systematic schema of our empirical study. It is comprised of one Main Study, which focuses on the evaluation of QMs, and four Ablation Studies analyzing the influence of specific design parameters on AL methods. *Query Method HP's* refers to the Noise strength in Noisy QMs ablation.

## F.1   AMOS

We show visualization of the queried patches for Predictive Entropy, PowerPE, Random 66% FG and Random on the AMOS Low-Label Regime in fig. 10.

An investigation of the performance of AL methods by the examples of Predictive Entropy and PowerPE when compared to Random and Random 66% FG are shown in fig. 11. It clearly shows that the main performance difference stems from a subset of classes which get less well predicted when not queried frequent enough.

Table 6: Fine-Grained Results for the Main Study for each dataset. Higher values are better and colorization goes from bright (best) to dark orange(worst). AUBC and Final Dice are reported with a factor ($\times 100$) for improved readability. AUBC, Final and Beta can only directly compared for each label regime on each dataset.

(a) ACDC

| Dataset | | | | ACDC | | | | | |
|---|---|---|---|---|---|---|---|---|---|
| Label Regime | | Low | | | Medium | | | High | |
| Metric | AUBC | Final Dice | FG-Eff | AUBC | Final Dice | FG-Eff | AUBC | Final Dice | FG-Eff |
| Query Method | | | | | | | | | |
| BALD | 79.84 ± 0.59 | 86.44 ± 0.96 | 26.99 ± 3.11 | 85.85 ± 0.45 | 89.62 ± 0.15 | 21.85 ± 4.16 | 87.74 ± 0.38 | 90.47 ± 0.18 | 14.99 ± 1.14 |
| PowerBALD | 81.18 ± 0.58 | 86.46 ± 0.55 | 46.30 ± 13.10 | 85.63 ± 0.37 | 89.07 ± 0.21 | 27.69 ± 3.96 | 87.50 ± 0.44 | 89.80 ± 0.17 | 17.83 ± 1.82 |
| SoftrankBALD | 80.71 ± 0.92 | 86.50 ± 0.95 | 35.72 ± 7.09 | 85.89 ± 0.49 | 89.33 ± 0.27 | 26.28 ± 4.97 | 87.28 ± 0.68 | 90.17 ± 0.14 | 14.44 ± 1.32 |
| Predictive Entropy | 80.02 ± 1.54 | 86.54 ± 0.95 | 26.50 ± 4.40 | 85.53 ± 0.59 | 89.42 ± 0.07 | 21.11 ± 3.07 | 87.65 ± 0.27 | 90.52 ± 0.06 | 13.50 ± 1.22 |
| PowerPE | 80.46 ± 0.30 | 86.56 ± 0.40 | 47.89 ± 14.09 | 85.24 ± 0.69 | 89.05 ± 0.22 | 27.86 ± 4.96 | 87.21 ± 0.60 | 89.67 ± 0.15 | 16.44 ± 1.18 |
| Random | 76.65 ± 0.81 | 80.34 ± 1.64 | 59.28 ± 33.54 | 82.24 ± 1.25 | 83.46 ± 0.87 | 38.10 ± 8.38 | 84.69 ± 0.96 | 86.28 ± 1.08 | 21.45 ± 3.85 |
| Random 33% FG | 81.28 ± 0.56 | 85.09 ± 1.14 | 40.89 ± 9.71 | 84.61 ± 0.65 | 87.51 ± 0.56 | 21.22 ± 1.48 | 86.95 ± 0.74 | 89.06 ± 0.44 | 15.72 ± 1.42 |
| Random 66% FG | 82.32 ± 0.33 | 86.70 ± 0.48 | 31.21 ± 4.32 | 86.16 ± 0.44 | 88.62 ± 0.52 | 18.92 ± 2.12 | 87.86 ± 0.33 | 89.94 ± 0.09 | 13.38 ± 0.80 |

(b) AMOS

| Dataset | | | | AMOS | | | | | |
|---|---|---|---|---|---|---|---|---|---|
| Label Regime | | Low | | | Medium | | | High | |
| Metric | AUBC | Final Dice | FG-Eff | AUBC | Final Dice | FG-Eff | AUBC | Final Dice | FG-Eff |
| Query Method | | | | | | | | | |
| BALD | 38.69 ± 2.34 | 34.05 ± 1.58 | -22.66 ± 8.50 | 52.56 ± 2.74 | 59.26 ± 2.73 | 1.54 ± 0.22 | 69.38 ± 0.70 | 74.95 ± 2.38 | -0.45 ± 0.20 |
| PowerBALD | 50.34 ± 3.00 | 56.18 ± 1.24 | 3.65 ± 14.56 | 66.11 ± 1.47 | 73.02 ± 2.01 | 18.28 ± 0.44 | 77.86 ± 0.14 | 80.48 ± 0.48 | 8.80 ± 0.08 |
| SoftrankBALD | 44.49 ± 1.56 | 45.75 ± 0.95 | -11.38 ± 4.19 | 60.01 ± 0.69 | 66.72 ± 0.65 | 5.70 ± 0.10 | 75.29 ± 1.46 | 81.23 ± 1.18 | 3.52 ± 0.38 |
| Predictive Entropy | 38.02 ± 3.35 | 39.19 ± 6.79 | -17.92 ± 8.49 | 56.30 ± 1.78 | 62.07 ± 1.39 | 2.65 ± 0.17 | 71.27 ± 1.52 | 80.79 ± 2.07 | 1.02 ± 0.41 |
| PowerPE | 47.66 ± 2.50 | 50.04 ± 2.30 | -9.80 ± 12.14 | 66.74 ± 2.80 | 73.68 ± 0.92 | 18.60 ± 1.18 | 77.92 ± 0.29 | 80.52 ± 0.16 | 8.87 ± 0.10 |
| Random | 42.26 ± 2.55 | 36.36 ± 2.92 | -134.82 ± 89.07 | 54.65 ± 2.82 | 56.22 ± 4.61 | 10.34 ± 3.29 | 73.82 ± 0.50 | 75.48 ± 0.37 | 7.38 ± 0.62 |
| Random 33% FG | 58.05 ± 1.54 | 62.95 ± 1.03 | 35.45 ± 11.43 | 71.78 ± 1.16 | 78.60 ± 0.37 | 36.58 ± 2.99 | 79.53 ± 0.38 | 82.68 ± 0.19 | 14.44 ± 0.47 |
| Random 66% FG | 62.84 ± 1.88 | 71.11 ± 1.42 | 43.63 ± 9.82 | 74.87 ± 0.64 | 80.72 ± 0.54 | 32.62 ± 6.15 | 80.98 ± 0.19 | 83.81 ± 0.32 | 12.33 ± 0.43 |

(c) Hippocampus

| Dataset | | | | Hippocampus | | | | | |
|---|---|---|---|---|---|---|---|---|---|
| Label Regime | | Low | | | Medium | | | High | |
| Metric | AUBC | Final Dice | FG-Eff | AUBC | Final Dice | FG-Eff | AUBC | Final Dice | FG-Eff |
| Query Method | | | | | | | | | |
| BALD | 88.46 ± 0.03 | 88.87 ± 0.06 | 9.58 ± 0.97 | 88.79 ± 0.02 | 89.18 ± 0.07 | 4.52 ± 0.06 | 89.03 ± 0.05 | 89.42 ± 0.05 | 3.49 ± 0.12 |
| PowerBALD | 88.20 ± 0.08 | 88.77 ± 0.11 | 9.21 ± 0.49 | 88.76 ± 0.04 | 89.16 ± 0.06 | 5.55 ± 0.07 | 88.98 ± 0.07 | 89.29 ± 0.10 | 3.90 ± 0.15 |
| SoftrankBALD | 88.44 ± 0.11 | 88.93 ± 0.18 | 9.61 ± 0.98 | 88.72 ± 0.08 | 89.12 ± 0.02 | 3.90 ± 0.05 | 89.03 ± 0.06 | 89.42 ± 0.07 | 3.60 ± 0.12 |
| Predictive Entropy | 88.50 ± 0.06 | 88.90 ± 0.10 | 9.75 ± 1.01 | 88.81 ± 0.04 | 89.18 ± 0.07 | 4.23 ± 0.06 | 89.07 ± 0.07 | 89.54 ± 0.03 | 3.74 ± 0.19 |
| PowerPE | 88.16 ± 0.08 | 88.70 ± 0.11 | 9.25 ± 0.52 | 88.63 ± 0.09 | 89.07 ± 0.21 | 4.41 ± 0.10 | 88.97 ± 0.07 | 89.33 ± 0.18 | 4.08 ± 0.24 |
| Random | 88.07 ± 0.10 | 88.58 ± 0.08 | 8.76 ± 0.47 | 88.65 ± 0.11 | 89.07 ± 0.04 | 5.10 ± 0.08 | 88.96 ± 0.09 | 89.29 ± 0.20 | 4.41 ± 0.25 |
| Random 33% FG | 88.22 ± 0.16 | 88.70 ± 0.08 | 9.60 ± 0.81 | 88.77 ± 0.13 | 89.22 ± 0.14 | 6.20 ± 0.17 | 88.94 ± 0.06 | 89.33 ± 0.10 | 3.85 ± 0.15 |
| Random 66% FG | 88.28 ± 0.13 | 88.76 ± 0.14 | 9.87 ± 0.73 | 88.63 ± 0.02 | 89.02 ± 0.04 | 4.21 ± 0.03 | 88.92 ± 0.08 | 89.26 ± 0.06 | 3.33 ± 0.11 |

(d) KiTS

| Dataset | | | | KiTS | | | | | |
|---|---|---|---|---|---|---|---|---|---|
| Label Regime | | Low | | | Medium | | | High | |
| Metric | AUBC | Final Dice | FG-Eff | AUBC | Final Dice | FG-Eff | AUBC | Final Dice | FG-Eff |
| Query Method | | | | | | | | | |
| BALD | 40.58 ± 2.75 | 44.03 ± 3.18 | 7.96 ± 0.82 | 55.06 ± 1.20 | 61.97 ± 1.49 | 6.52 ± 0.14 | 62.53 ± 0.84 | 67.57 ± 1.72 | 9.35 ± 0.46 |
| PowerBALD | 45.10 ± 2.91 | 47.67 ± 3.63 | 25.24 ± 6.06 | 54.53 ± 1.40 | 59.51 ± 1.15 | 10.18 ± 0.41 | 61.24 ± 0.57 | 65.04 ± 0.81 | 11.89 ± 0.63 |
| SoftrankBALD | 42.87 ± 2.91 | 47.12 ± 3.34 | 12.41 ± 2.03 | 54.83 ± 1.79 | 61.44 ± 2.02 | 7.00 ± 0.27 | 62.49 ± 0.74 | 67.00 ± 0.97 | 9.82 ± 0.65 |
| Predictive Entropy | 40.62 ± 2.74 | 45.53 ± 3.57 | 7.04 ± 0.64 | 57.42 ± 0.54 | 65.39 ± 0.51 | 6.20 ± 0.10 | 64.00 ± 0.15 | 68.74 ± 0.65 | 7.83 ± 0.21 |
| PowerPE | 45.30 ± 2.05 | 49.62 ± 1.13 | 28.70 ± 3.74 | 54.76 ± 1.10 | 58.67 ± 1.53 | 9.69 ± 0.27 | 60.66 ± 0.66 | 63.62 ± 1.19 | 9.60 ± 0.51 |
| Random | 38.75 ± 3.36 | 39.19 ± 4.13 | 28.46 ± 19.48 | 47.82 ± 1.84 | 48.41 ± 1.99 | 4.10 ± 2.74 | 53.80 ± 0.68 | 55.12 ± 1.27 | 8.85 ± 1.21 |
| Random 33% FG | 43.70 ± 0.87 | 47.35 ± 2.10 | 16.18 ± 1.32 | 51.50 ± 1.97 | 54.08 ± 2.76 | 3.28 ± 0.15 | 55.30 ± 1.26 | 56.79 ± 1.02 | 1.87 ± 0.04 |
| Random 66% FG | 44.97 ± 2.01 | 46.83 ± 2.53 | 11.28 ± 1.30 | 50.78 ± 0.97 | 51.67 ± 2.31 | 1.25 ± 0.02 | 53.73 ± 1.78 | 55.90 ± 0.84 | 0.68 ± 0.01 |

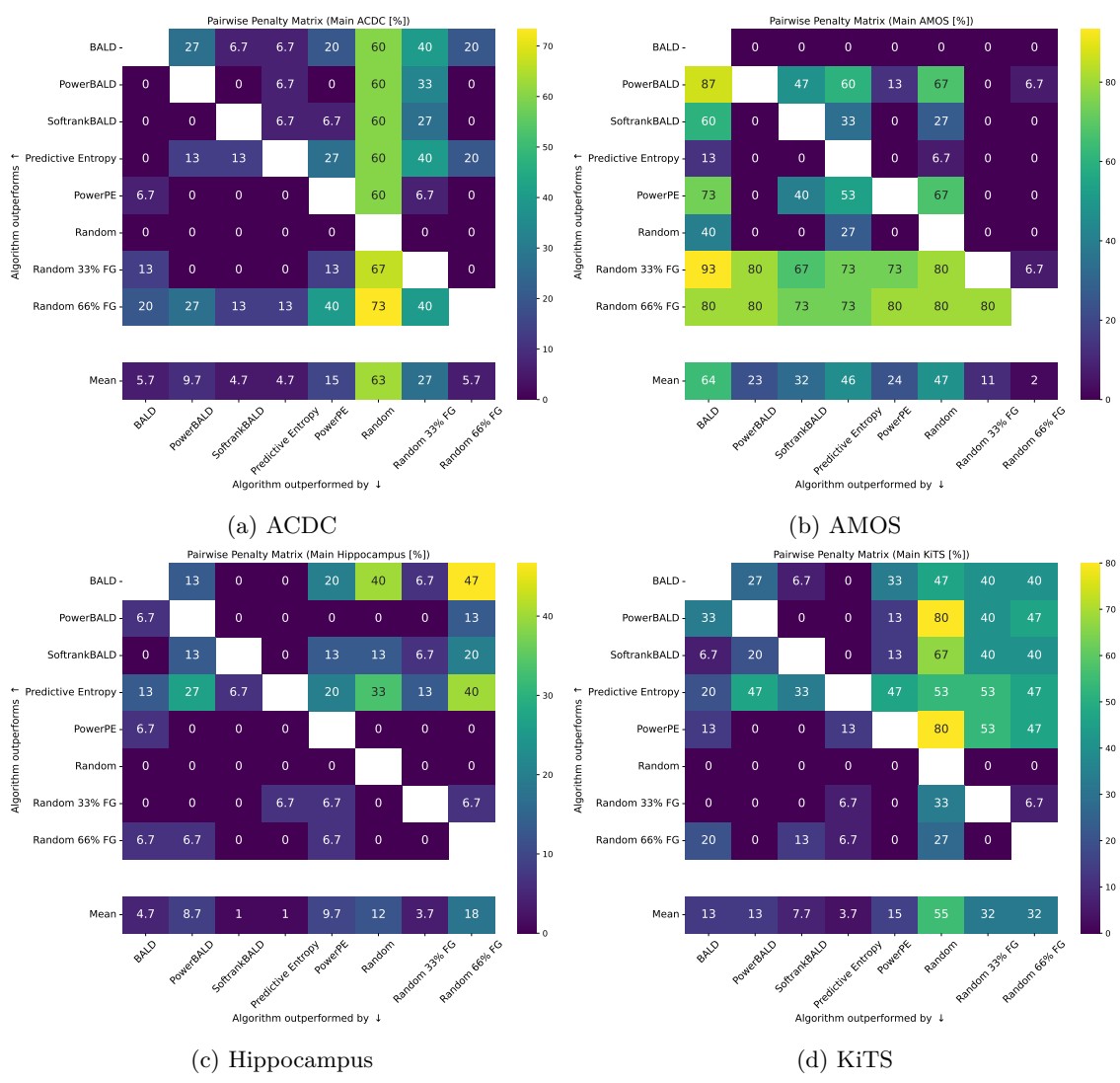

(a) ACDC

(b) AMOS

(c) Hippocampus

(d) KiTS

Figure 9: Pairwise Penalty Matrix aggregated over all Label Regimes for each dataset of the main study.

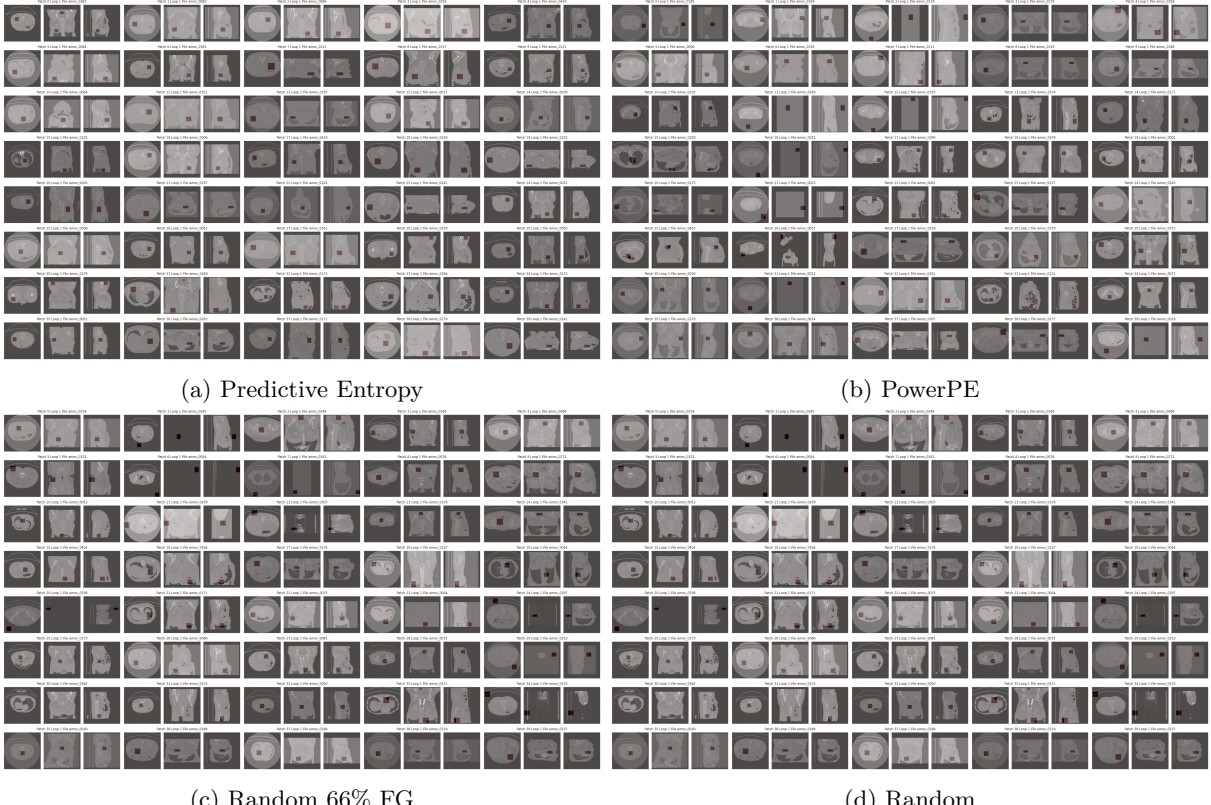

(a) Predictive Entropy

(b) PowerPE

(c) Random 66% FG

(d) Random

Figure 10: Queries of the first AL loop on the Low-Label Regime on **AMOS**. Red colored areas are selected patches.

Best viewed on screen with Zoom.

Predictive entropy purely queries regions inside the body with a specific focus on some regions, whereas PowerPE also queries some regions at the borders and is more diverse overall. Random 66% FG queries from multiple regions of the body, but also queries from the outside, and Random queries from quite a substantial amount of regions purely containing air.

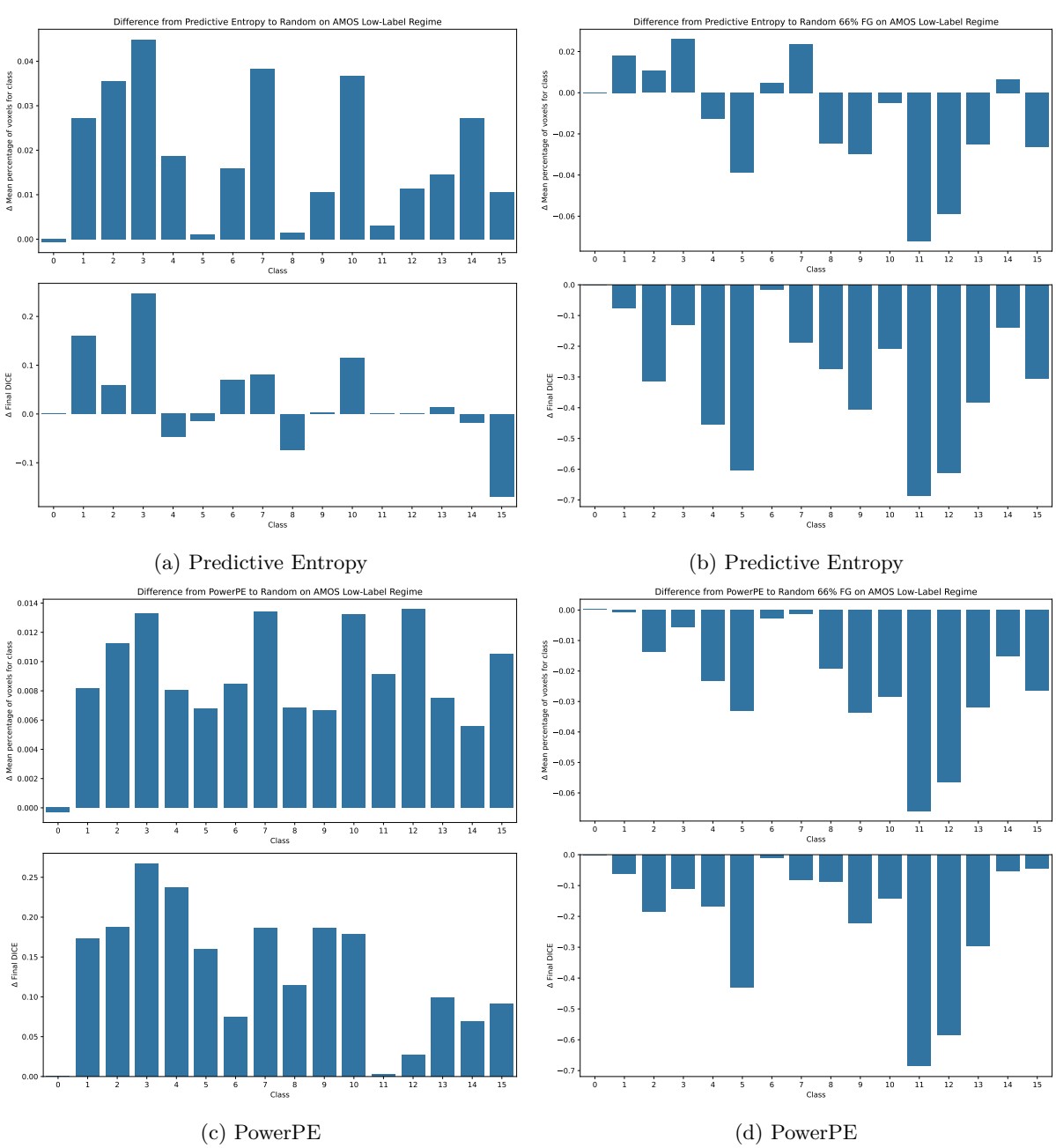

Figure 11: Visualization of the difference of the percentage of voxels for all classes alongside Final Dice performance on the AMOS Low-Label Regime from Predictive Entropy & PowerPE to Random and Random 66% FG. It shows that less data containing classes 11 & 12 (right & left adrenal gland) is queried by Predictive Entropy and PowerPe (also Random) (5% less of the overall voxels of that class), which is strongly correlated with the Final Dice for these classes being 0. For class 5 (esophagus), a similar behavior can be observed for Predictive Entropy, though not as pronounced. Compared to Predictive Entropy, this effect is weaker for PowerPE, which also queries more data from this class.

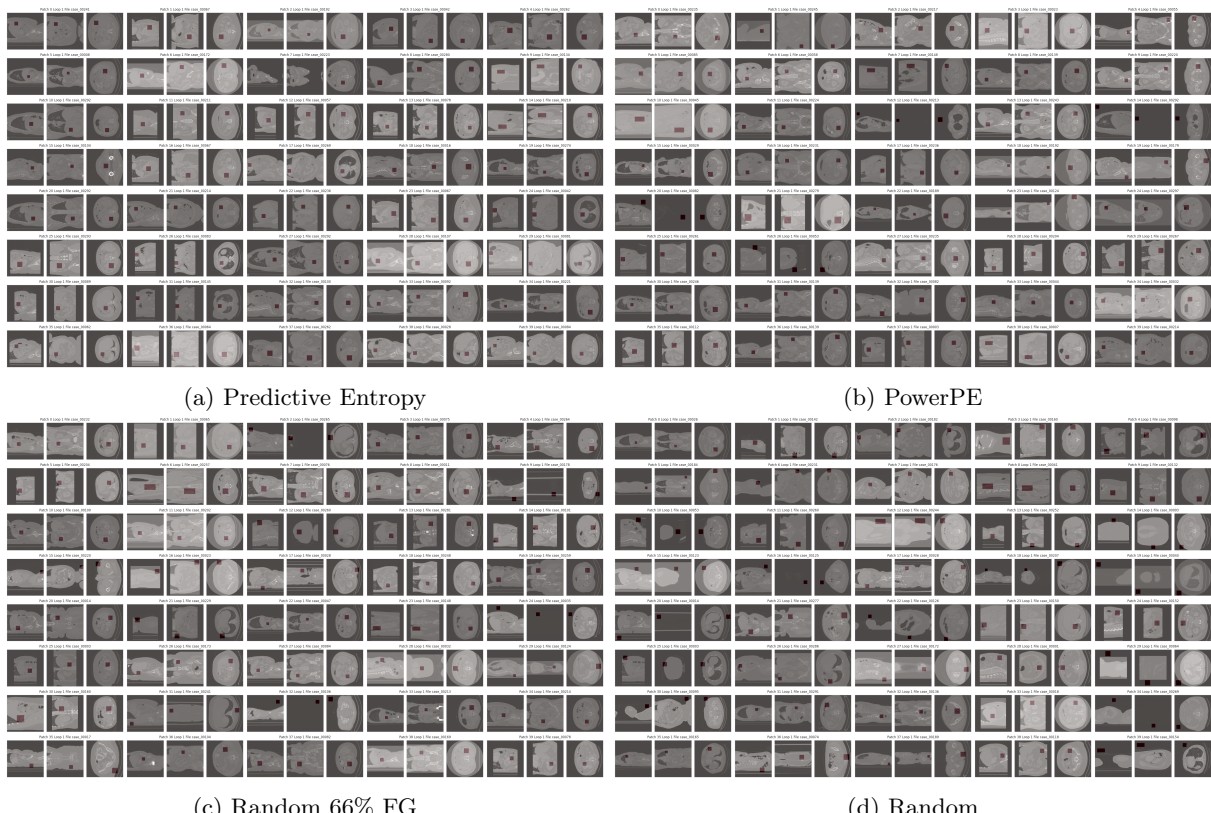

(a) Predictive Entropy

(b) PowerPE

(c) Random 66% FG

(d) Random

Figure 12: Queries of the first AL loop on the Low-Label Regime on **KiTS**. Red colored areas are selected patches.
Best viewed on screen with Zoom.
Predictive entropy purely queries regions inside the body with a specific focus on the kidneys. In contrast, PowerPE also covers different regions all over the body, still focusing on the kidney, but is more diverse overall. Random 66% FG queries regions in the area of the kidney, but also covers the entire body with some queries containing purely/mostly air. Random queries from quite a substantial number of regions purely containing air.

## F.2 KiTS

We show visualization of the queried patches for Predictive Entropy, PowerPE, Random 66% FG and Random on the AMOS Low-Label Regime in fig. 12.

# G    Detailed Ablations

Detailed analysis of the ablations can be found in the following subsections:

**Ablation 1** Query Size: appendix G.1
**Ablation 2** Training Length: appendix G.2
**Ablation 3** Noise strength in Noisy QMs: appendix G.3
**Ablation 4** Query Patch Size: appendix G.4

An overview of all experiments of the main study and the ablations is given in fig. 8.

Table 7: **Do smaller query sizes improve the performance of QMs?** Kendall's $\tau$ correlations between smaller query size and performance measures. Higher values indicate that smaller query sizes tend to yield better performance. The correlation values range between -1 and 1, where positive values suggest a beneficial effect of smaller queries, while negative values indicate the opposite. A two-sided test was performed with a significance level of $\alpha = 0.1$.

Colorscheme: ■ Significant & positive correleation, ■ positive correlation, ■ negative correlation, ■ significant & negative correlation

<table>
<tr><td colspan="7">(a) Query Size & AUBC</td></tr>
<tr><td>Dataset</td><td colspan="2">ACDC</td><td colspan="2">AMOS</td><td colspan="2">KiTS</td></tr>
<tr><td>Label Regime</td><td>Low</td><td>High</td><td>Low</td><td>High</td><td>Low</td><td>High</td></tr>
<tr><td>Query Method</td><td></td><td></td><td></td><td></td><td></td><td></td></tr>
<tr><td>BALD</td><td>0.711</td><td>0.604</td><td>-0.391</td><td>0.249</td><td>0.249</td><td>0.497</td></tr>
<tr><td>PowerBALD</td><td>0.178</td><td>0.497</td><td>0.426</td><td>-0.036</td><td>0.071</td><td>0.533</td></tr>
<tr><td>SoftrankBALD</td><td>0.640</td><td>0.569</td><td>0.462</td><td>-0.071</td><td>0.178</td><td>0.462</td></tr>
<tr><td>Predictive Entropy</td><td>0.462</td><td>0.711</td><td>0.142</td><td>0.178</td><td>0.391</td><td>0.640</td></tr>
<tr><td>PowerPE</td><td>0.391</td><td>0.355</td><td>0.462</td><td>-0.142</td><td>0.320</td><td>-0.142</td></tr>
</table>

<table>
<tr><td colspan="7">(b) Query Size & Final Dice</td></tr>
<tr><td>Dataset</td><td colspan="2">ACDC</td><td colspan="2">AMOS</td><td colspan="2">KiTS</td></tr>
<tr><td>Label Regime</td><td>Low</td><td>High</td><td>Low</td><td>High</td><td>Low</td><td>High</td></tr>
<tr><td>Query Method</td><td></td><td></td><td></td><td></td><td></td><td></td></tr>
<tr><td>BALD</td><td>0.640</td><td>0.284</td><td>0.107</td><td>0.711</td><td>0.426</td><td>0.178</td></tr>
<tr><td>PowerBALD</td><td>0.142</td><td>0.213</td><td>0.320</td><td>-0.107</td><td>-0.071</td><td>0.604</td></tr>
<tr><td>SoftrankBALD</td><td>0.533</td><td>0.426</td><td>0.391</td><td>-0.036</td><td>0.142</td><td>0.249</td></tr>
<tr><td>Predictive Entropy</td><td>0.355</td><td>0.569</td><td>0.569</td><td>0.853</td><td>0.391</td><td>0.462</td></tr>
<tr><td>PowerPE</td><td>0.213</td><td>0.426</td><td>0.497</td><td>0.036</td><td>0.320</td><td>-0.391</td></tr>
</table>

## G.1 Query Size Ablation

> **TLDR:**
> - Smaller query sizes generally lead to better performance than larger query sizes for AL across all our AL QMs.
> - Smaller query sizes have a stronger impact on Greedy QMs than on Noisy QMs
> - Smaller query sizes can change ranking to favoring AL QMs from favoring Random strategies.
> - As reducing the query size by a factor of 2 doubles the compute cost of AL we argue that 5 Loops for a Label Regime are sufficient compared to 9 Loops when training models from scratch.

To assess the impact of query size on AL QMs, we conduct ablation studies using the same absolute annotation budgets as in our main experiments while varying the query sizes. We evaluate three different query sizes of twice the size (QSx2), identical size (QSx1) and half the size (QSx1/2) for one specific starting budget of our main study. This variation results in approximately half or double the number of AL loops, allowing us to analyze how different query sizes influence the performance of AL QMs, separate from other factors. The evaluation is based on two key metrics: the Final Dice score and the AUBC, which is computed only on the overlapping annotation budgets available across all three settings to ensure comparability across different query sizes.

These experiments are conducted on the AMOS, KiTS, and ACDC datasets for both Low- and High-Label Regimes to observe the behavior at the extreme settings.

By analyzing multiple datasets and annotation scales, we aim to gain a comprehensive understanding of how query size affects the performance of AL in different medical imaging contexts through answering the following questions regarding the influence of the query size:

**Q1: Do AL QMs Benefit from Smaller query sizes?** To investigate this, we analyze the correlation between query size and performance using a Kendall's $\tau$ (Kendall, 1948) correlation test on AUBC and Final Dice values. The results, presented in table 7, indicate that smaller query sizes consistently improve performance of our benchmarked methods as across all evaluated QMs, we observe significant positive correlations and no significant negative correlations. Generally the effect of smaller query sizes have a strong positive impact on the Greedy QMs as they have three significant positive results for both AUBC and Final Dice.

Notably, in the ACDC and KiTS high-budget setting, fewer significant results are observed for Final Dice compared to AUBC, which is counterintuitive given that smaller query sizes are generally expected to provide cumulative benefits. We hypothesize that this occurs because, at high annotation budgets, a substantial portion of the foreground structures in ACDC is already annotated and the performance of the underlying segmentation model is already 'good' – meaning that the decision boundaries does not travel high-density

areas of potential queries. As a result, this makes it less likely for larger query sizes to select multiple redundant samples. A similar effect, that for generally larger budgets the benefits of smaller query sizes tend to reduce, has been previously reported by Kirsch et al. (2023) for object recognition.

**Q2: How does the Query Size influence rankings of annotation strategies?** To investigate this, we analyze the ranking of all annotation strategies for both AUBC and Final Dice with Kendall's $\tau$ for each Label Regime and Dataset between pairs of query sizes in table 8. Generally, we observe that no ranking is negatively correlated and significant, and that over half of the results are robust (positively correlated and significant).

The rankings for AMOS Low-Label Regime and KiTS High-Label Regime show very little change for both AUBC and Final Dice are robust across all compared query sizes. On the AMOS Low-Label Regime, Random FG strategies perform best for all query sizes (see table 9c), and on the KiTS High-Label Regime, AL QMs like Predictive Entropy perform best (see table 10b).

Looking at the non-robust settings we observe for the corresponding datasets and Label Regimes, we will elaborate on these changes based on detailed results with rankings shown in appendix G.1.

The strongest ranking perturbation is on ACDC where for the Low-Label Regime with QSx1/2 most AL QMs outperform all Random FG Strategies in terms of AUBC and for the Final Dice leading especially for the AUBC to a strong difference in ranking since the Random FG has the best AUBC for QSx1 and QSx2 (see table 9a).

Similar behavior can be observed for the ACDC High-Label Regime, where, however, again a change in ranking occurs from smaller to larger query sizes, which favors AL QMs over Random and Random FG strategies in terms of AUBC (table 9b). For the Final Dice no such trend can be observed and the ranking remains stable.

On the AMOS High-Label Regime, the ranking perturbations stem from increased performance of the Predictive Entropy, especially with regard to the Final Dice leads for smaller query sizes. These lead to its rankings being strongly influenced from the 2nd best ranked strategy for QSx1/2 to QSx2, the 2nd to worst ranked strategy for QSx2 in terms of Final Dice, with similar trends for all other AL QMs, which are more pronounced for the AUBC (table 9d).

On the KiTS Low-Label Regime the performance changes occur mostly from the QSx1/2 and QSx1 to the QSx2 ranking where for the smaller query sizes PowerPE leads to the best performance in terms of AUBC and Final DICE while for QSx2 it is among the worst performing methods and outperformed by Random 66% FG (table 10b).

Overall, a change in query size can lead to substantial changes in the ranking of AL QMs relative to random strategies, with Greedy QMs especially being affected, swinging from among the best-performing to the worst-performing methods for larger query sizes.

For benchmarking purposes, we believe that reasonably chosen query sizes for a given entire annotation budget, resulting in at least 4 annotation rounds, should suffice, as the correlation between QSx1/2 and QSx1 is significantly positively correlated 5 times out of 6. Especially considering that decreasing the QS by a factor of 2 essentially doubles the compute cost of employing AL the returns are diminishing.

Table 8: **How robust are method rankings to changes in query size?** Kendall's $\tau$ corellations between rankings of QMs with different query sizes. A high value indicates that the rankings between the two settings are similar while lower values denote that they differ. A two-sided test was performed with a significance level of $\alpha = 0.1$.

Colorscheme: ■ Significant & positive correleation, ■ positive correlation, ■ negative correlation, ■ significant & negative correlation

(a) Ranking Correlation AUBC

| Dataset | ACDC | | AMOS | | KiTS | |
|---|---|---|---|---|---|---|
| Label Regime Setting | Low | High | Low | High | Low | High |
| QSx2 vs QSx1 | 0.571 | 0.571 | 0.857 | 0.857 | 0.500 | 0.786 |
| QSx2 vs QSx1/2 | -0.214 | 0.286 | 0.786 | 0.929 | 0.500 | 0.786 |
| QSx1 vs QSx1/2 | 0.071 | 0.714 | 0.929 | 0.786 | 0.857 | 1.000 |
| Mean | 0.143 | 0.524 | 0.857 | 0.857 | 0.619 | 0.857 |

(b) Ranking Correlation Final Dice

| Dataset | ACDC | | AMOS | | KiTS | |
|---|---|---|---|---|---|---|
| Label Regime Setting | Low | High | Low | High | Low | High |
| QSx2 vs QSx1 | 0.786 | 0.714 | 0.786 | 0.571 | 0.643 | 0.929 |
| QSx2 vs QSx1/2 | 0.286 | 0.643 | 0.786 | 0.500 | 0.429 | 0.857 |
| QSx1 vs QSx1/2 | 0.357 | 0.786 | 1.000 | 0.643 | 0.643 | 0.929 |
| Mean | 0.476 | 0.714 | 0.857 | 0.571 | 0.571 | 0.905 |

**Query Size Ablation Detailed Results**

Table 9: Fine-Grained Results for the query size ablation on ACDC and AMOS. AUBC and Final Dice are reported with a factor (×100) for improved readability. Colors indicate the ranking, darker colors correspond to worse rankings.

(a) ACDC Low Label Regime

| Dataset | ACDC | | | | | |
|---|---|---|---|---|---|---|
| Setting | QSx1/2 | | QSx1 | | QSx2 | |
| Metric | AUBC | Final Dice | AUBC | Final Dice | AUBC | Final Dice |
| Query Method | | | | | | |
| BALD | 81.16 ± 0.36 | 87.42 ± 0.49 | 79.86 ± 1.00 | 86.44 ± 0.96 | 78.20 ± 1.87 | 85.21 ± 1.99 |
| PowerBALD | 79.55 ± 0.76 | 85.35 ± 0.90 | 80.43 ± 0.25 | 86.46 ± 0.55 | 79.08 ± 0.64 | 85.23 ± 0.31 |
| SoftrankBALD | 80.94 ± 1.02 | 87.13 ± 0.30 | 80.34 ± 1.05 | 86.50 ± 0.95 | 77.98 ± 0.98 | 85.66 ± 0.81 |
| Predictive Entropy | 80.70 ± 0.43 | 87.30 ± 0.80 | 79.73 ± 1.40 | 86.54 ± 0.95 | 78.32 ± 2.19 | 86.33 ± 0.46 |
| PowerPE | 79.81 ± 0.90 | 86.08 ± 1.17 | 79.96 ± 0.44 | 86.56 ± 0.40 | 78.81 ± 0.84 | 85.86 ± 0.28 |
| Random | 76.21 ± 1.07 | 80.61 ± 2.13 | 75.98 ± 1.29 | 80.34 ± 1.64 | 76.97 ± 1.56 | 81.32 ± 1.61 |
| Random 33% FG | 78.89 ± 1.46 | 83.80 ± 0.76 | 80.06 ± 0.95 | 85.09 ± 1.14 | 79.70 ± 1.01 | 85.26 ± 0.75 |
| Random 66% FG | 80.28 ± 0.56 | 86.10 ± 0.74 | 81.16 ± 0.34 | 86.70 ± 0.48 | 80.74 ± 0.92 | 86.52 ± 0.79 |

(b) ACDC High Label Regime

| Dataset | ACDC | | | | | |
|---|---|---|---|---|---|---|
| Setting | QSx1/2 | | QSx1 | | QSx2 | |
| Metric | AUBC | Final Dice | AUBC | Final Dice | AUBC | Final Dice |
| Query Method | | | | | | |
| BALD | 87.73 ± 0.55 | 90.52 ± 0.10 | 87.54 ± 0.28 | 90.47 ± 0.18 | 86.72 ± 0.58 | 90.38 ± 0.14 |
| PowerBALD | 87.36 ± 0.45 | 89.96 ± 0.25 | 87.26 ± 0.31 | 89.80 ± 0.17 | 86.69 ± 0.59 | 89.52 ± 0.69 |
| SoftrankBALD | 87.45 ± 0.35 | 90.18 ± 0.10 | 87.14 ± 0.61 | 90.17 ± 0.14 | 86.46 ± 0.45 | 89.65 ± 0.43 |
| Predictive Entropy | 87.93 ± 0.25 | 90.60 ± 0.12 | 87.61 ± 0.35 | 90.52 ± 0.06 | 86.82 ± 0.52 | 90.29 ± 0.19 |
| PowerPE | 87.27 ± 0.47 | 89.96 ± 0.19 | 86.99 ± 0.54 | 89.67 ± 0.15 | 86.55 ± 0.83 | 89.71 ± 0.21 |
| Random | 84.85 ± 1.10 | 86.65 ± 1.30 | 84.55 ± 0.83 | 86.28 ± 1.08 | 84.32 ± 0.98 | 85.69 ± 1.17 |
| Random 33% FG | 86.53 ± 0.68 | 89.02 ± 0.30 | 86.75 ± 0.69 | 89.06 ± 0.44 | 86.71 ± 0.61 | 89.28 ± 0.57 |
| Random 66% FG | 87.27 ± 0.60 | 89.79 ± 0.29 | 87.34 ± 0.25 | 89.94 ± 0.09 | 87.34 ± 0.34 | 90.13 ± 0.20 |

(c) AMOS Low Label Regime

| Dataset | AMOS | | | | | |
|---|---|---|---|---|---|---|
| Setting | QSx1/2 | | QSx1 | | QSx2 | |
| Metric | AUBC | Final Dice | AUBC | Final Dice | AUBC | Final Dice |
| Query Method | | | | | | |
| BALD | 38.70 ± 0.71 | 35.74 ± 2.82 | 38.86 ± 2.57 | 34.05 ± 1.58 | 41.79 ± 2.83 | 35.92 ± 5.10 |
| PowerBALD | 51.99 ± 2.05 | 55.70 ± 1.24 | 50.55 ± 3.18 | 56.18 ± 1.24 | 48.31 ± 3.13 | 52.32 ± 2.71 |
| SoftrankBALD | 44.30 ± 2.27 | 45.10 ± 5.16 | 44.55 ± 0.79 | 45.75 ± 0.95 | 41.53 ± 2.29 | 39.13 ± 5.29 |
| Predictive Entropy | 40.98 ± 2.77 | 41.45 ± 3.97 | 38.47 ± 3.86 | 39.19 ± 6.79 | 38.93 ± 4.05 | 29.96 ± 4.29 |
| PowerPE | 51.58 ± 3.18 | 54.49 ± 2.81 | 47.94 ± 3.15 | 50.04 ± 2.30 | 46.38 ± 5.15 | 49.41 ± 6.28 |
| Random | 43.47 ± 3.25 | 40.39 ± 2.87 | 42.24 ± 2.48 | 36.36 ± 2.92 | 40.85 ± 2.70 | 35.83 ± 2.23 |
| Random 33% FG | 54.84 ± 2.83 | 58.72 ± 2.01 | 57.64 ± 1.86 | 62.95 ± 1.03 | 56.45 ± 1.39 | 63.51 ± 3.30 |
| Random 66% FG | 62.24 ± 1.51 | 70.96 ± 0.83 | 62.04 ± 1.57 | 71.11 ± 1.42 | 61.96 ± 2.89 | 71.90 ± 1.58 |

(d) AMOS High Label Regime

| Dataset | AMOS | | | | | |
|---|---|---|---|---|---|---|
| Setting | QSx1/2 | | QSx1 | | QSx2 | |
| Metric | AUBC | Final Dice | AUBC | Final Dice | AUBC | Final Dice |
| Query Method | | | | | | |
| BALD | 72.87 ± 1.61 | 78.70 ± 3.51 | 69.94 ± 0.55 | 74.95 ± 2.38 | 70.67 ± 0.49 | 69.88 ± 1.31 |
| PowerBALD | 77.31 ± 0.22 | 80.42 ± 0.37 | 77.51 ± 0.25 | 80.48 ± 0.48 | 77.45 ± 0.45 | 80.55 ± 0.37 |
| SoftrankBALD | 75.12 ± 0.84 | 79.90 ± 1.13 | 75.11 ± 1.39 | 81.23 ± 1.18 | 75.42 ± 1.30 | 79.86 ± 1.11 |
| Predictive Entropy | 75.22 ± 2.04 | 83.05 ± 0.26 | 72.06 ± 1.50 | 80.79 ± 2.07 | 73.83 ± 1.56 | 71.98 ± 2.09 |
| PowerPE | 77.43 ± 0.67 | 80.48 ± 0.16 | 77.36 ± 0.26 | 80.52 ± 0.16 | 77.60 ± 0.27 | 80.29 ± 0.62 |
| Random | 73.22 ± 0.88 | 74.30 ± 1.46 | 73.95 ± 0.45 | 75.48 ± 0.37 | 73.78 ± 0.20 | 75.44 ± 0.57 |
| Random 33% FG | 78.97 ± 0.42 | 82.37 ± 0.33 | 79.00 ± 0.32 | 82.68 ± 0.19 | 79.21 ± 0.33 | 82.57 ± 0.12 |
| Random 66% FG | 80.15 ± 0.09 | 83.86 ± 0.17 | 80.32 ± 0.27 | 83.81 ± 0.32 | 80.12 ± 0.39 | 83.67 ± 0.31 |

Table 10: Fine-Grained Results for the query size ablation on KiTS. AUBC and Final Dice are reported with a factor ($\times100$) for improved readability. Colors indicate the ranking, darker colors correspond to worse rankings.

(a) KiTS Low Label Regime

| Dataset | KiTS | | | | | |
|---|---|---|---|---|---|---|
| Setting | QSx1/2 | | QSx1 | | QSx2 | |
| Metric | AUBC | Final Dice | AUBC | Final Dice | AUBC | Final Dice |
| Query Method | | | | | | |
| BALD | $41.15 \pm 2.91$ | $44.04 \pm 2.12$ | $40.43 \pm 3.18$ | $44.03 \pm 3.18$ | $38.47 \pm 2.89$ | $39.62 \pm 3.95$ |
| PowerBALD | $44.18 \pm 3.62$ | $47.71 \pm 4.15$ | $44.29 \pm 2.84$ | $47.67 \pm 3.63$ | $43.76 \pm 2.63$ | $48.13 \pm 2.91$ |
| SoftrankBALD | $42.52 \pm 2.76$ | $44.51 \pm 1.87$ | $42.91 \pm 2.75$ | $47.12 \pm 3.34$ | $40.26 \pm 3.93$ | $43.25 \pm 4.67$ |
| Predictive Entropy | $42.69 \pm 3.58$ | $47.44 \pm 4.65$ | $40.17 \pm 2.76$ | $45.53 \pm 3.57$ | $39.20 \pm 2.77$ | $41.81 \pm 5.50$ |
| PowerPE | $45.19 \pm 2.73$ | $49.57 \pm 3.10$ | $44.64 \pm 1.68$ | $49.62 \pm 1.13$ | $42.83 \pm 2.99$ | $46.39 \pm 2.75$ |
| Random | $38.09 \pm 2.22$ | $38.45 \pm 3.48$ | $38.71 \pm 3.46$ | $39.19 \pm 4.13$ | $37.76 \pm 3.10$ | $37.98 \pm 2.98$ |
| Random 33% FG | $43.89 \pm 0.63$ | $46.99 \pm 2.20$ | $43.60 \pm 0.92$ | $47.35 \pm 2.10$ | $43.97 \pm 1.42$ | $48.17 \pm 2.32$ |
| Random 66% FG | $44.48 \pm 1.89$ | $48.01 \pm 0.90$ | $44.32 \pm 2.08$ | $46.83 \pm 2.53$ | $43.63 \pm 1.56$ | $47.36 \pm 0.90$ |

(b) KiTS High Label Regime

| Dataset | KiTS | | | | | |
|---|---|---|---|---|---|---|
| Setting | QSx1/2 | | QSx1 | | QSx2 | |
| Metric | AUBC | Final Dice | AUBC | Final Dice | AUBC | Final Dice |
| Query Method | | | | | | |
| BALD | $62.85 \pm 0.60$ | $68.61 \pm 0.58$ | $61.95 \pm 1.22$ | $67.57 \pm 1.72$ | $61.54 \pm 0.59$ | $68.10 \pm 0.36$ |
| PowerBALD | $60.94 \pm 0.57$ | $66.00 \pm 0.58$ | $60.74 \pm 0.49$ | $65.04 \pm 0.81$ | $59.54 \pm 0.75$ | $64.14 \pm 1.23$ |
| SoftrankBALD | $61.29 \pm 1.04$ | $66.78 \pm 1.17$ | $61.46 \pm 0.90$ | $67.00 \pm 0.97$ | $59.91 \pm 0.54$ | $66.02 \pm 0.51$ |
| Predictive Entropy | $63.65 \pm 0.31$ | $69.04 \pm 0.61$ | $63.03 \pm 0.70$ | $68.74 \pm 0.65$ | $62.10 \pm 0.27$ | $68.12 \pm 0.80$ |
| PowerPE | $59.84 \pm 0.78$ | $63.48 \pm 0.81$ | $59.57 \pm 0.69$ | $63.62 \pm 1.19$ | $60.16 \pm 0.84$ | $64.44 \pm 1.04$ |
| Random | $53.47 \pm 1.01$ | $54.45 \pm 1.11$ | $53.65 \pm 0.64$ | $55.12 \pm 1.27$ | $54.09 \pm 0.89$ | $55.44 \pm 1.79$ |
| Random 33% FG | $54.30 \pm 1.35$ | $56.19 \pm 2.66$ | $54.80 \pm 0.83$ | $56.79 \pm 1.02$ | $54.28 \pm 1.72$ | $56.66 \pm 1.33$ |
| Random 66% FG | $53.56 \pm 1.11$ | $56.20 \pm 1.18$ | $53.92 \pm 1.23$ | $55.90 \pm 0.84$ | $53.96 \pm 0.74$ | $56.60 \pm 1.79$ |

## G.2 Training Length Ablation

**TLDR:**
- Longer training leads to better queries for AL QMs.
- Gains of using AL persist from shorter trained models to longer trained models .
- Uncertainty based QMs are more likely to oupterform Random strategies when training longer.
- It is possible to reduce AL compute cost by using shorter training for AL with a final longer training as performance gains over Random can translate.

To assess the impact of the training length on AL QMs we conduct ablation studies using the same setup as in our main study whilst varying the training length. Concretely we evaluate the following three settings of training the model for 500 epochs (500 Epochs), training the model for 200 epochs as in our main study (200 Epochs) and training the models for 500 epochs but using the query trajectories from the models trained with 200 epochs (Precomputed). This design allows us to investigate the effect of longer training while also separating the effects of extended training from its influence on query selections.

The Precomputed experiments are particularly useful in distinguishing whether performance differences arise from the query selection process itself or from the increased training duration.

We performed the experiments on the KiTS and AMOS dataset as ACDC and Hippocampus did not show improvements in Mean Dice when training for more than 200 Epochs on the entire dataset. Our focus is especially on the Medium and High Label Regimes as longer training typically mostly leads to improvements for larger datasets.

By comparing these experimental conditions, we aim to answer the following two questions regarding the relationship between query effectiveness, model training duration, and overall segmentation performance:

**Q1: Does longer training lead to better queries?** We investigate this by comparing the AUBC and Final Dice for all AL QMs of the Precomputed and 500 Epochs settings by computing their differences and testing for statistical significance with a t-test in table 11. We observe that the performance metrics for

Table 11: **Does an increased training length of the model lead to better queries?**
ΔMetric= Metric(500Epochs) - Metric(Precomputed) for the Training Length Ablation with all models trained for 500 epochs. Larger values show that the queries when training the model for longer are better than from a shorter trained model. Significance comparison performed with a two-sided t-test using a significance level $\alpha = 0.1$.
Colorscheme: ■ Significant & positive difference, ■ positive difference, ■ negative difference, ■ significant & negative difference

(a) ΔAUBC

| Dataset | AMOS | | KiTS | |
|---|---|---|---|---|
| Label Regime | Medium | High | Medium | High |
| Query Method | | | | |
| BALD | 0.98 | 0.87 | 1.73 | 1.64 |
| PowerBALD | 0.94 | 0.52 | 2.30 | 1.38 |
| SoftrankBALD | 1.00 | 0.32 | 1.41 | 1.04 |
| Predictive Entropy | 0.15 | 0.70 | 0.58 | 0.88 |
| PowerPE | 0.16 | 0.46 | 1.89 | 1.71 |

(b) Δ Final Dice

| Dataset | AMOS | | KiTS | |
|---|---|---|---|---|
| Label Regime | Medium | High | Medium | High |
| Query Method | | | | |
| BALD | 1.66 | 1.12 | 2.75 | 2.04 |
| PowerBALD | 1.45 | 0.57 | 3.92 | 1.68 |
| SoftrankBALD | 1.43 | 0.45 | 1.49 | 0.84 |
| Predictive Entropy | 1.76 | 0.76 | 2.06 | 0.44 |
| PowerPE | 0.20 | 0.55 | 2.18 | 2.30 |

the 500 Epochs models are higher in all cases than for the Precomputed models and with the exception of Predictive Entropy at least in 3 out of 4 settings statistically significant. This indicates that when performance increases with longer training uncertainty based QMs query data more effectively leading to performance improvements even when correcting for performance differences arising from training length.

**Ranking based analysis** To evaluate how each of our three training settings influences the ranking of our annotation strategies we perform a Kendall's $\tau$ (Kendall, 1948) correlation test for the AUBC and Final Dice on each Label Regime and dataset between two settings, the results are shown in table 12. We deem a ranking as stable when it is positively correlated and significant and will not discuss it except for a change where AL QMs outperform Random strategies where they previously did not or the other way around.

**Q2: Do gains obtained by using AL persist when training on the queried dataset for longer?**
Generally, the method rankings between 200 Epochs and Precomputed are stable in 3 out of 4 cases, as shown in table 12, with the exception of the AMOS High-Label Regime. We observe on the KiTS dataset that the rankings are stable and the trend that AL outperforms Random and Random FG strategies for both settings. Generally the performance gains of using AL persist from 200 Epochs to Precomputed but decrease in absolute value for the longer trainings on identical queries. This indicates that the results of our ranking for models trained with 200 epochs are likely to hold also for longer trained models on KiTS.

On the AMOS dataset the ranking is stable for the Medium but not for the High Label Regime. Generally observable is a large jump in performance for the models with the AL QMs from 200 Epochs to Precomputed (larger than for Random FG strategies) which we trace back to the Dice score of specific classes that are hard for the models to learn jumping from 0 to 0.5 for the longer training (see fig. 13). For 200 Epochs, Random 33% and 66% FG do not exhibit this behavior of individual classes having a Dice score of 0, presumably because they sample more data from these classes (see appendix F.1). On the Medium-Label Regime, PE and BALD have a strong increase in the AUBC, leading them to outperform Random for Precomputed, which they did not do for 200 Epochs, but otherwise, no big changes in ranking. On the High-Label Regime, for the AUBC Predictive Entropy and BALD increase from being outperformed by Random to outperforming Random with longer training and the Predictive Entropy and Softrank BALD outperform Random 33% FG which they did not for shorter training (table 13b). So, generally, longer training is beneficial for the AL QMs even when the queries are not performed with longer trained models.

Concluding, the gains obtained with AL QMs over Random strategies seem to translate from shorter trained to longer trained models for a shorter time and the performance losses seem to decrease.

**Q3: How does training length influence the ranking of strategies?** For this question the ranking differences between 500 Epochs and 200 Epochs and 500 Epochs and Precomputed from table 12 are evaluated.

Generally, the rankings between 500 Epochs and Precomputed showed higher correlation and were more stable than between 500 Epochs and 200 Epochs, being again robust in 3 out of 4 cases for both AUBC and Final Dice, with the exception of AMOS on the High-Label Regime.

For the KiTS dataset there are no changes with respect to the rankings of AL QMs and Random strategies on all Label Regimes and Metrics. The only unstable ranking appears on KiTS Medium for the AUBC comparing the 200 and 500 Epoch Setting, which is mostly due to the inter AL QMs ranking changing with Random and Random FG strategies occupying the worst three ranks in terms of AUBC (table 13c).

Meanwhile, for the AMOS dataset, the trend is that AL QMs perform better for longer training, which is reasonable as the models guiding the query selection are much better fitted onto the dataset. Most noteworthy for the 500 Epoch setting in the High-Label Regime Predictive and BALD are the only QMs to outperform Random 66% FG in terms of Final Dice which leads to large ranking differences between 500 Epochs and 200 Epochs (which is the only negative correlation) as well as Precomputed (table 13a). On the Medium-Label Regime, a similar trend can be observed, though not as pronounced, as only Random 33% FG becomes outperformed in terms of Final Dice in the 500 Epochs Setting (table 13b).

In conclusion, the overall results of the main study with 200 epochs extend to a large degree to 500 epoch settings, indicating that they also should hold for longer training lengths. On the AMOS dataset, this is, however, not the case, as apparently the short training of 200 epochs leads to a systematic disadvantage for the uncertainty-based QMs against the Foreground Aware Random strategies. An optimal AL QM should, however, be able to work under a variety of training settings.

**Q4: Can the compute cost of AL by reduced using shorter trainings and a final long training?** As training is a significant cost factor, this question asks whether we can reduce the training cost while still keeping the gains of AL over Random Strategies? Recalling the Analysis from Q2, it seems that in the scenarios where we obtain large gains from utilizing AL, they should persist while potential performance losses should reduce for the final long training.

However, in Q1 we showed that significant performance differences arise from queries of shorter to longer trained models even when accounting for performance differences due to training length.

In Q3 we observed on AMOS that these differences in query quality can cause the difference between a performance increase over Foreground Aware Random with queries from longer trained models to a performance loss compared to Foreground Aware Random.

For the KiTS dataset, we observed that even though ranking differences among AL methods appeared, the general trend of performance benefits over Random strategies was persistent.

Given this evidence, we suspect that it is likely feasible to perform AL experiments with shorter training. However, one must make sure, by means of validation, that the shorter trained models approximate the task "well enough" when compared to longer trained models.

Table 12: **How does the training length influence method ranking?** Kendall's $\tau$ correlation coefficients comparing rankings under different training setups on the AMOS and KiTS for the Medium- and High-Label Regime. Larger values mean rankings are consistent across experiments.
Colorscheme: ■ Significant & positive correlation, ■ positive correlation, ■ negative correlation, ■ significant & negative correlation

(a) Ranking Correlation AUBC         (b) Ranking Correlation Final Dice

| Dataset | AMOS | | KiTS | | Dataset | AMOS | | KiTS | |
| Label Regime | Medium | High | Medium | High | Label Regime | Medium | High | Medium | High |
| Setting | | | | | Setting | | | | |
|---|---|---|---|---|---|---|---|---|---|
| Precomputed & 500 Epochs | 0.810 | 0.333 | 0.714 | 0.810 | Precomputed & 500 Epochs | 0.619 | 0.524 | 0.810 | 0.714 |
| 200 Epochs & 500 Epochs | 0.810 | -0.143 | 0.524 | 0.905 | 200 Epochs & 500 Epochs | 0.524 | -0.143 | 1.000 | 0.810 |
| 200 Epochs & Precomputed | 0.929 | 0.500 | 0.857 | 0.857 | 200 Epochs & Precomputed | 1.000 | 0.500 | 0.857 | 0.929 |

**Training Length Ablation Detailed Results**

We show detailed results for the training length ablation focusing on the ranking in table 13.

**AMOS Training length**   We observe an especially strong performance increase for longer trained models on AMOS across all AL methods and Random compared to Random 66%FG, which is discussed in appendix F.1. We observe that the longer training leads to substantial performance improvements on classes 11 & 12.

Table 13: Fine-Grained Results for the training length ablation. AUBC and Final Dice are reported with a factor (×100) for improved readability. Colors indicate the ranking, darker colors correspond to worse rankings.

(a) AMOS Medium Label Regime

| Dataset | AMOS | | | | | |
|---|---|---|---|---|---|---|
| Setting | 200 Epochs | | Precomputed | | 500 Epochs | |
| Metric | AUBC | Final Dice | AUBC | Final Dice | AUBC | Final Dice |
| Query Method | | | | | | |
| BALD | 52.56 ± 2.74 | 59.26 ± 2.73 | 74.79 ± 1.97 | 80.01 ± 2.07 | 75.76 ± 1.20 | 81.67 ± 2.40 |
| PowerBALD | 66.11 ± 1.47 | 73.02 ± 2.01 | 78.93 ± 0.58 | 82.51 ± 0.42 | 79.87 ± 0.33 | 83.96 ± 0.41 |
| SoftrankBALD | 60.01 ± 0.69 | 66.72 ± 0.65 | 78.04 ± 0.34 | 82.12 ± 0.97 | 79.04 ± 0.29 | 83.55 ± 0.08 |
| Predictive Entropy | 56.30 ± 1.78 | 62.07 ± 1.39 | 77.06 ± 0.61 | 81.64 ± 0.50 | 77.21 ± 0.53 | 83.40 ± 0.41 |
| PowerPE | 66.74 ± 2.80 | 73.68 ± 0.92 | 79.11 ± 0.30 | 83.15 ± 0.45 | 79.27 ± 0.36 | 83.35 ± 0.21 |
| Random | 54.65 ± 2.82 | 56.22 ± 4.61 | 72.81 ± 1.31 | 75.46 ± 0.94 | 72.81 ± 1.31 | 75.46 ± 0.94 |
| Random 33% FG | 71.78 ± 1.16 | 78.60 ± 0.37 | 79.63 ± 0.53 | 83.70 ± 0.37 | 79.63 ± 0.53 | 83.70 ± 0.37 |
| Random 66% FG | 74.87 ± 0.64 | 80.72 ± 0.54 | 81.31 ± 0.39 | 84.94 ± 0.46 | 81.31 ± 0.39 | 84.94 ± 0.46 |

(b) AMOS High Label Regime

| Dataset | AMOS | | | | | |
|---|---|---|---|---|---|---|
| Setting | 200 Epochs | | Precomputed | | 500 Epochs | |
| Metric | AUBC | Final Dice | AUBC | Final Dice | AUBC | Final Dice |
| Query Method | | | | | | |
| BALD | 69.38 ± 0.70 | 74.95 ± 2.38 | 83.50 ± 0.17 | 86.06 ± 0.09 | 84.37 ± 0.10 | 87.18 ± 0.07 |
| PowerBALD | 77.86 ± 0.14 | 80.48 ± 0.48 | 83.98 ± 0.29 | 85.78 ± 0.06 | 84.50 ± 0.16 | 86.35 ± 0.04 |
| SoftrankBALD | 75.29 ± 1.46 | 81.23 ± 1.18 | 84.28 ± 0.11 | 86.39 ± 0.10 | 84.60 ± 0.26 | 86.83 ± 0.16 |
| Predictive Entropy | 71.27 ± 1.52 | 80.79 ± 2.07 | 84.00 ± 0.14 | 86.77 ± 0.13 | 84.70 ± 0.03 | 87.52 ± 0.08 |
| PowerPE | 77.92 ± 0.29 | 80.52 ± 0.16 | 83.86 ± 0.16 | 85.69 ± 0.19 | 84.32 ± 0.32 | 86.23 ± 0.19 |
| Random | 73.82 ± 0.50 | 75.48 ± 0.37 | 81.31 ± 0.41 | 82.73 ± 0.06 | 81.31 ± 0.41 | 82.73 ± 0.06 |
| Random 33% FG | 79.53 ± 0.38 | 82.68 ± 0.19 | 84.30 ± 0.13 | 86.28 ± 0.14 | 84.30 ± 0.13 | 86.28 ± 0.14 |
| Random 66% FG | 80.98 ± 0.19 | 83.81 ± 0.32 | 85.06 ± 0.10 | 86.98 ± 0.13 | 85.06 ± 0.10 | 86.98 ± 0.13 |

(c) KiTS Medium Label Regime

| Dataset | KiTS | | | | | |
|---|---|---|---|---|---|---|
| Setting | 200 Epochs | | Precomputed | | 500 Epochs | |
| Metric | AUBC | Final Dice | AUBC | Final Dice | AUBC | Final Dice |
| Query Method | | | | | | |
| BALD | 55.06 ± 1.20 | 61.97 ± 1.49 | 61.96 ± 0.66 | 67.84 ± 0.71 | 63.69 ± 0.96 | 70.60 ± 0.55 |
| PowerBALD | 54.53 ± 1.40 | 59.51 ± 1.15 | 61.76 ± 1.07 | 65.86 ± 1.17 | 64.06 ± 0.95 | 69.78 ± 1.38 |
| SoftrankBALD | 54.83 ± 1.79 | 61.44 ± 2.02 | 62.44 ± 1.22 | 68.32 ± 0.38 | 63.85 ± 1.01 | 69.80 ± 0.50 |
| Predictive Entropy | 57.42 ± 0.54 | 65.39 ± 0.51 | 63.79 ± 0.65 | 69.77 ± 1.10 | 64.37 ± 0.33 | 71.83 ± 0.37 |
| PowerPE | 54.76 ± 1.10 | 58.67 ± 1.53 | 62.22 ± 0.82 | 66.53 ± 0.39 | 64.11 ± 0.80 | 68.71 ± 0.89 |
| Random | 47.82 ± 1.84 | 48.41 ± 1.99 | 55.35 ± 1.22 | 56.68 ± 0.91 | 55.35 ± 1.22 | 56.68 ± 0.91 |
| Random 33% FG | 51.50 ± 1.97 | 54.08 ± 2.76 | 59.33 ± 2.33 | 62.56 ± 3.22 | 59.33 ± 2.33 | 62.56 ± 3.22 |
| Random 66% FG | 50.78 ± 0.97 | 51.67 ± 2.31 | 58.43 ± 1.08 | 61.27 ± 1.72 | 58.43 ± 1.08 | 61.27 ± 1.72 |

(d) KiTS High Label Regime

| Dataset | KiTS | | | | | |
|---|---|---|---|---|---|---|
| Setting | 200 Epochs | | Precomputed | | 500 Epochs | |
| Metric | AUBC | Final Dice | AUBC | Final Dice | AUBC | Final Dice |
| Query Method | | | | | | |
| BALD | 62.53 ± 0.84 | 67.57 ± 1.72 | 68.83 ± 0.94 | 72.47 ± 0.46 | 70.47 ± 0.47 | 74.51 ± 0.50 |
| PowerBALD | 61.24 ± 0.57 | 65.04 ± 0.81 | 68.44 ± 0.47 | 71.47 ± 0.54 | 69.82 ± 0.50 | 73.15 ± 0.49 |
| SoftrankBALD | 62.49 ± 0.74 | 67.00 ± 0.97 | 69.13 ± 0.23 | 73.44 ± 0.49 | 70.17 ± 0.62 | 74.28 ± 0.57 |
| Predictive Entropy | 64.00 ± 0.15 | 68.74 ± 0.65 | 69.77 ± 0.46 | 73.78 ± 0.87 | 70.65 ± 0.31 | 74.21 ± 0.14 |
| PowerPE | 60.66 ± 0.66 | 63.62 ± 1.19 | 68.19 ± 0.33 | 70.48 ± 0.67 | 69.91 ± 0.25 | 72.78 ± 0.84 |
| Random | 53.80 ± 0.68 | 55.12 ± 1.27 | 63.30 ± 1.11 | 65.36 ± 0.94 | 63.30 ± 1.11 | 65.36 ± 0.94 |
| Random 33% FG | 55.30 ± 1.26 | 56.79 ± 1.02 | 65.27 ± 1.59 | 68.81 ± 1.08 | 65.27 ± 1.59 | 68.81 ± 1.08 |
| Random 66% FG | 53.73 ± 1.78 | 55.90 ± 0.84 | 64.63 ± 2.52 | 68.03 ± 0.46 | 64.63 ± 2.52 | 68.03 ± 0.46 |

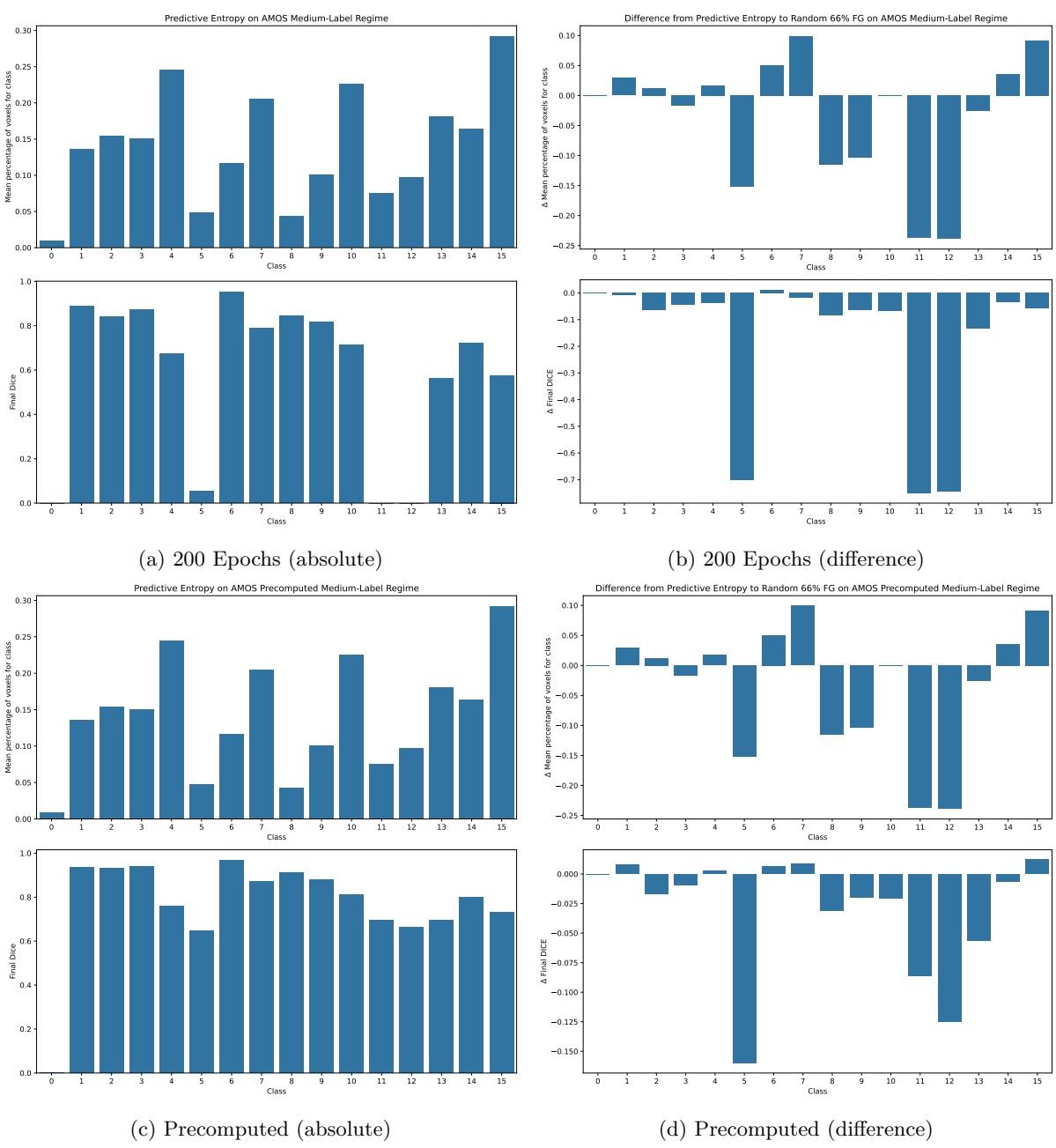

Figure 13: Visualization of absolute values and the difference of the percentage of voxels for all classes alongside Final Dice performance on the AMOS Medium-Label Regime from Predictive Entropy. It shows that less data containing classes 11 & 12 (right & left adrenal gland) is queried by Predictive Entropy (also Random) (5% less of the overall voxels of that class), which is strongly correlated with the Final Dice for these classes being 0 for the 200 Epochs results (a) whereas it is at ≈ 0.7 for the Precomputed models on the exact same data just trained for 500 epochs (c). This substantially reduces the performance gap compared to Random 66% FG as can be seen in (b &d).

### G.3 Noise strength in Noisy QMs Ablation

> **TLDR:**
> - Given more data the optimal noise for a Noisy QM decreases.
> - Preemptively optimizing the noise parameter remains an open research question.

Our aim is to understand the influence of the noise strength for the Noisy QMs (PowerBALD, SoftrankBALD, PowerPE) in the experimental setup of our main study by an exemplary systematic ablation for PowerBALD.

For PowerBALD $\beta$ is the parameter which perturbs the ranking of the BALD scores $s_{\mathrm{BALD}}$ on a logarithmic scale with Gumbel noise as follows:

$$s_{\mathrm{PowerBALD}} = \log(s_{\mathrm{BALD}}) + \epsilon \tag{2}$$

where $\epsilon \sim \mathrm{Gumbel}(0, \beta^{-1})$. The standard deviation of $\epsilon$ is proportional to $\beta^{-1}$, meaning that smaller values of $\beta$ introduce greater randomness in query selection, while larger values preserve the original ranking. As $\beta \to \infty$, the ranking remains unchanged after adding noise, whereas as $\beta \to 0$, query selection becomes entirely random. By varying $\beta$, we can control the balance between exploration and exploitation in the selection process. It has already been noted by Kirsch et al. (2023) that in later stages of training the correlation of queries for Greedy Methods due to top-k sampling decreases. We suspect therefore that the optimal choice of $\beta$ will differ across our experiments leaving room for method improvement from the standard setting $\beta = 1$ (Kirsch et al. (2023)) we used in our main study.

To assess the influence of data distribution and label regime we perform experiments on the ACDC, AMOS and KiTS dataset for the Low-, Medium- and High- Label Regime whilst varying the parameter $\beta = \{1, 5, 10, 20, 40, \infty\}$ with $\beta = \infty$ being identical to BALD. Generally we only analyze larger values of $\beta$ as our implementation adds Gumbel noise on the mean aggregated scores leading to the standard deviation of aggregated values naturally being smaller than for singular values.

Using this experimental setting with the results shown in fig. 14 we aim to answer the following two questions (Q1-Q2):

**Q1: How is optimal $\beta$ influenced by amount of data?** When evaluating the results on each dataset separately, we observe that the optimal parameter of $\beta$ with regard to the AUBC and Final Dice generally increases from Low to Medium to High Label Regime. This aligns with our broader observations that noise-perturbed QMs generally outperform their greedy counterparts in the early stages of AL but are often overtaken in later loops as training progresses. With regard to foreground efficiency, we observe a steady decrease for higher values of $\beta$, converging toward the FG-Eff of BALD across all label regimes, indicating that the reduction in queried foreground voxels is greater than the difference in performance.

Generally, the optimal $\beta$ is therefore strongly correlated with the amount of data and increases with more data.

**Q2: How to select optimal $\beta$ preemptively** Despite the observation from Q1, we do not identify a single, universally optimal value range of $\beta$ across all datasets, as they differ greatly across the different datasets. On AMOS, optimal values range from 0 to 5, with a sharp decline in performance for higher values. In ACDC, the optimal range shifts to 5–40, while in KiTS, it spans 1–40. This indicates that dataset properties play an important role in the optimal selection of this parameter, such as – but not limited to – the number of classes and their diversity. Furthermore, we hypothesize the following design decisions of the AL Pipeline to be important: Training length, Query Method (uncertainties and aggregation function) and query patch size.

Based on this, we conclude that setting this value preemptively remains an open question.

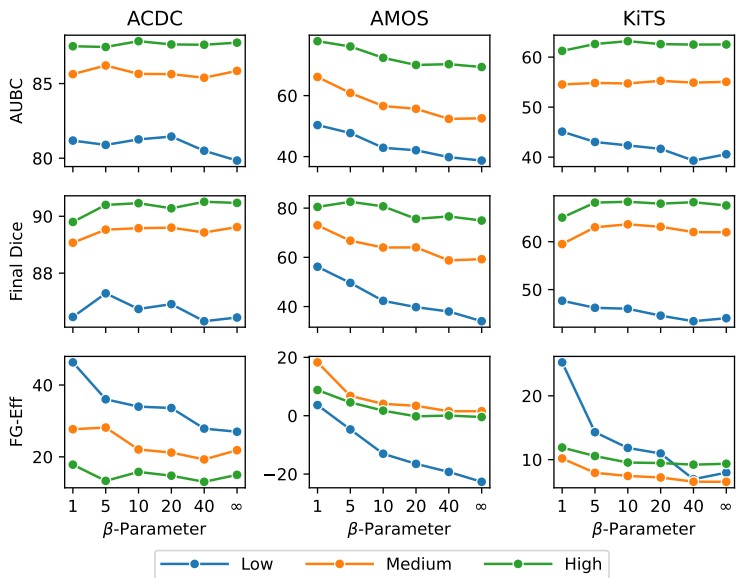

Figure 14: The $\beta$-parameter for PowerBALD plotted against AUBC, Final DICE and FG Eff. for the Low-, Medium- and High-Label Regimes. $\beta$-values leading to the best AUBC and Final DICE tends to increase for higher budgets. This indicates that for higher budgets less ranking perturbations perform better. At the same time the FG Eff. decreases which shows that the reduction in perturbation means that more FG is queried.

**Noise Strength Detailed Results**

Table 14: Ablating the influence of the noise parameter for PowerBALD. AUBC and Final Dice are reported with a factor (×100) for improved readability. The values leading to the highest AUBC and Final Mean Dice increase for larger budgets across all datasets.

(a) ACDC

| Dataset
Label Regime
Metric
Query Method | ACDC
Low
AUBC | Final Dice | FG-Eff | Medium
AUBC | Final Dice | FG-Eff | High
AUBC | Final Dice | FG-Eff |
|---|---|---|---|---|---|---|---|---|---|
| PowerBALD (b=1) | 81.18 ± 0.58 | 86.46 ± 0.55 | 46.30 ± 13.10 | 85.63 ± 0.37 | 89.07 ± 0.21 | 27.69 ± 3.96 | 87.50 ± 0.44 | 89.80 ± 0.17 | 17.83 ± 1.82 |
| PowerBALD (b=5) | 80.89 ± 1.11 | 87.29 ± 0.34 | 36.03 ± 6.27 | 86.21 ± 0.39 | 89.53 ± 0.35 | 28.16 ± 6.55 | 87.45 ± 0.54 | 90.40 ± 0.34 | 13.31 ± 1.83 |
| PowerBALD (b=10) | 81.26 ± 1.25 | 86.74 ± 0.38 | 33.98 ± 5.87 | 85.65 ± 0.65 | 89.58 ± 0.23 | 22.05 ± 3.47 | 87.84 ± 0.46 | 90.46 ± 0.20 | 15.80 ± 1.85 |
| PowerBALD (b=20) | 81.45 ± 1.11 | 86.91 ± 0.79 | 33.60 ± 8.86 | 85.63 ± 0.31 | 89.60 ± 0.14 | 21.19 ± 3.06 | 87.62 ± 0.46 | 90.28 ± 0.20 | 14.75 ± 1.35 |
| PowerBALD (b=40) | 80.50 ± 1.36 | 86.31 ± 0.34 | 27.86 ± 4.46 | 85.39 ± 0.75 | 89.43 ± 0.26 | 19.28 ± 3.46 | 87.60 ± 0.31 | 90.51 ± 0.19 | 13.05 ± 0.64 |
| PowerBALD (b=∞) | 79.84 ± 0.59 | 86.44 ± 0.96 | 26.99 ± 3.11 | 85.85 ± 0.45 | 89.62 ± 0.15 | 21.85 ± 4.16 | 87.74 ± 0.38 | 90.47 ± 0.18 | 14.99 ± 1.14 |

(b) AMOS

| Dataset
Label Regime
Metric
Query Method | AMOS
Low
AUBC | Final Dice | FG-Eff | Medium
AUBC | Final Dice | FG-Eff | High
AUBC | Final Dice | FG-Eff |
|---|---|---|---|---|---|---|---|---|---|
| PowerBALD (b=1) | 50.34 ± 3.00 | 56.18 ± 1.24 | 3.65 ± 14.56 | 66.11 ± 1.47 | 73.02 ± 2.01 | 18.28 ± 0.44 | 77.86 ± 0.14 | 80.48 ± 0.48 | 8.80 ± 0.08 |
| PowerBALD (b=5) | 47.72 ± 1.70 | 49.61 ± 1.31 | -4.72 ± 3.70 | 60.87 ± 0.97 | 66.76 ± 1.09 | 6.78 ± 0.13 | 76.11 ± 1.65 | 82.58 ± 0.60 | 4.56 ± 0.57 |
| PowerBALD (b=10) | 42.89 ± 0.29 | 42.32 ± 3.03 | -13.04 ± 4.86 | 56.58 ± 2.17 | 63.99 ± 0.66 | 4.06 ± 0.17 | 72.39 ± 1.69 | 80.71 ± 2.03 | 1.74 ± 0.49 |
| PowerBALD (b=20) | 42.08 ± 2.58 | 39.77 ± 2.02 | -16.51 ± 4.56 | 55.68 ± 1.62 | 64.04 ± 2.21 | 3.40 ± 0.18 | 70.05 ± 1.00 | 75.65 ± 4.15 | -0.22 ± 0.27 |
| PowerBALD (b=40) | 39.82 ± 3.50 | 37.98 ± 6.99 | -19.27 ± 12.73 | 52.37 ± 1.83 | 58.76 ± 1.76 | 1.54 ± 0.17 | 70.31 ± 1.00 | 76.62 ± 4.52 | 0.05 ± 0.30 |
| PowerBALD (b=∞) | 38.69 ± 2.34 | 34.05 ± 1.58 | -22.66 ± 8.50 | 52.56 ± 2.74 | 59.26 ± 2.73 | 1.54 ± 0.22 | 69.38 ± 0.70 | 74.95 ± 2.38 | -0.45 ± 0.20 |

(c) KiTS

| Dataset
Label Regime
Metric
Query Method | KiTS
Low
AUBC | Final Dice | FG-Eff | Medium
AUBC | Final Dice | FG-Eff | High
AUBC | Final Dice | FG-Eff |
|---|---|---|---|---|---|---|---|---|---|
| PowerBALD (b=1) | 45.10 ± 2.91 | 47.67 ± 3.63 | 25.24 ± 6.06 | 54.53 ± 1.40 | 59.51 ± 1.15 | 10.18 ± 0.41 | 61.24 ± 0.57 | 65.04 ± 0.81 | 11.89 ± 0.63 |
| PowerBALD (b=5) | 43.03 ± 3.65 | 46.20 ± 4.98 | 14.30 ± 2.54 | 54.83 ± 1.30 | 63.02 ± 1.43 | 7.93 ± 0.24 | 62.62 ± 1.09 | 68.17 ± 0.36 | 10.56 ± 0.36 |
| PowerBALD (b=10) | 42.35 ± 3.72 | 46.00 ± 3.58 | 11.83 ± 1.95 | 54.73 ± 1.70 | 63.63 ± 1.06 | 7.44 ± 0.23 | 63.19 ± 0.38 | 68.34 ± 0.57 | 9.54 ± 0.15 |
| PowerBALD (b=20) | 41.66 ± 4.43 | 44.56 ± 6.96 | 10.96 ± 2.12 | 55.26 ± 1.63 | 63.12 ± 0.22 | 7.20 ± 0.17 | 62.60 ± 0.57 | 67.95 ± 0.70 | 9.46 ± 0.28 |
| PowerBALD (b=40) | 39.31 ± 4.50 | 43.40 ± 4.69 | 6.93 ± 1.93 | 54.90 ± 1.20 | 62.01 ± 1.46 | 6.54 ± 0.14 | 62.51 ± 0.63 | 68.25 ± 0.61 | 9.21 ± 0.39 |
| PowerBALD (b=∞) | 40.58 ± 2.75 | 44.03 ± 3.18 | 7.96 ± 0.82 | 55.06 ± 1.20 | 61.97 ± 1.49 | 6.52 ± 0.14 | 62.53 ± 0.84 | 67.57 ± 1.72 | 9.35 ± 0.46 |

### G.4 Query Patch Size Ablation

> **TLDR:**
> 1. AL methods are resilient with regard to smaller Query Patch Sizes, even though absolute annotated voxels are greatly reduced.
> 2. The Query Patch Size can induce substantial changes in method rankings, making it important to systematically evaluate the benefits of AL methods.

Here we aim to understand the influence of the query patch size parameter on our AL experiments.

The query patch size is a hyperparameter of our AL pipeline, setting our work apart as we are the first to allow completely free 3D Patch selection, differentiating our experimental setup from related work, which uses either 2D slice or 3D image queries.

To evaluate its influence, we repeat our entire main study with all four datasets with the respective query patch size halved along each axis whilst keeping the number of patches for each label regime identical. We motivate these design decisions as we are interested in seeing whether a more fine-grained selection of areas helps AL methods and the annotation effort for smaller patches does not necessarily decrease linearly with the voxel size.

As the changes with regard to the query patch size make experiments across Label Regimes incomparable, we compare instead across the dataset mean ranking and the overall mean ranking. To do so, we first perform bootstrap sampling to obtain a mean method ranking for each label regime of each dataset, which we then aggregate to the dataset and overall level. These mean aggregated rankings are then compared using Kendall's $\tau$ (Kendall, 1948) and a significance test; the results are shown in table 15, and the mean ranking values are shown in appendix G.4.

With this setup, we evaluate the following questions:

**Q1: Does the Query Patch Size influence AL Performance?** When comparing the Average Mean rank for Patch$\times 1$ and Patch$\times \frac{1}{2}$, it appears that AL has improved Performance compared to Random strategies (appendix G.4), especially with regard to the AUBC. Even though the absolute annotated voxels are reduced by a factor of 16 for Patch$\times \frac{1}{2}$, the trend indicates that the AL methods perform better compared to the Foreground Aware Random strategies on all datasets with the exception of AMOS where for both Patch Sizes the Foreground Aware Random strategies perform best.

When comparing the mean PPMs, we observe that (fig. 15) similar trends also with the Predictive Entropy being the method with the best win/lose-ratio against Random 66% FG.

In conclusion, we observe that AL methods are surprisingly resilient with regard to the Query Patch Size.

**Q2: How does the Query Patch Size influence the ranking?** The mean rankings of the Final Dice across all datasets are stable, with Predictive Entropy being the best performing method, followed by most other AL methods, with Random FG 66% mixed in between, followed by Random FG 33%, and finally Random as the worst performing. For the AUBC we observe a change in trend for the smaller Query Patch Size where all Noisy QMs are outperformed by their Greedy counterparts. We hypothesize that there are two reasons for this behavior: the reduced amount of training data and/or the higher chance of highly similar patterns in the dataset, resulting in high uncertainty values.

On the dataset level, the trend is that for AMOS and KiTS the rankings across Query Patch Sizes are stable, whereas they are less so for Hippocampus and almost completely unstable for ACDC. On ACDC BALD and its derivatives perform better for the smaller than the larger Query Patch Size in terms of AUBC and Final Dice (table 18). On Hippocampus BALD and SoftrankBALD also perform better for the smaller than the larger Query Patch Size in terms of AUBC and Final Dice, PowerBALD less so presumably due to the noise parameter being too large (table 20).

We conclude that different Query Patch Sizes can lead to substantial differences in the ranking of QMs.

|            | ACDC  | AMOS  | Hippocampus | KiTS  | Average |
|------------|-------|-------|-------------|-------|---------|
| AUBC       | 0.357 | 0.786 | 0.571       | 0.793 | 0.143   |
| Final Dice | 0.036 | 0.764 | 0.5         | 0.837 | 0.714   |

Table 15: **How does the Query Patch Size influence method benchmarking?** High values indicate that method rankings are consistent across different query patch sizes. Kendall's $\tau$ correlation coefficients comparing the mean rankings for all datasets and each dataset separately with different patch sizes. A two-sided test was performed with a significance level of $\alpha = 0.1$.

Colorscheme: ■ Significant & positive correlation, ■ positive correlation, ■ negative correlation, ■ significant & negative correlation

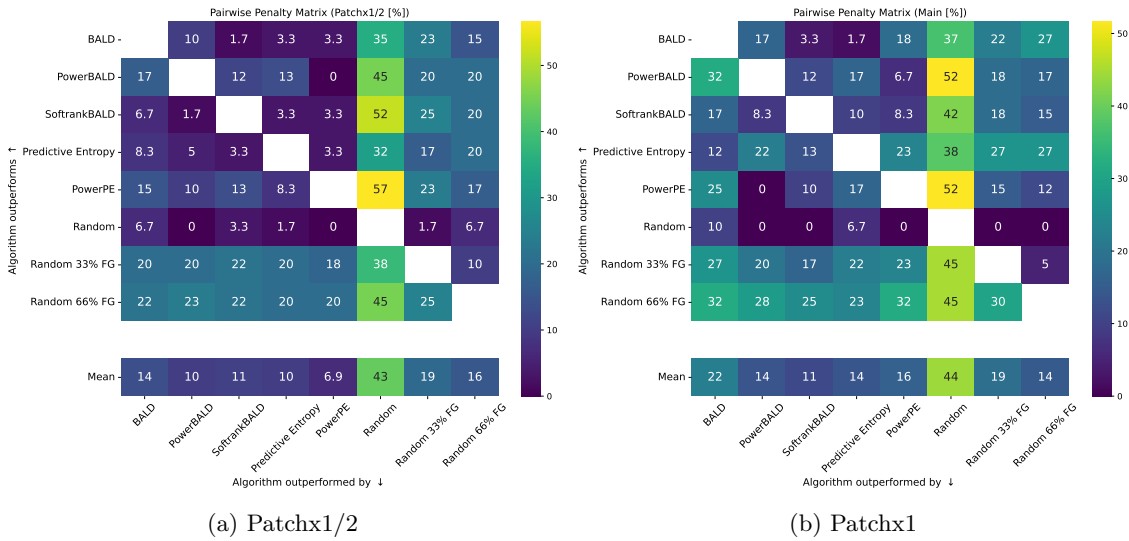

(a) Patchx1/2          (b) Patchx1

Figure 15: PPM aggregated over all Label Regimes for each dataset for the Patch Size Ablation with size Patchx1/2 and Patchx1 (Main Study).

**Query Patch Size Ablation Detailed Results**

Table 16: Fine-Grained Results for the patch ablation with setting Patch$\times\frac{1}{2}$ for each dataset. Higher values are better and colorization goes from bright (best) to dark orange(worst). Final Dice is reported with a factor ($\times 100$) for improved readability. AUBC, Final and Beta can only directly compared for each label regime on each dataset.

(a) ACDC

| Dataset
Label Regime
Metric
Query Method | ACDC | | | | | | | | |
| | | Low | | | Medium | | | High | |
| | AUBC | Final Dice | FG-Eff | AUBC | Final Dice | FG-Eff | AUBC | Final Dice | FG-Eff |
| --- | --- | --- | --- | --- | --- | --- | --- | --- | --- |
| BALD | 68.23 ± 2.31 | 77.72 ± 1.46 | 230.94 ± 445.25 | 75.80 ± 1.10 | 82.59 ± 1.24 | 149.13 ± 432.05 | 79.59 ± 1.05 | 83.99 ± 0.85 | 75.63 ± 69.85 |
| PowerBALD | 65.90 ± 5.61 | 75.24 ± 2.32 | 306.70 ± 948.85 | 77.07 ± 1.11 | 83.01 ± 1.21 | 207.23 ± 491.64 | 80.27 ± 1.39 | 84.54 ± 1.25 | 121.75 ± 194.86 |
| SoftrankBALD | 66.81 ± 3.68 | 75.98 ± 0.13 | 245.78 ± 446.20 | 76.84 ± 1.31 | 82.16 ± 0.47 | 198.50 ± 456.74 | 79.69 ± 1.03 | 84.03 ± 1.56 | 118.55 ± 183.10 |
| Predictive Entropy | 65.27 ± 2.45 | 75.79 ± 2.45 | 185.36 ± 203.07 | 74.67 ± 1.26 | 81.18 ± 1.32 | 119.08 ± 160.70 | 79.25 ± 0.95 | 83.58 ± 1.33 | 85.12 ± 103.40 |
| PowerPE | 65.70 ± 3.90 | 74.46 ± 2.28 | 301.78 ± 894.94 | 76.26 ± 2.36 | 82.16 ± 2.15 | 211.14 ± 448.40 | 79.85 ± 1.31 | 84.48 ± 1.55 | 133.05 ± 271.55 |
| Random | 59.38 ± 5.56 | 65.19 ± 4.17 | 480.79 ± 2317.97 | 70.99 ± 3.17 | 76.66 ± 1.22 | 459.67 ± 713.79 | 76.30 ± 0.80 | 79.09 ± 0.46 | 261.56 ± 531.48 |
| Random 33% FG | 67.98 ± 1.51 | 77.43 ± 0.27 | 216.59 ± 97.30 | 75.09 ± 1.78 | 81.65 ± 1.01 | 127.48 ± 64.58 | 79.55 ± 1.12 | 84.44 ± 0.32 | 88.02 ± 27.63 |
| Random 66% FG | 64.33 ± 1.17 | 73.97 ± 0.55 | 101.88 ± 47.08 | 74.69 ± 0.28 | 82.18 ± 1.52 | 71.88 ± 20.74 | 80.33 ± 0.56 | 85.88 ± 0.64 | 56.98 ± 8.47 |

Table 17: AMOS

| Dataset
Label Regime
Metric
Query Method | AMOS | | | | | | | | |
| | | Low | | | Medium | | | High | |
| | AUBC | Final Dice | FG-Eff | AUBC | Final Dice | FG-Eff | AUBC | Final Dice | FG-Eff |
| --- | --- | --- | --- | --- | --- | --- | --- | --- | --- |
| BALD | 13.98 ± 1.24 | 10.96 ± 2.19 | -149.85 ± 369.03 | 16.93 ± 2.33 | 17.85 ± 4.60 | -10.43 ± 20.83 | 30.15 ± 1.72 | 27.72 ± 0.83 | -19.51 ± 3.37 |
| PowerBALD | 14.54 ± 2.70 | 11.74 ± 2.59 | -247.20 ± 763.03 | 21.71 ± 1.48 | 25.83 ± 2.25 | 8.77 ± 16.48 | 40.14 ± 1.86 | 42.40 ± 1.47 | 8.16 ± 4.96 |
| SoftrankBALD | 13.63 ± 2.69 | 11.39 ± 1.68 | -127.00 ± 355.48 | 19.95 ± 1.55 | 23.48 ± 3.19 | -0.92 ± 8.31 | 35.13 ± 2.40 | 39.37 ± 1.87 | -3.29 ± 5.42 |
| Predictive Entropy | 13.83 ± 2.12 | 12.28 ± 1.98 | -83.38 ± 257.65 | 24.61 ± 2.34 | 27.37 ± 5.21 | 7.05 ± 2.20 | 36.63 ± 6.19 | 43.86 ± 6.88 | 1.59 ± 4.86 |
| PowerPE | 15.18 ± 2.85 | 13.00 ± 4.65 | -210.89 ± 558.96 | 23.28 ± 1.26 | 27.05 ± 2.03 | 14.76 ± 8.90 | 43.20 ± 1.78 | 47.34 ± 3.06 | 18.40 ± 3.35 |
| Random | 12.78 ± 2.02 | 8.89 ± 1.91 | -937.96 ± 5770.12 | 16.14 ± 1.62 | 16.99 ± 3.32 | -91.07 ± 168.72 | 37.56 ± 1.57 | 37.28 ± 3.44 | -4.41 ± 22.82 |
| Random 33% FG | 22.10 ± 1.18 | 24.14 ± 4.37 | 15.47 ± 104.65 | 39.32 ± 2.68 | 51.61 ± 4.21 | 103.40 ± 24.63 | 56.68 ± 1.86 | 65.54 ± 1.82 | 63.35 ± 10.78 |
| Random 66% FG | 31.10 ± 2.19 | 39.70 ± 0.34 | 135.25 ± 66.74 | 48.12 ± 0.68 | 60.25 ± 0.45 | 94.66 ± 28.61 | 62.07 ± 0.73 | 70.71 ± 0.60 | 51.43 ± 8.39 |

(a) Hippocampus

| Dataset
Label Regime
Metric
Query Method | Hippocampus | | | | | | | | |
| | | Low | | | Medium | | | High | |
| | AUBC | Final Dice | FG-Eff | AUBC | Final Dice | FG-Eff | AUBC | Final Dice | FG-Eff |
| --- | --- | --- | --- | --- | --- | --- | --- | --- | --- |
| BALD | 86.42 ± 0.47 | 87.85 ± 0.15 | 72.53 ± 176.66 | 87.64 ± 0.17 | 88.43 ± 0.19 | 15.03 ± 2.49 | 87.99 ± 0.16 | 88.76 ± 0.08 | 12.55 ± 1.38 |
| PowerBALD | 86.07 ± 0.35 | 87.45 ± 0.37 | 79.38 ± 99.29 | 87.32 ± 0.04 | 88.12 ± 0.04 | 18.16 ± 1.27 | 87.82 ± 0.04 | 88.47 ± 0.07 | 17.41 ± 1.33 |
| SoftrankBALD | 86.44 ± 0.33 | 87.66 ± 0.32 | 73.37 ± 156.28 | 87.54 ± 0.17 | 88.27 ± 0.10 | 16.64 ± 2.19 | 87.92 ± 0.07 | 88.66 ± 0.14 | 15.25 ± 1.73 |
| Predictive Entropy | 86.34 ± 0.22 | 87.69 ± 0.09 | 63.90 ± 130.04 | 87.43 ± 0.14 | 88.41 ± 0.09 | 13.37 ± 1.22 | 87.99 ± 0.14 | 88.74 ± 0.09 | 12.40 ± 1.77 |
| PowerPE | 86.21 ± 0.70 | 87.56 ± 0.51 | 84.64 ± 146.26 | 87.43 ± 0.11 | 88.29 ± 0.11 | 19.82 ± 2.64 | 87.94 ± 0.11 | 88.43 ± 0.15 | 18.24 ± 2.48 |
| Random | 85.62 ± 0.65 | 86.74 ± 0.31 | 118.84 ± 225.57 | 87.06 ± 0.21 | 87.76 ± 0.10 | 25.66 ± 5.49 | 87.58 ± 0.15 | 88.13 ± 0.15 | 26.30 ± 3.24 |
| Random 33% FG | 85.69 ± 0.56 | 87.02 ± 0.19 | 79.09 ± 126.53 | 87.26 ± 0.17 | 88.00 ± 0.07 | 17.20 ± 2.50 | 87.74 ± 0.06 | 88.31 ± 0.11 | 15.53 ± 1.30 |
| Random 66% FG | 86.24 ± 0.13 | 87.54 ± 0.15 | 57.33 ± 56.56 | 87.49 ± 0.21 | 88.27 ± 0.10 | 14.92 ± 1.15 | 87.85 ± 0.21 | 88.54 ± 0.17 | 12.51 ± 0.66 |

(b) KiTS

| Dataset
Label Regime
Metric
Query Method | KiTS | | | | | | | | |
| | | Low | | | Medium | | | High | |
| | AUBC | Final Dice | FG-Eff | AUBC | Final Dice | FG-Eff | AUBC | Final Dice | FG-Eff |
| --- | --- | --- | --- | --- | --- | --- | --- | --- | --- |
| BALD | 25.10 ± 0.55 | 31.76 ± 4.51 | 87.05 ± 97.39 | 38.56 ± 3.27 | 43.25 ± 3.79 | 30.75 ± 9.84 | 48.46 ± 1.19 | 53.50 ± 1.28 | 24.01 ± 6.81 |
| PowerBALD | 27.91 ± 1.74 | 29.39 ± 1.30 | 185.27 ± 580.70 | 41.70 ± 1.09 | 45.59 ± 1.40 | 77.67 ± 43.37 | 49.60 ± 0.95 | 54.00 ± 1.25 | 43.11 ± 17.12 |
| SoftrankBALD | 25.67 ± 2.68 | 31.47 ± 1.54 | 90.97 ± 90.60 | 41.08 ± 1.00 | 46.14 ± 1.77 | 45.78 ± 16.15 | 49.08 ± 1.12 | 54.39 ± 1.53 | 31.79 ± 10.31 |
| Predictive Entropy | 24.08 ± 1.56 | 29.07 ± 5.82 | 41.91 ± 25.47 | 40.99 ± 3.00 | 46.80 ± 3.63 | 23.66 ± 2.82 | 50.22 ± 1.42 | 55.79 ± 1.07 | 14.88 ± 1.68 |
| PowerPE | 27.96 ± 3.53 | 30.88 ± 4.84 | 208.04 ± 653.54 | 42.26 ± 0.77 | 46.55 ± 0.95 | 82.17 ± 50.00 | 49.48 ± 1.57 | 53.59 ± 1.03 | 44.22 ± 21.94 |
| Random | 22.00 ± 1.62 | 22.85 ± 2.15 | 140.75 ± 532.82 | 35.14 ± 2.00 | 37.95 ± 1.74 | 90.43 ± 136.75 | 42.73 ± 1.09 | 44.35 ± 1.60 | 47.22 ± 74.17 |
| Random 33% FG | 23.88 ± 3.43 | 28.83 ± 1.46 | 49.19 ± 31.52 | 37.88 ± 0.74 | 41.24 ± 0.88 | 17.18 ± 1.39 | 42.28 ± 1.19 | 44.19 ± 1.31 | 3.51 ± 0.15 |
| Random 66% FG | 24.43 ± 1.96 | 28.80 ± 2.90 | 29.92 ± 7.79 | 34.12 ± 1.40 | 36.52 ± 1.24 | 4.17 ± 0.26 | 40.24 ± 1.31 | 42.58 ± 0.88 | 0.69 ± 0.04 |

Table 18: ACDC Mean Ranks

| Setting | Rank Mean Dice AUBC | | Rank Mean Dice Final | |
|---|---|---|---|---|
| Query Method | Main | Patchx1-2 | Main | Patchx1-2 |
| BALD | 3.83 | 3.25 | 2.50 | 2.92 |
| PowerBALD | 3.67 | 2.67 | 4.58 | 2.92 |
| SoftrankBALD | 3.92 | 3.17 | 3.25 | 4.33 |
| Predictive Entropy | 4.50 | 6.08 | 2.17 | 5.75 |
| PowerPE | 5.58 | 3.83 | 4.42 | 4.50 |
| Random | 8.00 | 8.00 | 8.00 | 8.00 |
| Random 33% FG | 5.42 | 4.08 | 7.00 | 3.58 |
| Random 66% FG | 1.08 | 4.92 | 4.08 | 4.00 |

Table 19: AMOS Mean Ranks

| Setting | Rank Mean Dice AUBC | | Rank Mean Dice Final | |
|---|---|---|---|---|
| Query Method | Main | Patchx1-2 | Main | Patchx1-2 |
| BALD | 7.75 | 6.92 | 7.58 | 7.17 |
| PowerBALD | 3.58 | 4.50 | 4.08 | 4.92 |
| SoftrankBALD | 5.00 | 6.42 | 4.42 | 5.83 |
| Predictive Entropy | 6.92 | 4.83 | 5.42 | 4.00 |
| PowerPE | 3.42 | 3.33 | 4.08 | 3.50 |
| Random | 6.33 | 7.00 | 7.42 | 7.58 |
| Random 33% FG | 2.00 | 2.00 | 2.00 | 2.00 |
| Random 66% FG | 1.00 | 1.00 | 1.00 | 1.00 |

Table 20: Hippocampus Mean Ranks

| Setting | Rank Mean Dice AUBC | | Rank Mean Dice Final | |
|---|---|---|---|---|
| Query Method | Main | Patchx1-2 | Main | Patchx1-2 |
| BALD | 2.33 | 1.67 | 2.67 | 1.17 |
| PowerBALD | 4.75 | 5.83 | 4.83 | 5.50 |
| SoftrankBALD | 3.33 | 2.67 | 3.00 | 3.08 |
| Predictive Entropy | 1.08 | 3.00 | 1.83 | 2.25 |
| PowerPE | 6.17 | 3.92 | 5.92 | 4.58 |
| Random | 6.83 | 7.92 | 6.83 | 8.00 |
| Random 33% FG | 5.17 | 7.00 | 4.33 | 7.00 |
| Random 66% FG | 6.33 | 4.00 | 6.58 | 4.42 |

Table 21: KiTS Mean Ranks

| Setting | Rank Mean Dice AUBC | | Rank Mean Dice Final | |
|---|---|---|---|---|
| Query Method | Main | Patchx1-2 | Main | Patchx1-2 |
| BALD | 3.92 | 4.83 | 3.83 | 3.83 |
| PowerBALD | 3.50 | 1.92 | 3.58 | 3.75 |
| SoftrankBALD | 3.58 | 3.58 | 3.00 | 2.33 |
| Predictive Entropy | 2.75 | 3.58 | 2.58 | 2.67 |
| PowerPE | 3.50 | 1.67 | 3.67 | 3.08 |
| Random | 7.83 | 7.08 | 8.00 | 7.17 |
| Random 33% FG | 5.42 | 6.25 | 5.17 | 6.00 |
| Random 66% FG | 5.50 | 7.08 | 6.17 | 7.17 |

Table 22: Average Mean Ranks over all datasets

| Setting | Rank Mean Dice AUBC | | Rank Mean Dice Final | |
|---|---|---|---|---|
| Query Method | Main | Patchx1-2 | Main | Patchx1-2 |
| BALD | 4.46 | 4.17 | 4.15 | 3.77 |
| PowerBALD | 3.88 | 3.73 | 4.27 | 4.27 |
| SoftrankBALD | 3.96 | 3.96 | 3.42 | 3.90 |
| Predictive Entropy | 3.81 | 4.38 | 3.00 | 3.67 |
| PowerPE | 4.67 | 3.19 | 4.52 | 3.92 |
| Random | 7.25 | 7.50 | 7.56 | 7.69 |
| Random 33% FG | 4.50 | 4.83 | 4.62 | 4.65 |
| Random 66% FG | 3.48 | 4.25 | 4.46 | 4.15 |

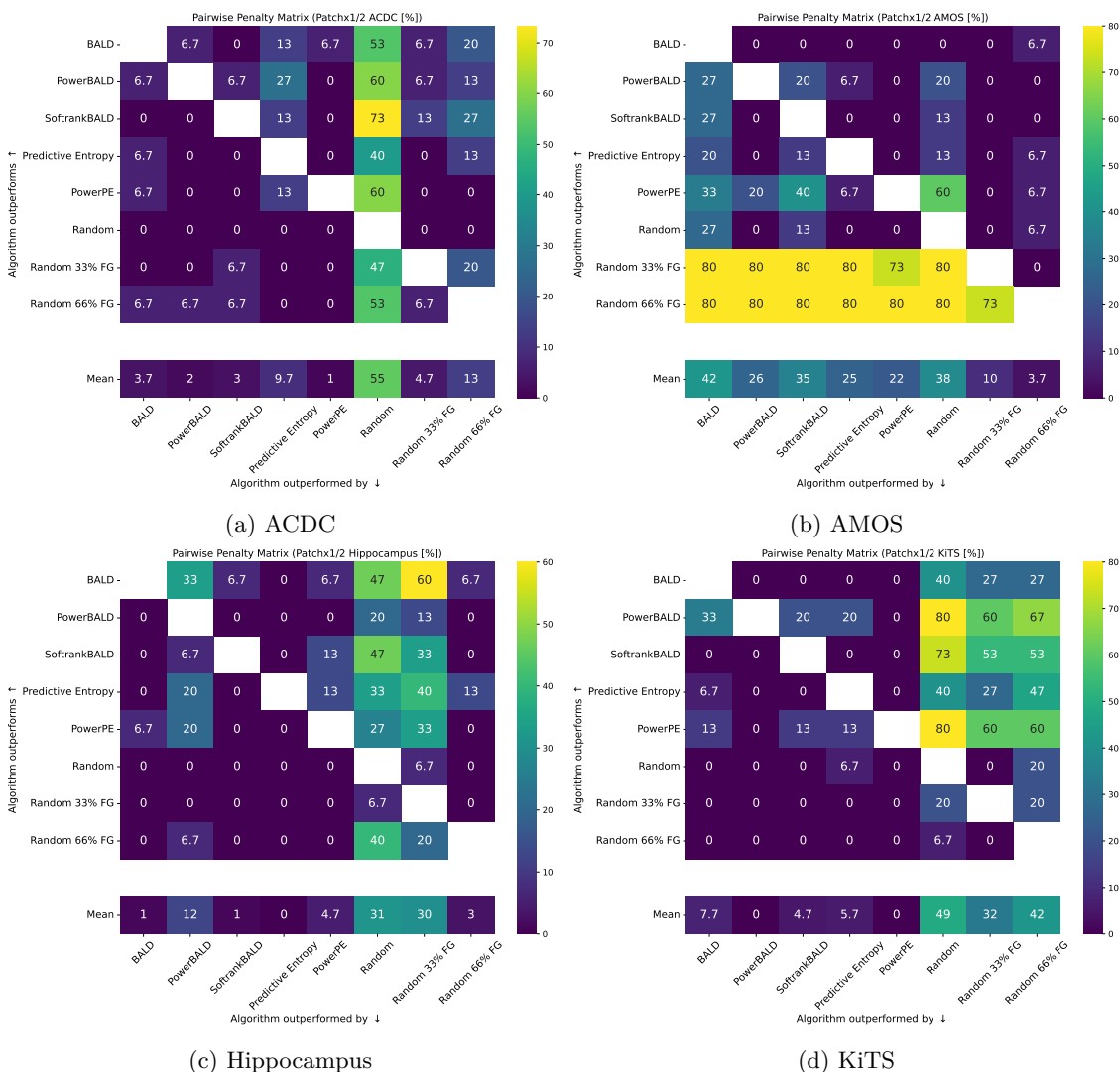

(a) ACDC

(b) AMOS

(c) Hippocampus

(d) KiTS

Figure 16: Pairwise Penalty Matrix aggregated over all Label Regimes for each dataset for the Patch Size Ablation with size Patchx1/2.

# H    Leave-One-Out Analysis of Rankings on the Main Study

We additionally analyze the results of the main study shown in appendix F by means of computing the rankings for AUBC and Final Dice in a leave-one-out fashion based on experimental seeds.

**Results**  Alternative versions of the main overview figure (shown in fig. 4) which are obtained by means of aggregating the mean rank for each scenario from the 4 leave-one-out rankings, are shown for the AUBC in fig. 17 and for the Final Dice in fig. 18.

Detailed results showing also the distribution of the four obtained rankings are shown for the AUBC in fig. 19 and for the Final Dice in fig. 20.

**Take-Away:**  General groups of ranking performance of QMs can be observed in all scenarios where certain groups of QMs are better than others. Overall, based on this analysis, little overall changes compared to the ranking shown in fig. 4 are observed.

**Details**  Each experiment is performed with 4 different seeds, therefore each ranking is obtained 4 times.

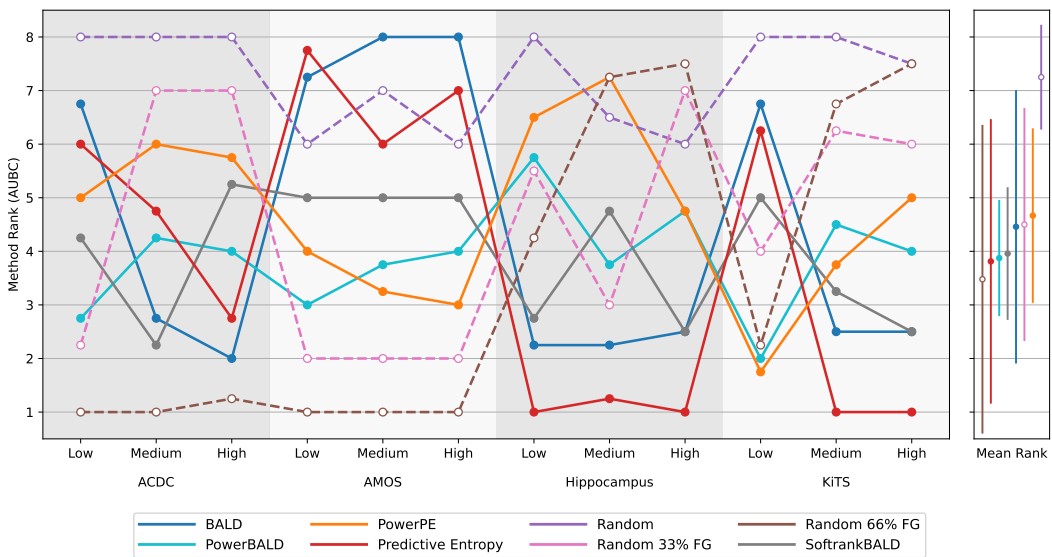

Figure 17: **Leave-One-Out Overview of Main Study Overview for AUBC.** Ranking of methods according to AUBC for each dataset and its Label Regimes (Low, Medium & High) alongside mean with standard deviations (bar).

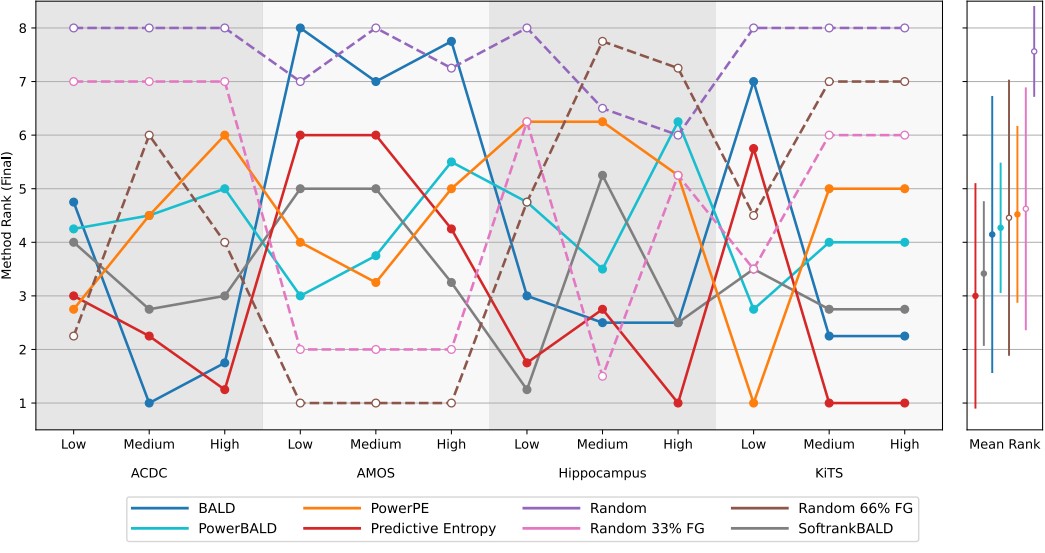

Figure 18: **Leave-One-Out Overview of Main Study for Final Dice.** Ranking of methods according to Final Dice for each dataset and its Label Regimes (Low, Medium & High) alongside mean with standard deviations (bar).

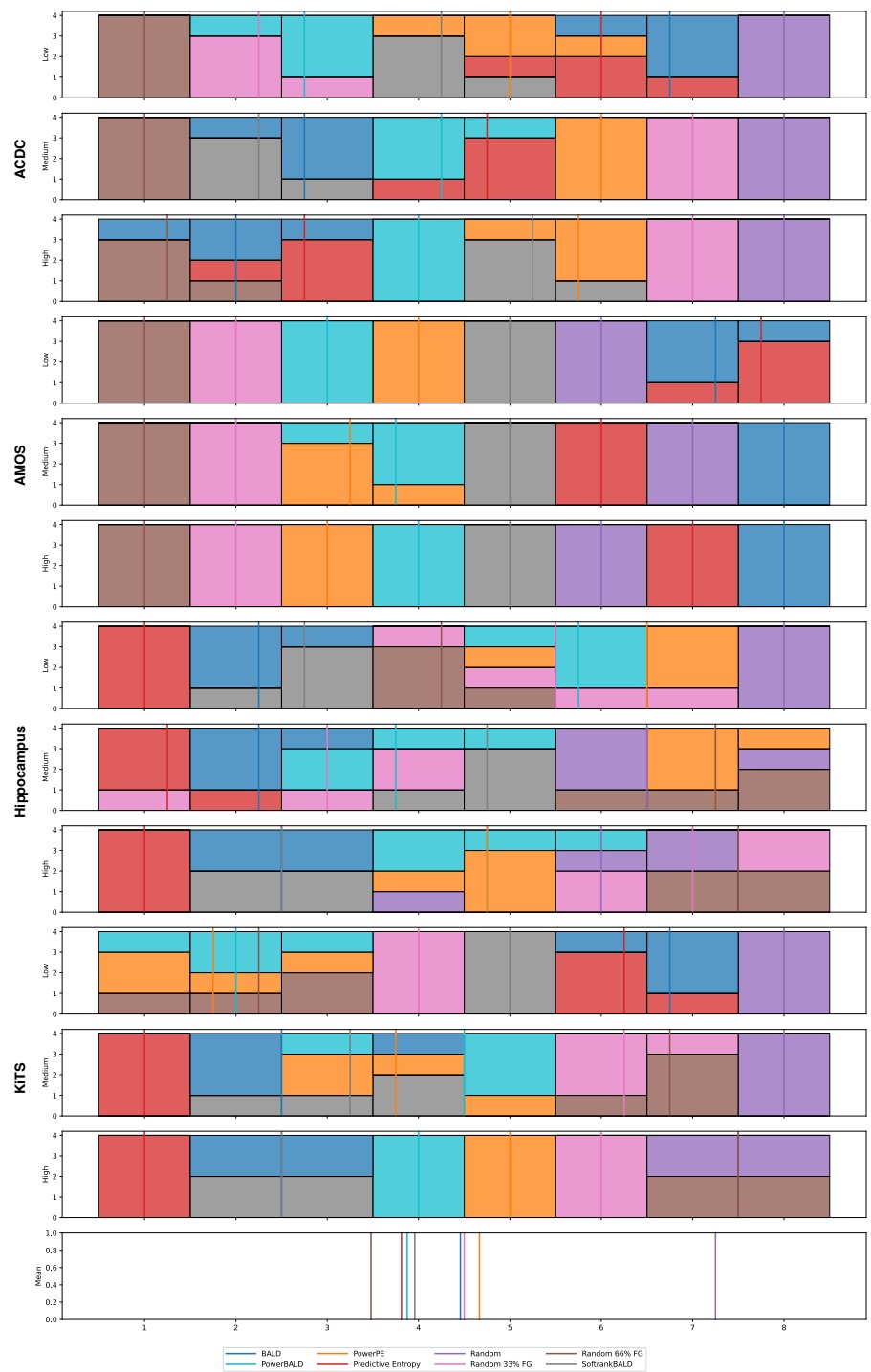

Figure 19: **Leave-One-Out Detailed Results of Main Study for AUBC.** Ranking of methods according to AUBC for each dataset and its Label Regimes (Low, Medium & High). A specific colored field of height 1 at x-axis $x$ denotes that for one of the four seeds the method corresponding to this color obtained in the leave-one-out (seed based) ranking place $x$.

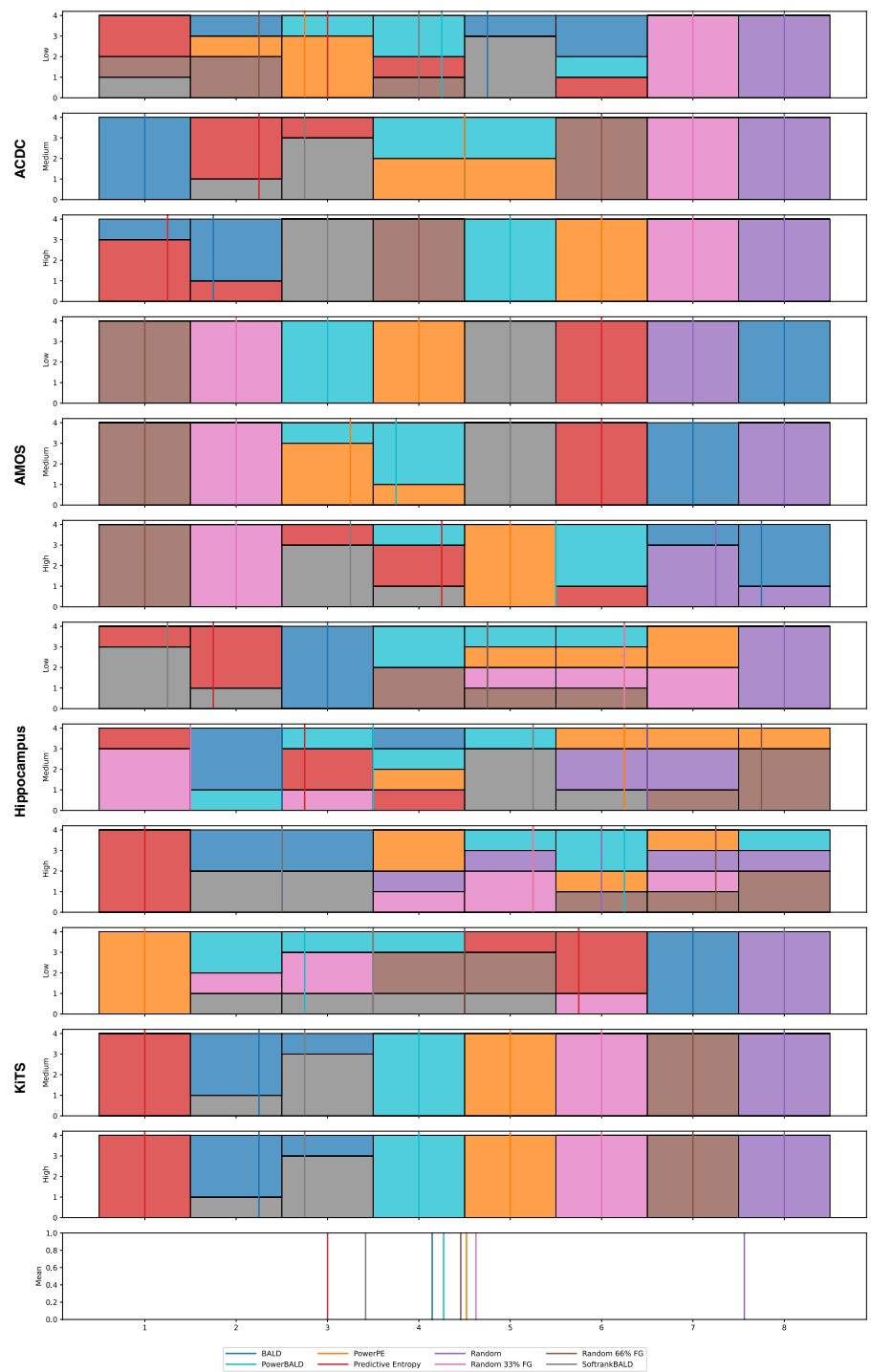

Figure 20: **Leave-One-Out Detailed Results of Main Study for Final Dice.** Ranking of methods according to AUBC for each dataset and its Label Regimes (Low, Medium & High). A specific colored field of height 1 at x-axis $x$ denotes that for one of the four seeds the method corresponding to this color obtained in the leave-one-out (seed based) ranking place $x$.

# I  Model Prediction Visualizations

We provide exemplary visualizations of the predicted segmentation masks for different QMs for ACDC (fig. 21), AMOS (fig. 22), Hippocampus (fig. 23), and KiTS (fig. 24) of the Main Study. We selected the following model configurations:

- ACDC (fig. 21): Low-Label setting of the main study (annotation budget: 150, query patch size: $4 \times 40 \times 40$); seed: 12347
- AMOS (fig. 22): Low-Label setting of the main study (annotation budget: 200, query patch size: $32 \times 74 \times 74$); seed: 12347
- Hippocampus (fig. 23): Low-Label setting of the main study (annotation budget: 100, query patch size: $20 \times 20 \times 20$); seed: 12345
- KiTS (fig. 24): Low-Label setting of the main study (annotation budget: 200, query patch size: $64 \times 64 \times 64$); seed: 12347

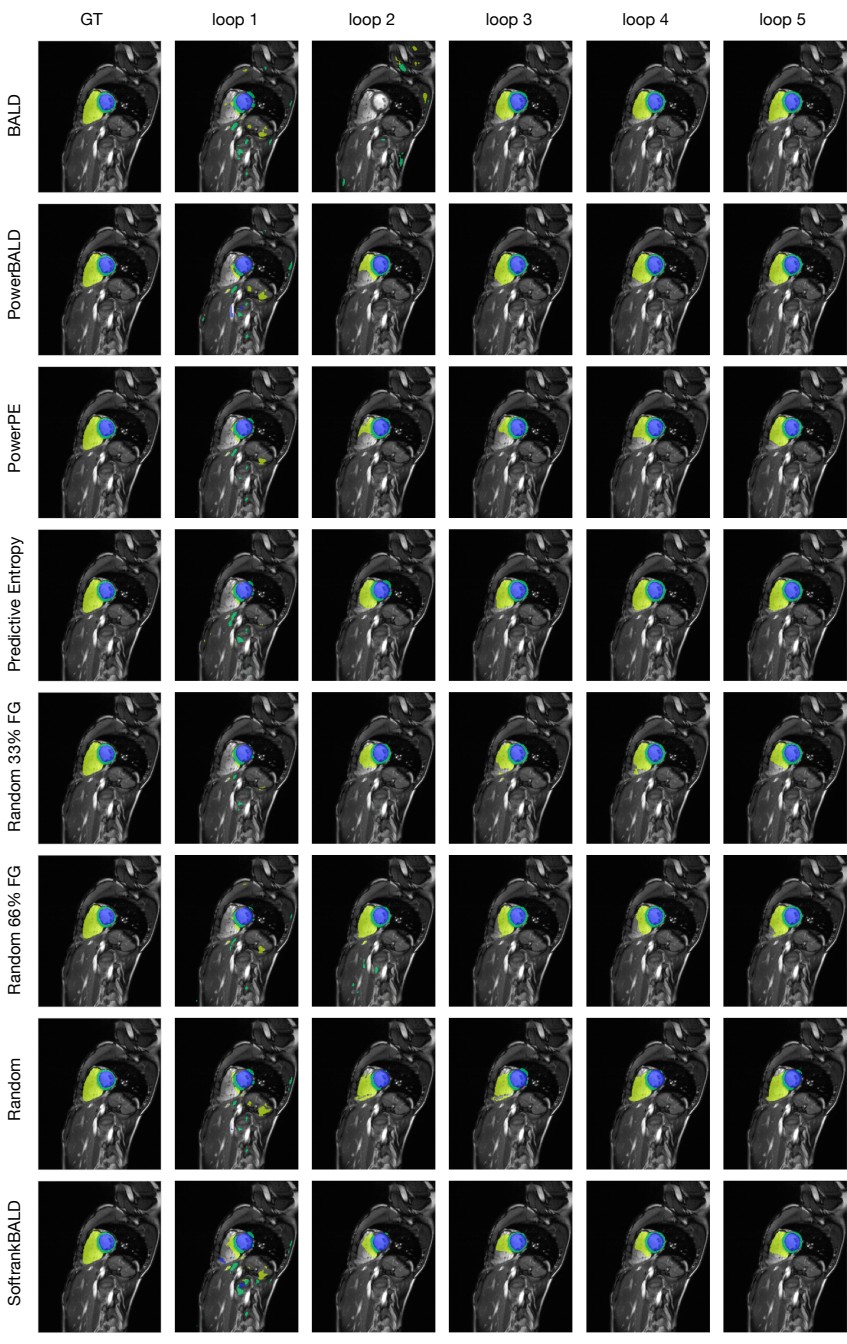

Figure 21: **Exemplary Model Predictions on ACDC for different QMs.** Column 1: Ground Truth (GT) segmentation masks; Column 2-6: predicted segmentations after each AL loop.

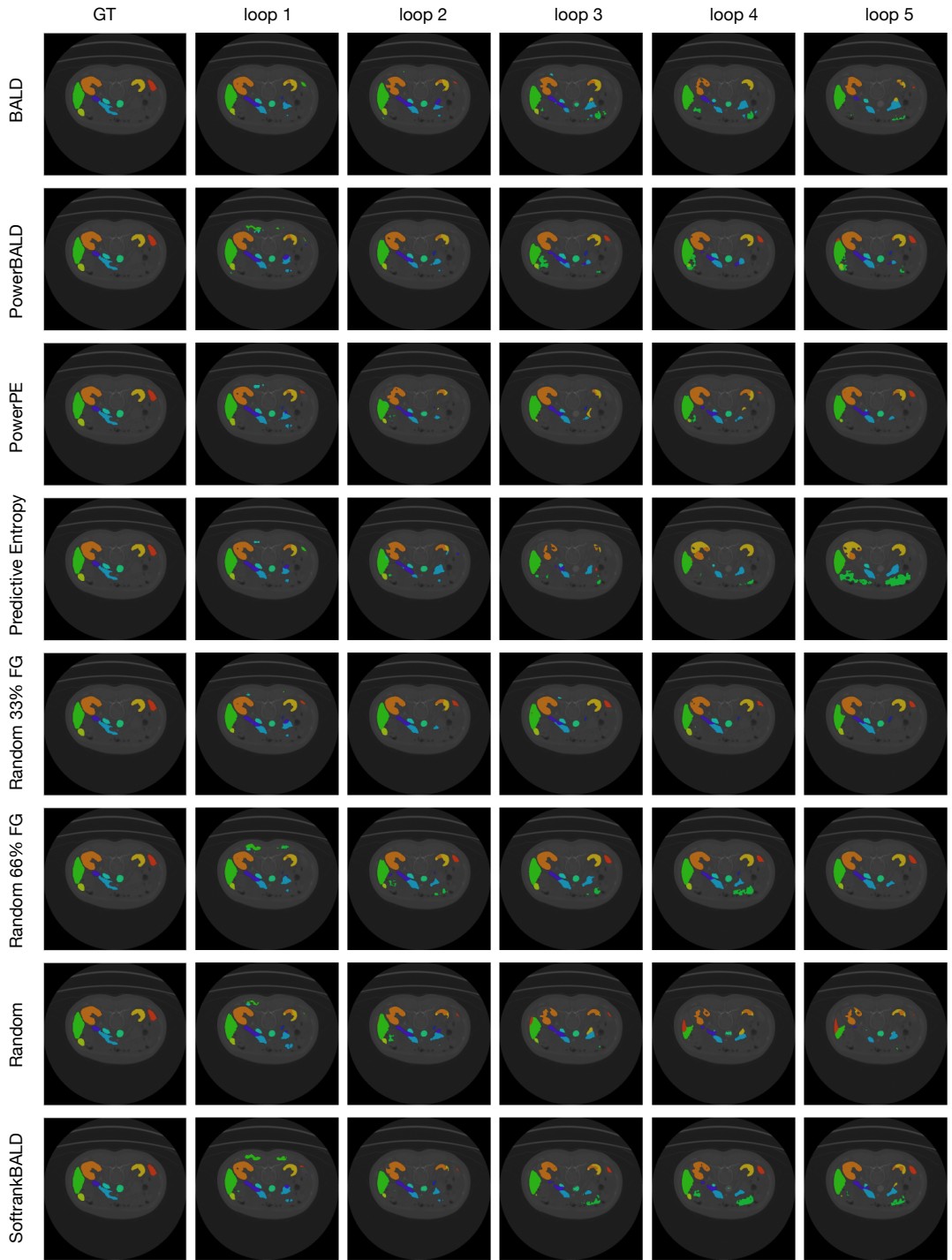

Figure 22: **Exemplary Model Predictions on AMOS for different QMs.** Column 1: Ground Truth (GT) segmentation masks; Column 2-6: predicted segmentations after each AL loop.

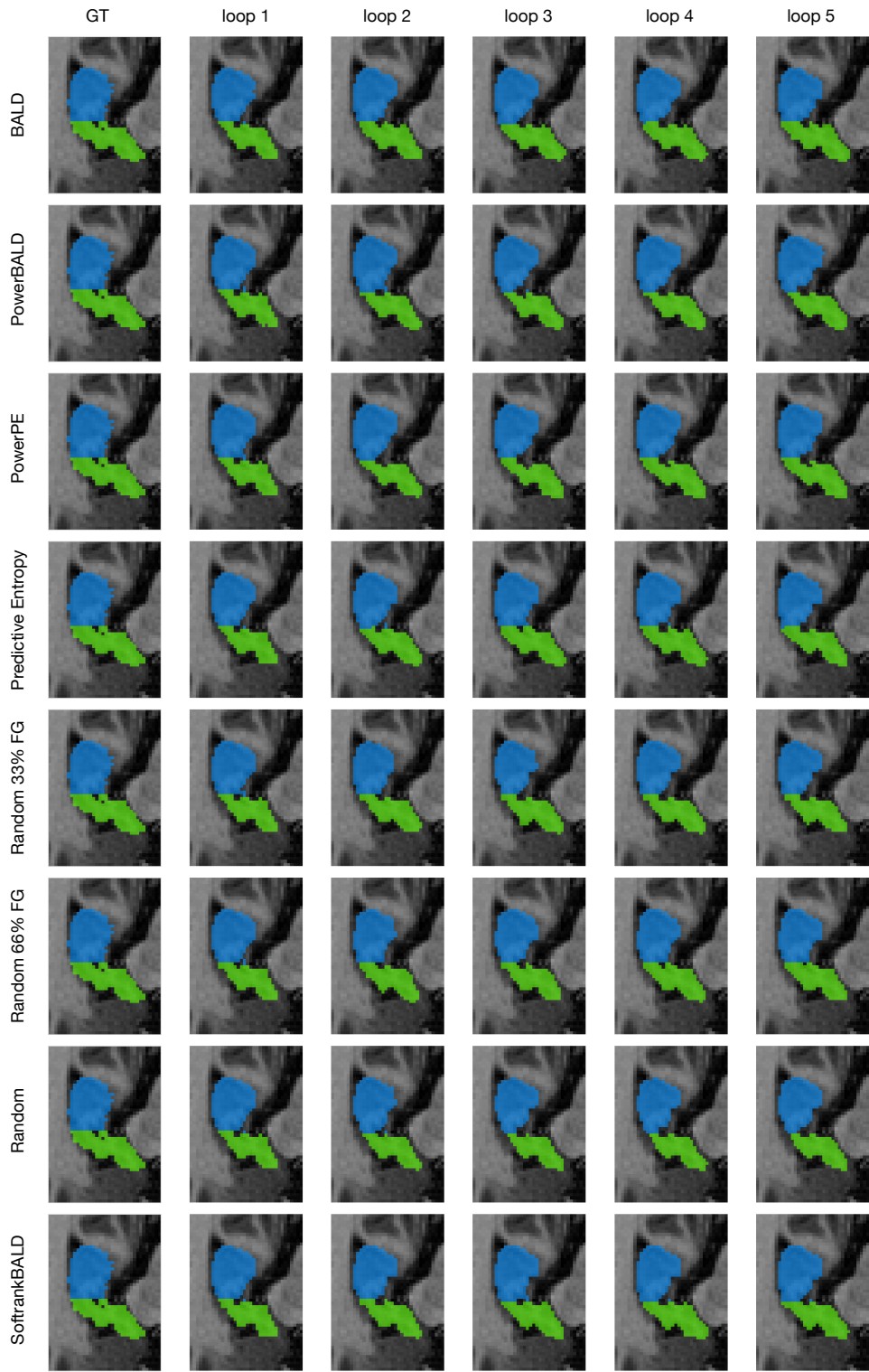

Figure 23: **Exemplary Model Predictions on Hippocampus for different QMs.** Column 1: Ground Truth (GT) segmentation masks; Column 2-6: predicted segmentations after each AL loop.

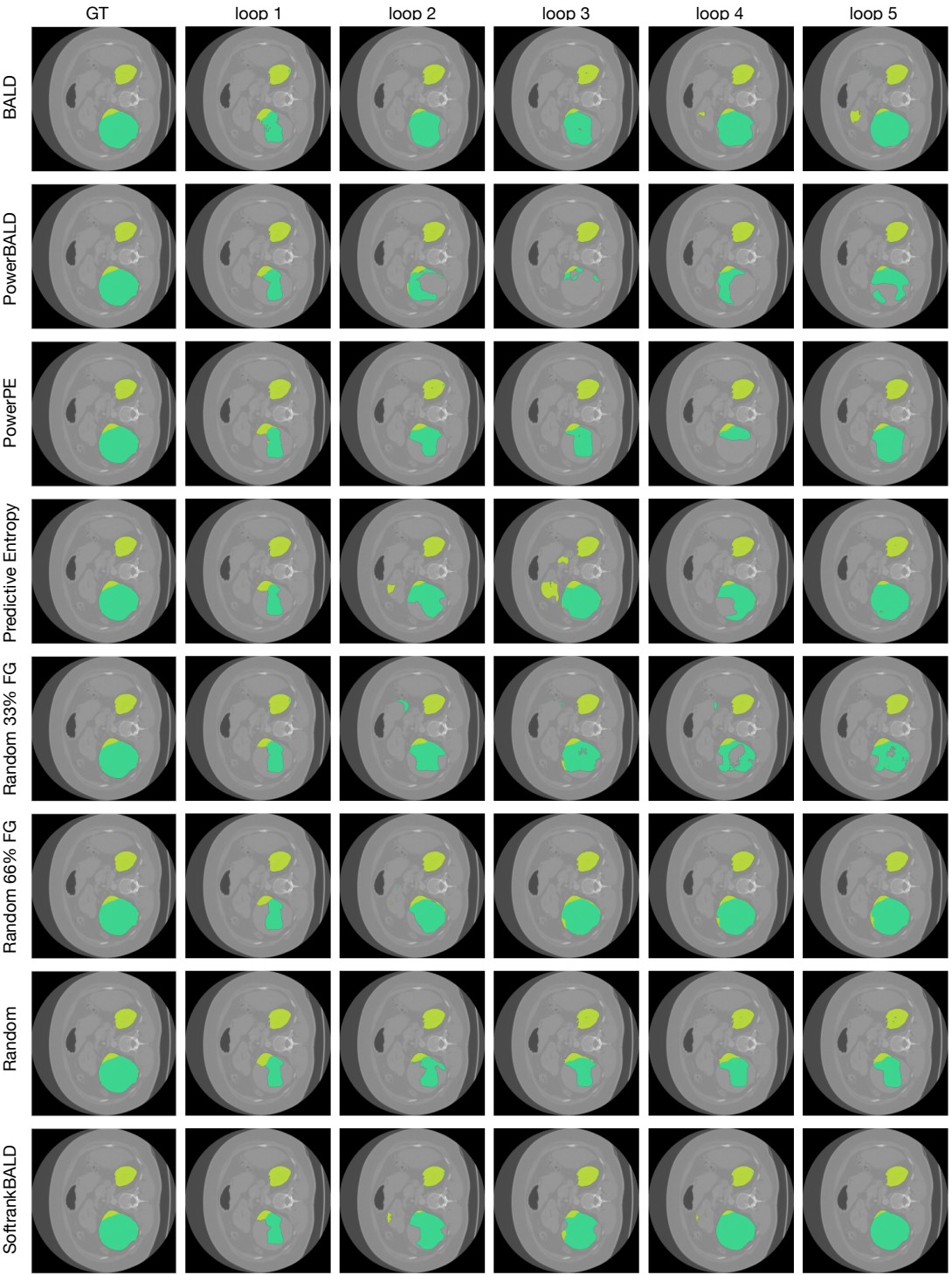

Figure 24: **Exemplary Model Predictions on KiTS for different QMs.** Column 1: Ground Truth (GT) segmentation masks; Column 2-6: predicted segmentations after each AL loop.

