# OpenReview forum: "nnActive: A Framework for Evaluation of Active Learning in 3D Biomedical Segmentation"
_TMLR — Accepted by TMLR_

### Review · Reviewer_By9h · 2025-06-18

**Summary Of Contributions:**

This work pinpoints four recurring pitfalls of AL methods in medical imaging: too few datasets/budgets are tested, 2-D models are trained on 3-D data without exploiting partial labels, random baselines are ill-suited to the foreground–background imbalance, and annotation “cost” is measured only in voxels rather than in true human effort.

The authors propose a framework for AL evaluation, including baselines, metrics, and benchmarks.

The authors run their framework on a suite of experiments and find that foreground-aware random sampling is tough to beat. Other insights were on best query methods, noisy vs. greedy selection, improvements with longer training, etc.

**Audience:**

Yes

**Claims And Evidence:**

Yes

**Requested Changes:**

- QM abbreviation is used but never defined.
- What does the model see during a single forward pass? Is it a cropped patch centered around a label? The authors say, "However, we enhanced the standard patch-based model trainer through region sampling, enriching the region observed by the model with additional unlabeled context from the rest of the image." Concretely, does this mean including x number of voxels of the input patch about every dimension of the label patch?

**Strengths And Weaknesses:**

Strengths:
- Very comprehensive experimentation that exhibits familiarity and respect of the medical imaging domain.
- Provides several insights and conclusions that can be taken away by the reader. I can imagine this paper being useful for practitioners who want to deploy AL setups in real-world contexts.
- Clear writing and easy to understand

Weaknesses:
- Regarding foreground-related arguments (including foreground-aware random sampling and FG-Eff), do the authors assume that the foreground can be readily extracted? I can imagine scenarios where the foreground is not clearly delineated. E.g. suppose one wants to segment structures (e.g. tumors) in one organ within an abdominal MRI scan. Can the authors comment on this?

---

> ### Author Response · Authors · 2025-07-02
>
> Thank you again for your valuable comments, and for taking the time to read our general reply, as well as considering our point-by-point comments here:
>
> ---
>
> **W1.** Foreground-Related Arguments – Do we assume foreground can be readily extracted in practice?
> * We agree that there are many instances where the foreground can not be readily extracted and we do not assume this to be the case. Regarding Foreground related arguments, the answer is different for Foreground-Efficiency and Foreground Aware Random sampling:
>   * **Foreground-Efficiency (FG-Eff)** is in its current formulation a pure benchmarking metric which is only functional when a model is present which gives an upper bound performance (e.g. in our case trained on the entire dataset) and not supposed to be used during application in the absence of such a model (as it necessitates in our case a completely annotated dataset).
>   * **Foreground Aware Random sampling:** In a practical use-case Foreground Aware Sampling / Random would indeed require a map of foreground readily extractable in order to be completely automated. However, it does not need to be completely automated as the task of finding singular foreground voxels for each patch around which a bounding box gets drawn is very fast and low in human effort.
>     * E.g., (a) detecting a foreground structure and then (b) clicking on it followed by an automated bounding box creation with an annotation tool is quite simple and most importantly fast and low in manual effort.
>     * This workflow could easily be implemented for the described example of abdominal MRI scans with tumors in one organ with the added benefit, that the area that needs to be parsed by the radiologist (or whoever selects the patches) is only the organ itself speeding up this process as not the entire image needs to be parsed.
>   * We see that this process does increase the overall human effort per selected patch compared to completely random sampling or Active Learning, but it is definitely feasible and has the advantage of being highly parallelizable allowing for very large batch sizes in difference to Active Learning where always a specific budget based on the current state needs to be queried.
>
> ---
>
> **Req. Ch. 1.** QM abbreviation used but not defined:
> * Thank you for pointing this out, we added the corresponding definitions to the Appendix and the Introduction.
>
> ---
>
> **Req. Ch. 2.** What does the model see during a single forward pass?
> * Short answer: Using our custom nnActive Trainer the model sees during training patches with at least one annotated voxel (generally more due to our dense patch annotations) somewhere inside. Concretely speaking, a maximum of the forward patch size -1 per dimension is given as context without annotations.
> * Detailed Answer (Training nnU-Net with partial loss): The patchsize nnU-Net uses for forward passes is fixed for each dataset based on the nnU-Net plans and configuration. Importantly the patchsize of nnU-Net is not necessarily identical to the query patchsize of nnActive. Going into the scenario of training with partial annotations:
>   * The standard nnU-Net Trainer selects a patch as follows: sample an annotated voxel (either random, or of a specific class) and construct the final patch for the forward pass centered around it.
>   * The custom nnActive Trainer selects a patch as follows: sample an annotated voxel (either random, or of a specific class) and construct based on sampling so that the final patch for the forward pass is randomly located within the final patch (following an uniform distribution).
> * We added this description to the Appendix in Section C.
>
> ---
>
> Thank you once more for your constructive feedback. As we believe to have resolved your comments, please let us know in case you have remaining suggestions to further increase the quality of our work.

---

### Review · Reviewer_aV1L · 2025-06-22

**Summary Of Contributions:**

Active learning is an important field for applying medical image segmentation methods in the medical domain, especially given that the annotation can be quite expensive and limited. In this paper, the author proposes nnActive, a highly configurable AL extension for nnUnet, utilizing 3D patch-level partial annotation. The other contributions are as follows:

1. Investigate the foreground-aware sampling method as a stronger and more realistic baseline in AL, which tackles the class imbalance in the 3D images. This proves to be a strong baseline.

2. Perform a large-scale analysis and experiments of different AL methods over 7500 nnU-Net trainings for more than 12 dataset settings.

3. To evaluate the AL performance, FG-Eff is proposed as a novel metric to measure the annotation efficiency. Such a metric takes into account that the annotation of the background actually takes negligible effort compared to the foreground.

**Audience:**

Yes

**Claims And Evidence:**

Yes

**Requested Changes:**

See the above

**Strengths And Weaknesses:**

This paper has the following metrics:

1. Comprehensive related works and comparison with the previous method to show the benefits of this paper.  For example, the section on the requirement of Active learning evaluation and the analysis of different related baselines undoubtedly demonstrates the unique advantage of this paper in its detailed evaluation across various aspects.  The author further proposes solutions to address the limitations point by point


2. This paper is well organized. The figures make it easier to understand the key concepts and compare with the previous method. Even the appendix is quite detailed and well-organzied.

3. The experiments conducted are pretty comprehensive and convincing. The main study answers the performance of the current AL methods compared to the random baseline and the foreground-aware random baseline; the latter one is quite inspiring for the community's following direction, as current AL methods fail to outperform this strong baseline consistently. This paper also analyzes how the budget and dataset will influence the AL performance,  which also makes the results more convincing the a more generalized manner. Sufficient ablation has been provided to supplement the main study.



Weakness.

As the author mentions, though this paper aims to be comprehensive, the mainly covers the uncertainty-based AL methods and does not cover the methods that change the architecture or the loss function. Therefore, I would like the author to reflect this in the title and introduction. The way it is organized now may express the wrong information.

The main takeaways in the final section can be more organized according to the topics. Right now it is a little bit scattered in different aspects.

---

> ### Author Response · Authors · 2025-07-02
>
> Thank you again for your valuable comments, and for taking the time to read our general reply, as well as considering our point-by-point comments here:
>
> ---
>
> **W1.** Our study mainly covers uncertainty-based AL methods.
> * We agree that an overgeneralization of our study to all AL methods may express the wrong information and therefore updated the Abstract, Introduction and Discussion & Conclusion section to better reflect that our current benchmark only incorporates uncertainty-based AL methods/QMs.
> * In combination with the already present Limitations Paraph in the Discussion & Conclusion, we believe this should convey the limitation of our study sufficiently.  As we state there:
> “Due to the depth and rigor of our evaluation, combined with several orthogonal improvements to the AL experiment design s.a. partial loss and queries in form of freely adaptable 3D patches, we focus our evaluation on uncertainty-based AL methods, which are widely used whilst not requiring changes in model architecture and training. …”
> * As the overall framework for evaluation can be easily extended to other AL methods which are not yet covered by our study, we would prefer to keep the title as it is right now. To ensure that the community is aware that the framework for evaluation is not limited purely to uncertainty-based AL methods as our results can easily be extended by the community since we will publish our code (which is accessible right now) alongside our exact results on HuggingFace and all details to extend and reproduce our results.
>
> ---
>
> **W2.** Reorganization of the main takeaways in the final section
> * We reorganized the empirical finding take aways in the Conclusion & Discussion (changes highlighted in orange), for more information we refer to the general response.
>
> ---
>
> Thank you once more for your constructive feedback. As we believe to have resolved your comments, please let us know in case you have remaining suggestions to further increase the quality of our work.

---

### Review · Reviewer_Y74B · 2025-06-23

**Summary Of Contributions:**

The paper presents nnActive, a unified evaluation framework for active learning (AL) in 3D biomedical image segmentation. Built upon the established nnUNet pipeline, nnActive introduces a modular and reproducible setup for simulating various AL scenarios. It includes support for uncertainty-based sampling, retraining strategies, and multiple datasets. The paper aims to standardize benchmarking in AL for biomedical segmentation by providing a comprehensive and extendable platform.

**Audience:**

Yes

**Claims And Evidence:**

Yes

**Requested Changes:**

1. Include additional statistical analysis (e.g., confidence intervals, significance tests) to substantiate empirical claims.
2. Provide more guidance or recommendations based on the experiments, e.g., when is AL beneficial and when not?
3. Discuss the computational overhead of retraining strategies and their implications in clinical settings.
4. A qualitative analysis (e.g., visualizations of queried samples, segmentation errors) could offer more insight into what the model learns via AL.
5. If feasible, add a real-world experiment with radiologists or annotators to validate simulation realism.

**Strengths And Weaknesses:**

Strengths:
1. The authors effectively solve an interesting and important problem: the lack of standardized AL benchmarks in 3D biomedical imaging is a genuine gap.
2. With nnUNet, a widely adopted baseline, enhances usability and promotes reproducibility.
3. The framework is tested across several datasets and active learning strategies, showcasing its flexibility and extensibility.
4. The code is promised to be made publicly available, which aligns with the goals of community benchmarking.

Weaknesses:
1. The core contribution is a well-structured benchmark framework, but not a new algorithm or methodological innovation in active learning or segmentation; thus, the novelty is limited in methodology.
2. While various AL strategies are implemented, the paper lacks deep theoretical or empirical exploration of why certain strategies perform better.
3. While the framework simulates AL, it’s unclear how well it maps to real clinical workflows with human-in-the-loop feedback. An experiment with real annotators would strengthen the claims.
4. The improvement margins of AL over random sampling are relatively small in several experiments, which may raise doubts about the practical benefit of AL in these settings.

---

> ### Author Response · Authors · 2025-07-02
>
> Thank you again for your valuable comments, and for taking the time to read our general reply, as well as considering our point-by-point comments here:
>
> ---
>
> **W1.** “the novelty is limited in methodology”
> * We want to highlight that, while we do not introduce a novel AL method, our contribution lies in addressing critical pitfalls in current evaluation practices for AL in 3D biomedical segmentation, alongside an extensive study which provides insights regarding the benefits of existing AL methods. We believe that establishing a rigorous and extensible development and evaluation framework is a necessary step for enabling future methodological innovations.
>
> ---
>
> **W2.** “the paper lacks deep theoretical or empirical exploration of why certain strategies perform better”
> * We acknowledge that a theoretical analysis of strategy performance is beyond the scope of this work. However, we provide intuitions on the behavior of different strategies as well as empirical exploration, as detailed in Req. Ch. 4.
>
> ---
>
> **W3 & W4.** As these two points directly correspond to Req. Ch. 2 & 5, we address them in the respective sections below.
>
> ---
>
> **Req. Ch. 1.** “Include additional statistical analysis (e.g., confidence intervals, significance tests) to substantiate empirical claims”
> * We thank the reviewer for highlighting the importance of robust statistical analysis and completely agree. All of our results are obtained from running experiments with 4 seeds and we compute the following statistics:
>   * Mean and Std by means of empiric estimation (AUBC, Final Dice) [e.g. Table 6 in App.]
>   * Mean and Std by means of an exponential fit (FG-Eff) [e.g. Table 6 in App.]
>   * Pairwise Penalty Matrix [e.g. Figure 2, 6, 9] is based on pairwise t-tests as described in App. D.3
>   * Mean and Std of rankings across all experiments [e.g. Fig. 4 bar]
>   * Kendall’s Tau Correlation with significance tests for ablations [e.g. Table 7, 8, 12 in App.]
>   * T-Test with alpha=0.1 for ablations [e.g. Table 11 in App. for training length ablation]
> * Additionally, we now include a leave-one-out ranking analysis showing the variability of rankings for each setting in App. G.
> * We would appreciate specific recommendations of tests or statistics to further strengthen our claims.
>
> ---
>
> **Req. Ch. 2.** “Provide more guidance or recommendations based on the experiments”
> * Thank you for bringing this up. We present our distilled Take-Away findings in the Conclusion. Further, we provide a dedicated section containing guidelines for practitioners and developers. Based on your suggestion, we reorganized the take-away findings in the Conclusion for the revised manuscript.
>
> ---
>
>
> **Req. Ch. 3.** “Discuss the computational overhead of retraining strategies and their implications in clinical settings”
> * Thank you for raising this point. Retraining introduces considerable computational overhead, therefore using AL must bring benefits in the average scenario. As we state in the introduction: “This reduction in annotation cost needs to be weighed against an increase in computational and setup cost stemming from the use of AL. Therefore, to justify a general recommendation for an AL method, it needs to reliably bring performance benefits [...] that are substantial enough to ensure amortization of its cost increases during application”. In clinical settings, annotation costs can far outweigh compute costs, making it a prominent use-case for AL as a reliable reduction of annotation effort already makes it worthwhile. However, we want to highlight that nnActive is not a product for clinical application, but serves as a basis for future AL method development and evaluation.
>
> ---
>
> **Req. Ch. 4.** A qualitative analysis (e.g., visualizations of queried samples, segmentation errors) could offer more insight into what the model learns via AL.
> * We agree with the reviewer. We already provide visualizations of the queries of AL methods in App. F (Fig. 10 and 12).
> Further, we are right now preparing visualizations of the model predictions (i.e. segmentation masks) for different AL methods at different AL stages. We plan to update the manuscript with the new visualizations by Monday and will notify you once we upload it.
>
> ---
>
> **Req. Ch. 5.** If feasible, add a real-world experiment with radiologists or annotators to validate simulation realism.
> * We agree that this would be an interesting experiment. Unfortunately, performing these experiments is not feasible, due to the extremely high costs of doing experiments in real-world clinical settings. Doing this once is already expensive but to obtain meaningful scientific measurements such experiments would need to be repeated multiple times. Would you like to see this addressed in the Future Work paragraph in the Conclusion?
>
> ---
>
> Thank you once more for your constructive feedback. As we believe to have resolved your comments, please let us know in case you have remaining suggestions to further increase the quality of our work.

---

> > ### Author Response · Authors · 2025-07-05
> > **Update Regarding Visualization (Req. Ch. 4)**
> >
> > We just uploaded the revised version with additional visualizations of the segmentation masks for each loop alongside ground truth for all Query Methods and for each dataset (ACDC, AMOS, Hippocampus and KiTS) in the newly created Appendix Section H.

---

### Author Response · Authors · 2025-07-02

We sincerely thank all reviewers for their valuable comments. The reviewers generally agreed on the added value of our work (“The authors effectively solve an interesting and important problem: lack of standardized AL benchmarks in 3D biomedical imaging is a genuine gap”, “this paper being useful for practitioners who want to deploy AL setups in real-world contexts” ,“Very comprehensive experimentation that exhibits familiarity and respect of the medical imaging domain.”, “experiments conducted are pretty comprehensive and convincing.”).

The core goal of nnActive was to build a research platform allowing for systematic development and evaluation of Active Learning Methods for 3D biomedical segmentation with all the best-practices implemented. It is not a product to be used in a clinical environment. In fact we do not deem AL to be ready for wide-spread clinical application yet, as the performance improvements over the foreground-aware random strategies are as of now not consistent enough. This finding leads to our recommendation towards foreground aware random strategies [in Conclusion, Paragraph: Guidelines, For Practitioners, point 1] since it does not require intermediate retraining.

We hope that nnActive will serve as a catalyst for developing future AL methods allowing developers to make use of our results and benchmark their methods with all the best-practices readily implemented.

We have thoroughly revised our manuscript to address the provided feedback and uploaded the revision, where we **highlight the modified parts in orange**. Changes include:
* added rankings obtained using a leave-one-out scheme to assess the stability of the rankings for each experimental setting of the main study. These results are now included in the newly created Appendix Chapter G.
* clearly stating the focus of the currently benchmarked AL methods in our study again as uncertainty-based in: Abstract, Introduction and Conclusion
* reorganization of the findings section in the Discussion & Conclusion

We believe these updates resolve the stated concerns. Please find our point-by-point answers in the respective reviewer sections.

---

### Decision · Action_Editor_KcT8 · 2025-08-03

**Recommendation:** Accept as is

**Additional Comments:**

N/A

**Audience:**

Yes

**Audience Explanation:**

Yes, a subcommunity of TMLR, particularly focusing on active learning in medical imaging would be interested on this benchmark.

**Claims And Evidence:**

Yes

**Claims Explanation:**

This paper addresses an important gap by introducing a well-structured benchmark for active learning in 3D biomedical imaging, which is supported by comprehensive experiments and strong practical relevance. The claims made in this work are supported by the extensive empirical experiments. Furthermore, its clarity, flexibility and extensibility, as well as its usefulness to particularly the community justify acceptance. Finally, the use of standard baselines and commitment to open-sourcing further enhance its value.